# Bright ligand-activatable fluorescent protein for high-quality multicolor live-cell super-resolution microscopy

Jiwoong Kwon [1,9], Jong-Seok Park[2,8,9], Minsu Kang[1,3], Soobin Choi[4], Jumi Park[5], Gyeong Tae Kim [6], Changwook Lee[5], Sangwon Cha [4], Hyun-Woo Rhee[2,7]* & Sang-Hee Shim [1,3]*

We introduce UnaG as a green-to-dark photoswitching fluorescent protein capable of high-quality super-resolution imaging with photon numbers equivalent to the brightest photo-switchable red protein. UnaG only fluoresces upon binding of a fluorogenic metabolite, bilirubin, enabling UV-free reversible photoswitching with easily controllable kinetics and low background under Epi illumination. The on- and off-switching rates are controlled by the concentration of the ligand and the excitation light intensity, respectively, where the dissolved oxygen also promotes the off-switching. The photo-oxidation reaction mechanism of bilirubin in UnaG suggests that the lack of ligand-protein covalent bond allows the oxidized ligand to detach from the protein, emptying the binding cavity for rebinding to a fresh ligand molecule. We demonstrate super-resolution single-molecule localization imaging of various subcellular structures genetically encoded with UnaG, which enables facile labeling and simultaneous multicolor imaging of live cells. UnaG has the promise of becoming a default protein for high-performance super-resolution imaging.

[1] Center for Molecular Spectroscopy and Dynamics, Institute for Basic Science (IBS), Seoul 02841, Republic of Korea. [2] Department of Chemistry, Ulsan National Institute of Science and Technology (UNIST), Ulsan 44919, Republic of Korea. [3] Department of Chemistry, Korea University, Seoul 02841, Republic of Korea. [4] Department of Chemistry, Hankuk University of Foreign Studies, Yongin 17035, Republic of Korea. [5] Department of Biological Sciences, Ulsan National Institute of Science and Technology (UNIST), Ulsan 44919, Republic of Korea. [6] Department of Biomedical Engineering, Ulsan National Institute of Science and Technology (UNIST), Ulsan 44919, Republic of Korea. [7] Department of Chemistry, Seoul National University, Seoul 08826, Republic of Korea. [8] Present address: SK Biopharmaceuticals Co., Ltd.,, Daejeon 34124, Republic of Korea. [9] These authors contributed equally: Jiwoong Kwon, Jong-Seok Park. *email: rheehw@snu.ac.kr; sangheeshim@korea.ac.kr

Single-molecule localization (SML) microscopy such as STORM[1], (F)PALM[2,3] and PAINT[4] overcomes the optical resolution limit by using fluorescent probes capable of on–off switching of fluorescence emission. Fluorescent proteins (FPs) are popular for live-cell SML imaging due to the simplicity and specificity of labeling. However, FPs have limitations in robust multicolor SML imaging despite many years of probe development[5–8]. EosFPs, the brightest FPs, convert the fluorescence emission from green to red upon UV illumination[9,10]. The photoconvertible proteins occupy two color channels, leaving little window for the second FP in the visible spectrum[10]. On the other hand, photoactivatable proteins that are activated from a dark state have lower photon numbers than EosFPs, thereby lower localization precisions[11–13]. Spectrally similar FPs can be distinguished by spectral demixing, but the high color crosstalks can result in false positives[14]. FPs with different activation photochemistry can be separated by activation light sequences[15], which slow down the imaging speed. Therefore, a new FP with improved photophysics is still desired for multicolor SML imaging.

Synthetic dyes, superior to FP in terms of brightness, photostability and color option, face challenges in labeling live cells. A protein of interest (POI) in a live cell can be labeled with an organic dye via self-labeling enzymes such as FlAsH, SNAP, CLIP and Halo[16–19]. However, these labeling schemes suffer from high background fluorescence from non-specific binding of organic dyes to the substrate and cellular membranes. The background issue has been addressed by developing fluorogenic substrates for the self-labeling enzymes or fluorogen-activating proteins such as CRABPII, PYP and FAST[20–22]. However, synthetic materials should be developed to exhibit cell-permeable characters or delivered by perturbative means such as injection, electroporation or lipofection, thereby impeding their application to in vivo imaging. Thus, it is highly desired to find a probe that fulfills all requirements for live-cell SML imaging, particularly combining the labeling advantages of FPs and excellent photophysics of organic dyes.

UnaG is the first fluorogenic ligand-activated protein that emits green fluorescence with high quantum yield on par with EGFP. The ligand binds to UnaG via multiple noncovalent interactions, in contrast to conventional FPs whose chromophores are constitutive of the protein residues or covalently linked to the proteins (Supplementary Fig. 1)[23,24]. The ligand for UnaG, bilirubin (BR), is an endogenous metabolite produced in the heme catabolism, and absorbs blue light efficiently, but does not emit fluorescence. As a biological antioxidant, the oxidation process of BR has been extensively investigated[25–27]. Light-mediated reactions of BR, such as photo-isomerization and photo-oxidation, have long been studied due to its clinical importance[28–31].

UnaG strongly fluoresces only when it binds with BR (Fig. 1a)[23]. The fluorogenic center of ligand-bound UnaG (holoUnaG) is considered as BR rather than the protein, because the protein without the ligand (apoUnaG) is not fluorescent. The high contrasts between holoUnaG and apoUnaG and between holoUnaG and BR, as well as the highly specific binding between BR and UnaG, allowed efficient sensing of BR in vitro and in cells[32,33]. The application area of UnaG was further extended to monitor the activity of membrane transporters[34], to regulate the activity of a POI[35], to report the protein–protein interactions[36], to investigate hypoxia states of cells[37,38], and to monitor calcium and BR simultaneously[39]. Despite these various applications as sensors, UnaG has not been extensively used in general imaging applications probably due to the sensitive photobleaching through BR photo-oxidation, which we characterized and utilized in this article.

Here, we report the photo-switchable nature of holoUnaG with investigation on the switching kinetics and mechanism as well as application to super-resolution imaging. The off-switching reaction of photo-oxidation is reminiscent of PALM/STORM and the replenishable on-switching reaction of rebinding is similar to PAINT. The characterization of the on-off transitions allows us to optimize the switching rates and to acquire high localization precisions and spot densities in standard Epi-illuminated SML super-resolution imaging of various subcellular structures. We demonstrate live-cell SML imaging up to three colors with high resolutions and virtually no color crosstalk.

## Results

**Reversible switching of fluorescent holoUnaG.** When a sample of purified holoUnaG proteins was irradiated with intense 488-nm laser, the fluorescence emission was switched off (Fig. 1b, red lines). Then, upon addition of external BR, the bleached fluorescence was spontaneously recovered (Fig. 1b, blue lines). Interestingly, this fluorescence depletion and recovery could be repeated when excess BR was added into the solution (Fig. 1c). Here the imperfect fluorescence recovery might come from the insufficient incubation time for holoUnaG transition to the brighter state[40]. From these observations, we assumed that the photobleaching of holoUnaG causes damage to the ligand, not to the protein[41]. Due to the highly specific interaction and the lack of ligand–protein covalent bond, a damaged BR may be easily displaced from UnaG (Supplementary Fig. 1c). Note that FP chromophores, except for BR in holoUnaG, are covalently attached to the proteins and cannot be removed from the proteins after photobleaching (Supplementary Fig. 1)[24]. When undamaged BRs are around the freely exposed apoUnaG, a fresh BR molecule can rebind to the empty protein and emit fluorescence again. In contrast, the covalently bonded chromophores in photo-switching FPs undergo photo-induced chemical reactions such as isomerization and bond breakage (Supplementary Fig. 1d, e).

Based on these observations, we tested SML capability of UnaG in a fixed mammalian cell transfected with UnaG-Sec61β targeted to endoplasmic reticulum (ER) with a standard Epi-illuminated widefield instrument, and analyzed the localization statistics under different buffer conditions (Fig. 1d–i). After taking a widefield image with a low excitation intensity of 488-nm laser (Fig. 1d), we increased the intensity to ~300 W cm$^{-2}$ and observed the on- and off-switching events of single UnaG molecules. In a simple buffer, only a limited number of switching events were detected (Fig. 1e) because the fluorescence of many single molecules per camera frame was too weak to detect above the signal-to-noise threshold (Fig. 1h). When external BR was supplemented to the buffer, the number of switching events increased notably (Fig. 1f). When an oxygen scavenger (OS) system was supplemented in addition to BR, the single-molecule images became brighter (Fig. 1h), allowing SML imaging with UnaG protein (Fig. 1g). Note that we could obtain a series of localization dataset from the same field of view, thanks to the reversible switching of UnaG. The photon counts were increased upon oxygen depletion (Fig. 1h), and the BR supplementation increased the number of detected switching events (Fig. 1i). These results indicate that the off-switching reaction from the fluorescent state to the dark state is related to the concentration of dissolved oxygen while the on-switching reaction is associated to the concentration of BR.

**Fluorescence switching kinetics.** To characterize the switching reactions, we investigated the switching kinetics of UnaG proteins

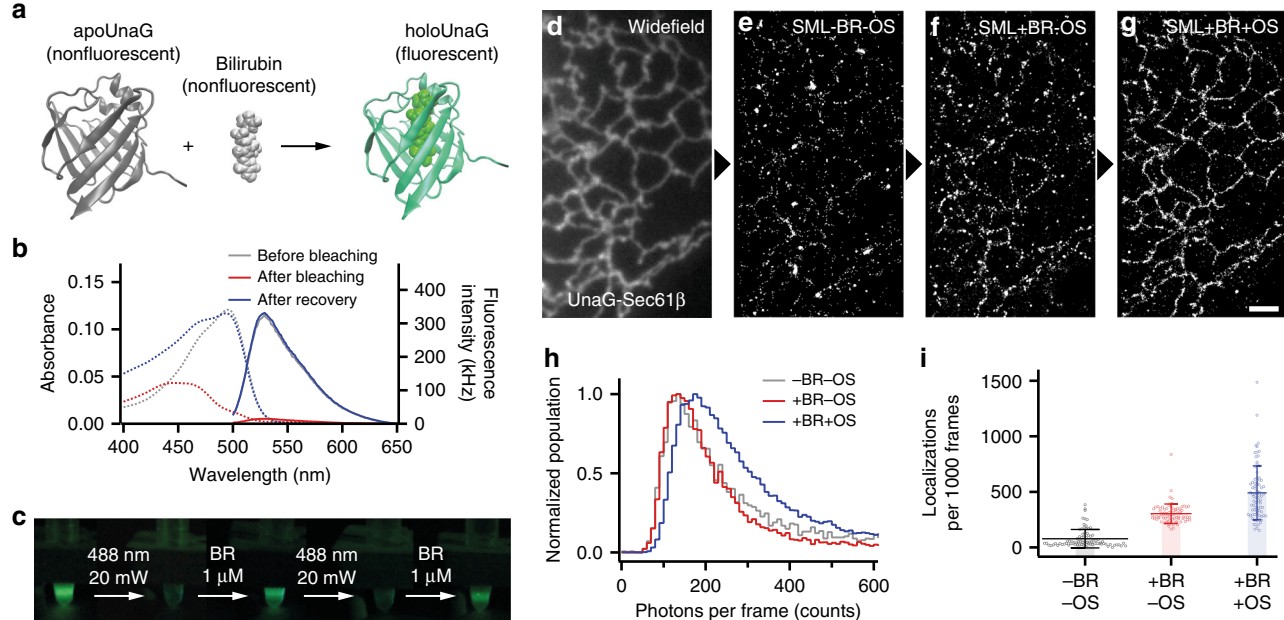

**Fig. 1 Repetitive on- and off-switching of holoUnaG by external bilirubin (BR) and light exposure. a** Scheme of BR-inducible fluorescence of UnaG. Crystal structures of apoUnaG, BR and holoUnaG are obtained from Protein Data Bank (PDB, ID: 4I3B). **b** Absorption (dashed lines) and fluorescence emission (solid lines) spectra of holoUnaG before bleaching (gray), after bleaching (red) and after recovery (blue). Abnormally large absorbance at < 460 nm in the blue dashed line may come from the additional BR at 1 μM for the recovery. **c** Repetitive photobleaching and fluorescence recovery of holoUnaG. Each picture was captured with 200 μW of 488-nm excitation light after indicated treatments for 60 min each. **d–g** Widefield image and series of localization dataset of UnaG-Sec61β transiently expressed in a fixed Cos7 cell in different imaging buffer compositions. Raw data for single-molecule localization were recorded with 10 ms camera exposure time for 30,000 frames. The same field of view was observed sequentially from the left (**d**) to the right (**g**) under identical optical conditions for the localization datasets. **h** Normalized photon counts distribution and **i** the average number of localized molecules from the localization datasets in **d–g**. The photon counts distributions were normalized by their own maximum counts. Scale bar: 2 μm. Error bars: standard deviations ($n = 80$). BR bilirubin, OS oxygen scavenger.

in both directions between the fluorescent and non-fluorescent states by using single-molecule pull-down assay[42]. Under continuous illumination of 488 nm at a certain intensity, the fluorescence decreased in time with bi-exponential behaviors (Fig. 2a), where the bi-exponential fits yielded two off-switching rate constants ($k_{off1}$ and $k_{off2}$). Higher light intensity accelerated both of the bleaching rates (Fig. 2b), and the linear fits gave two first-order rate constants of $2.0 \times 10^{-2}$ cm$^2$ W$^{-1}$ s$^{-1}$ ($R^2 = 0.98$) for the slower rate ($k_{off1}$) and $1.21 \times 10^{-1}$ cm$^2$ W$^{-1}$ s$^{-1}$ ($R^2 = 0.99$) for the faster rate ($k_{off2}$). These off-switching rates were unaffected by various buffer supplements and pHs, except for an OS (Fig. 2c, d). Depleting the dissolved oxygen using glucose oxidase (GLOX) significantly slowed down both the off-switching reactions (Fig. 2c), suggesting that oxygen participates in the off-switching reactions. Neither β-mercaptoethylamine (MEA) nor potassium iodide (KI) affected the off-switching rates, indicating that the off-switching processes are not related to the intersystem crossing (Fig. 2c)[43,44]. Addition of sodium ascorbate (Ascb), which is known to perturb the concentration of reactive oxygen species (ROS) in the imaging buffer, did not change the off-switching rates considerably[45]. Also, the off-switching rates with GLOX were nearly the same with and without catalase (CAT), indicating hydrogen peroxide ($H_2O_2$) did not influence the reaction[46]. In summary, oxygen concentration was the only factor to significantly alter the off-switching rates among all the buffer conditions that we tested.

Bleached UnaG could recover its fluorescence only when additional BRs were supplemented to the solution (Fig. 2e). The fluorescence recovery after adding BR also showed bi-exponential behavior due to the two different, but spectrally indistinguishable, fluorescent forms of holoUnaG[40], including a less bright state

(holoUnaG1 in Supplementary Fig. 2a) that is formed immediately after apoUnaG binds to BR and a brighter state (holoUnaG2 in Supplementary Fig. 2a) that is formed via a reversible conversion of the less bright state (Fig. 2e)[40]. To elicit the on-switching rates from the experimental results, we derived the analytical solution for the complex fluorescence switching model (Supplementary Note 1 and Supplementary Fig. 2), which was well fitted to our results and provided the on-switching rate constants (Fig. 2e, solid lines). Higher concentration of BR linearly accelerated the binding of BR to apoUnaG ($k_{on} = k_{ah}'$ in Supplementary Fig. 2a), and the linear fit yielded 0.12 μM$^{-1}$ s$^{-1}$ ($R^2 = 0.98$) for the first-order kinetics (Fig. 2f) with averaged reaction rate constants of 0.002 s$^{-1}$ and 0.008 s$^{-1}$ for the spontaneous transitions between holoUnaG1 and holoUnaG2 (i.e. $k_h$ and $k_h'$ in Supplementary Fig. 2a, respectively). In contrast, the fluorescence emission remained near the background level when external BR was not supplemented (Fig. 2e, gray line) and when photo-damaged BR (dmBR) was added (Fig. 2e, purple dashed line). Adding dmBR to existing fresh BR also did not affect the on-switching rate (Supplementary Fig. 3, blue solid line).

From the kinetic studies, we concluded that the photo-oxidation of BR is the major cause of the photobleaching of holoUnaG since only the concentration of dissolved oxygen significantly affected to the off-switching. The interaction between UnaG and BR is known to be highly specific. Even structurally very close compounds, such as conjugated BR species and biliverdin (BV) (Supplementary Fig. 1b), do not bind to apoUnaG[23]. Hence, photo-oxidation of BR may lead to the breakage of some ligand-protein interactions and subsequent dissociation of the BR oxidation products from UnaG, emptying

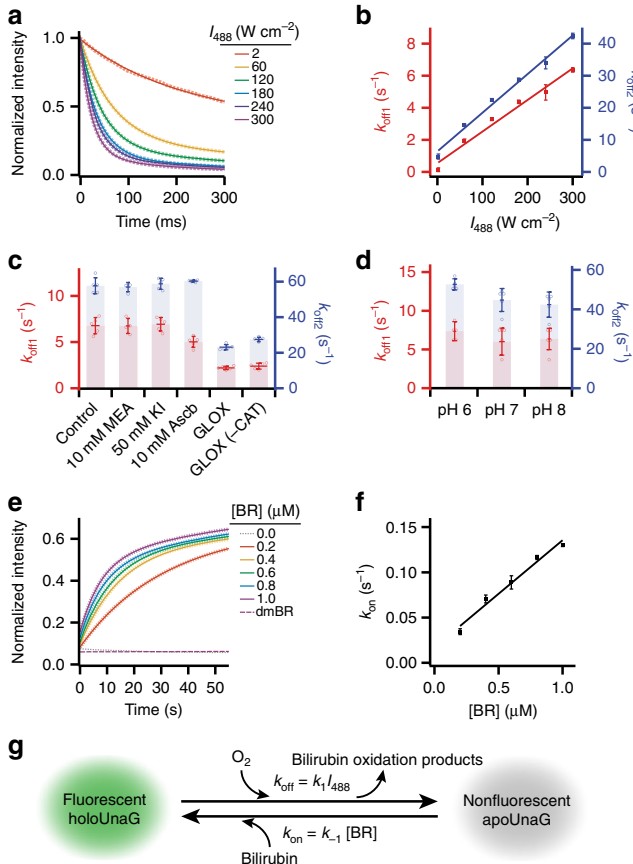

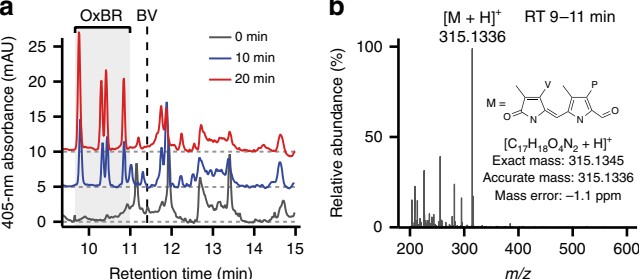

**Fig. 3 Separation and mass spectrometry analysis of the major photo-oxidation products. a** UV/vis chromatograms, at 405 nm, of photo-oxidation products of BR (OxBR) extracted from photobleached holoUnaG. For guidance, each chromatogram was offset by 0, 5 and 10 for irradiation times of 0 (gray), 10 (blue) and 20 min (red), respectively. Vertical black dashed line marks the retention time for BV obtained from a control experiment (Supplementary Fig. 4b, c). **b** An averaged mass spectrum for the retention time (RT) 9–11 min region of the LC–HRMS analysis. The most abundant ion species at $m/z$ 315.1336 could correspond to the protonated ion ($[M + H]^+$) of the possible oxidation product (M) inserted as an inset. P propionic acid (-CH2CH2COOH), V vinyl (-CH=CH2).

By summarizing the kinetics results, we propose a simplified two-state fluorescence switching model for UnaG (Fig. 2g). The fluorescent holoUnaG is bleached under light exposure by the photo-oxidation of BR, and the damaged BR displaces from UnaG (Fig. 2g, forward reaction). The $k_{off}$ can be controlled by both the light intensity and the concentration of dissolved oxygen, with the overall rate constant (i.e. $k_{off} = [a_1 k_{off1}/(a_1 + a_2)] + [a_2 k_{off2}/(a_1 + a_2)]$) of 0.10 cm² W⁻¹ s⁻¹ for the atmospheric oxygen level and 0.02 cm² W⁻¹ s⁻¹ for the oxygen-depleted condition by GLOX, when 488-nm laser is used. The dark apoUnaG can restore its fluorescence by rebinding with another, undamaged BR (Fig. 2g, backward reaction). The $k_{on}$ can be controlled by the concentration of BR, with a rate constant of $1.2 \times 10^5$ M⁻¹ s⁻¹.

**Photo-oxidation products of UnaG-encapsulated BR.** To elucidate the reaction mechanism of the photo-oxidation of holoUnaG, we performed liquid chromatography (LC)–UV/vis and LC/mass spectrometry (MS) analysis to identify the major photo-oxidation products (Fig. 3). As increasing the light irradiation time, a few oxidation products with 405-nm absorbance were observed in the retention times of 9–11 min (Fig. 3a). BV, a well-known oxidation product of BR, was almost absent in LC chromatogram (Fig. 3a and Supplementary Fig. 4b, c). Instead, we noticed one major set (Fig. 3a, OxBR) of oxidation products whose amounts significantly increased with longer light exposures. The major ion signals for OxBR were detected around at $m/z$ 315.1 (Supplementary Fig. 4d) and accurate mass of these ions was found to be $m/z$ 315.1336 by high-resolution MS (HRMS, Fig. 3b). HRMS/MS analysis of the ion corresponding to $m/z$ 315.1336 via collision-induced dissociation yielded fragment ions (Supplementary Fig. 4e, f) that allowed us to predict the chemical structure in Fig. 3b. OxBR has fully conjugated bipyrrole rings segmented from either left or right half of BR (Supplementary Fig. 5), whose highly conjugated structure is consistent to its UV-absorbance (Fig. 3a). OxBR has *cis/trans* isomers and structural isomers in which the vinyl group swaps the position with the methyl group on the same ring, resulting in four consecutive peaks in the LC–UV/vis chromatogram and in the extracted ion chromatogram (EIC) (Fig. 3a and Supplementary Fig. 4d).

**Fig. 2 Kinetic study of the photo-switching process. a** Time-dependent fluorescence decay of holoUnaG under different irradiation intensity of the 488-nm laser source. The fluorescence intensities were recorded with 5-ms time resolution. Dotted lines are the averaged low decay curves, and solid lines indicate bi-exponential fits to the decay curves. **b** Off-switching rates ($k_{off1}$ and $k_{off2}$) of holoUnaG under various laser intensities obtained from **a**. Linear fits (solid lines) gave first-order kinetic rate constants for the bleaching reactions. **c** Off-switching rates at 300 W cm⁻² of intensity in different imaging buffer supplements such as β-mercaptoethylamine (MEA), potassium iodide (KI) and ascorbic acid (Ascb). Only the oxygen scavenging systems based on glucose oxidase and catalase (GLOX and CAT) significantly slowed down the fluorescence decay, indicating that the major photo-degradation pathway of holoUnaG is the photo-oxidation. **d** Off-switching rates at 300 W cm⁻² of intensity in different pH conditions. Little notable differences were observed. **e** Time-dependent fluorescence recovery of bleached holoUnaG in various concentration of external BR from 0.0 to 1.0 μM (dotted lines) and in photo-damaged BR (dmBR, purple dashed line). Solid lines indicate the fitting results using Supplementary Equation 14. The fluorescence intensities were recorded with 1-s time resolution. **f** On-switching rates ($k_{on}$) of holoUnaG in different BR concentrations obtained from **c**, displayed with a linear fit. **g** Proposed two-state fluorescence switching model of holoUnaG. Light-induced oxidation of BR inside of holoUnaG by dissolved oxygen turns off the fluorescence, whereas the reverse reaction is purely affected by freely diffusing BR in solution. Thus, the switching rates in both directions can be controlled individually by either the light intensity or the BR concentration. Intensities in **a** and **e** were normalized by the initial value. Error bars: standard deviations ($n = 5$, from independent bulk measurements of pulled-down samples).

the binding pocket. If there is no damage to the protein during the oxidation reaction, apoUnaG can capture freely diffusing, undamaged BR to fluoresce again.

There are a number of previous studies on the reaction mechanism of BR oxidation[25–31]. Our proposed structure for OxBR was also reported in the previous studies on chemical or light-induced oxidation of BR[26,27]. In Supplementary Fig. 5, we propose the reaction mechanism of the photo-oxidation reaction for generating OxBR. Previous studies reported that excited BR can react with ROS such as singlet oxygen ($^1O_2$), superoxide radical ($O_2^{•−}$), $H_2O_2$ and hydroxide ion ($OH^−$), to form BV or radical species of BR[25–27,29,47]. Since BV was not detected in our LC analysis (Fig. 3a), we ruled out BV formation and we hypothesized that $^1O_2$ or $O_2^{•−}$ can further oxidize the reactive BR radicals via 1,2-cycloaddition forming four-membered rings, which can readily fragment into two aldehyde species (Supplementary Fig. 5)[26]. Each pyrrole unit in BR forms one or more hydrogen bonds (H-bonds) with UnaG, and the loss of any pyrrole unit results in the loss of the related H-bonds (Supplementary Fig. 1c). When we produced BR fragments outside the protein[26], the photo-damaged BR solution failed to recover fluorescence (Fig. 2e, purple dashed line), indicating that the reduced number for H-bonding groups are insufficient for binding to UnaG. Likewise, the reduced H-bonds between the fragmented photo-oxidation products in UnaG may lead to the dissociation from the protein. Since the two different conformations of holoUnaG proteins contain the same BR chromophore, one oxidation reaction of BR may give rise to two different off-rates observed in Fig. 2a–d. Indeed, both the off-rates showed similar behavior for various buffer conditions (Fig. 2c, d), indicating that the photoreactions are the same for the two different holoUnaG forms.

**Super-resolution imaging of various subcellular structures**. No fluorescence recovery without external BR of UnaG proteins in vitro and in fixed cells indicates that the repetitive binding of BR to the protein mainly causes the reversible photoswitching of UnaG (Fig. 2e and Supplementary Fig. 6). Since the binding kinetics of UnaG can be fully controlled by the light intensity and the concentration of BR and the reaction mechanisms of the off- and on-switching are independent to one another, we can control the $k_{on}$ and $k_{off}$ independently to tune the on–off duty cycle, which is a critical advantage for SML imaging[48]. The on–off duty cycle is defined as the fraction of fluorophores in the on state. Since a fluorophore with a duty cycle of $1/N$ allows less than $N$ molecules to be localized in a diffraction-limited area, a low duty cycle is preferred. The lower the duty cycle, the more fluorophores can be localized without causing artifact related to overlapped images. The duty cycle of UnaG can be estimated from the rates:

$$(\text{duty cycle}) = \frac{[\text{holoUnaG}]}{[\text{holoUnaG}] + [\text{apoUnaG}]} = \frac{1}{1 + k_{off}/k_{on}}, \quad (1)$$

where $k_{on} = k_{-1}[\text{BR}]$ and $k_{off} = k_1 I_{ex}$. For instance, when $I_{ex} = 300\ \text{W cm}^{-2}$ and $[\text{BR}] = 1\ \mu\text{M}$ under which we obtained localization dataset in Fig. 1d–i, Fig. 2b, d estimates $k_{off} \approx 37\ \text{s}^{-1}$ and $k_{on} \approx 0.13\ \text{s}^{-1}$, resulting in the duty cycle of 0.0035 from Eq. 1, which is on par with commercial dyes widely used for SML imaging including Atto 488[48]. The duty cycle can be further reduced by decreasing the BR concentration. Increasing the oxygen concentration is not preferred because the number of photons emitted in a switching cycle would be reduced as observed in Fig. 1h, thereby worsening the localization precision.

By using the controllable photo-switching properties of UnaG, we optimized the SML imaging conditions and then performed super-resolution imaging of UnaG-labeled subcellular structures by genetically incorporating UnaG to proteins of interest in fixed mammalian cells (Fig. 4). Before SML imaging, we could obtain conventional widefield images with a low excitation intensity. Then, at a high excitation intensity of ~300 W cm$^{-2}$ and an exogenous BR concentration of ~1 μM, we could shelve most of UnaG molecules in the view field to the non-fluorescent state and leave only a small fraction of the molecules stochastically bound with BR to fluoresce. The photon numbers per frame were exponentially increased as the exposure time increased and saturated at a sufficiently long exposure time (>150 ms), emitting ~1200 photons on average before off-switching, which is on par with EosFPs, the brightest FPs for SML imaging[7], and slightly less than Atto 488, the preferred commercial blue-absorbing synthetic dye for SML imaging (Fig. 4a–c)[48]. Since the localization uncertainty is inversely proportional to the square root of photon numbers, UnaG is equivalent to EosFPs and Atto 488 in terms of localization precision. From repetitive localization from surface-immobilized single UnaG molecules, we measured the localization precision as ~12 nm and the resolution in full-width at half-maximum (FWHM) as ~28 nm in lateral directions (Fig. 4d).

We labeled various subcellular structures such as the ER (UnaG-Sec61β), vimentin filaments (Vim-UnaG), mitochondrial matrix (Mito-UnaG), peroxisomes (PMP70-UnaG), clathrin-coated pits (UnaG-CLC, clathrin light chain) and lamin filaments (UnaG-LaminA/C) with UnaG and took SML images from each sample for 5000–100,000 frames at 100 Hz frame rate under Epi illumination (Fig. 4e, f and Supplementary Figs. 7–9). The expression of subcellular structures labeled with UnaG did not severely perturb the cellular morphology observed by a bright-field microscope and the cytoskeletal distributions were similar to those labeled with other FPs that are known to minimally perturb the structure (Supplementary Fig. 10)[7]. In various types of subcellular structures including membrane, fibril and coated vesicle, the morphologies appeared to be free from labeling artifacts while the on-/off-switching properties of UnaG were largely unchanged. The widefield and SML images of the three subcellular structures (Fig. 4e, f) clearly demonstrated the resolution enhancement of UnaG-based SML imaging. The average cross-sectional profiles of vimentin filaments gave FWHM of 57 nm, whereas the widefield measurements of the same fibers resulted in FWHM of 290 nm (Supplementary Fig. 7b). The mean localization uncertainty of each localization from the reconstruction image of vimentin filament was measured as ~25 nm (Supplementary Fig. 7c). The lengthwise labeling coverage along each thin vimentin filament was 76% in average (Supplementary Fig. 11), which is higher than that of mMaple3 (56%) and similar to the value obtained from dye-conjugated nanobody for BC2 tag, a small peptide tag[49]. The high labeling coverage of UnaG stems from the small size of the probe (about half of most FPs) as well as the near complete labeling of FPs (Supplementary Fig. 12).

Comparing to other FPs used in SML microscopy, UnaG uniquely offers both high photon number and reversible switching. EosFPs undergo irreversible photoconversion and photobleaching[7]. Reversibly photoswitching FPs offer low photon numbers[5,13,50]. Some organic dyes offer significantly more photons from reversible switching and can be genetically incorporated via self-labeling proteins such as SnapTag and HaloTag for SML imaging[51,52]. However, synthetic dyes must be exogenously supplied and are often impermeable to the cell membrane, and also increase the background signals due to the nonspecific bindings. Moreover, covalent bonding between dyes and SnapTag/HaloTag prevent replacement of photobleached fluorophores. Replenishable probes, such as DNA-PAINT probes, are always fluorescent and can cause severe background signal at nanomolar concentrations, and require specialized illumination

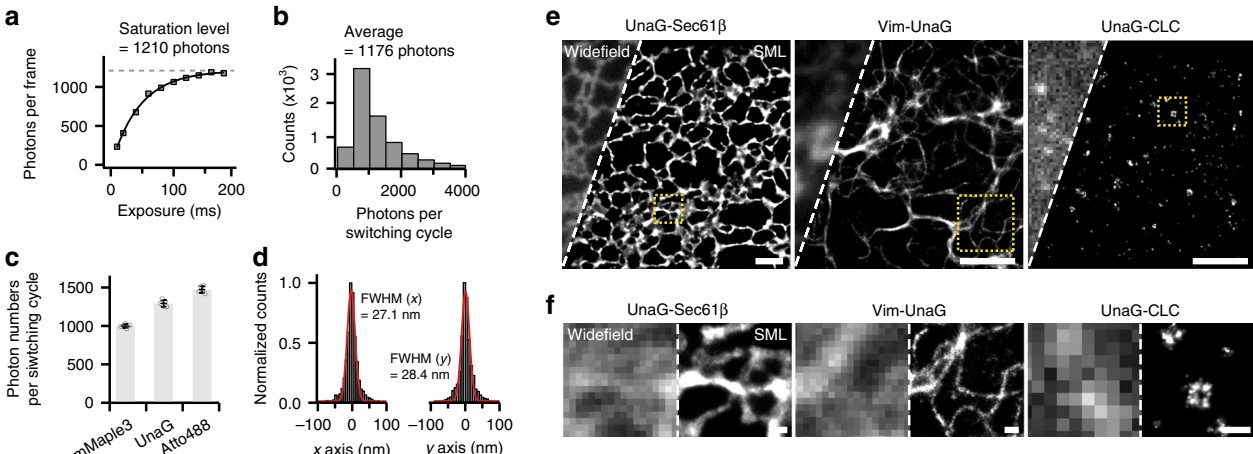

**Fig. 4 Applications to super-resolution microscopy of UnaG. a** Averaged photon numbers per frame from single holoUnaG as increasing the camera exposure time. **b** Distribution of photon counts per single switching event at 180 ms exposure time. **c** Comparison of the photon numbers per switching cycle for conventional super-resolution probes measured in an identical experimental setup, with 200 ms of camera exposure time. High illumination powers at 561 nm (mMaple3) or 488 nm (UnaG and Atto 488) were used so that most of the single fluorophores switched off in a single camera frame. As a result, UnaG gave more than 1200 photons, which stands between the mMaple3 and Atto 488. **d** Repetitive localization positions measured from surface-immobilized single UnaG molecules, projected in x axis and y axis in the lateral directions. Multiple localization distributions were aligned by their centroid positions obtained from Gaussian fits. The combined datasets were again fitted with Gaussian that yielded ~12 nm of localization precision and ~28 nm of full-width at half-maximum (FWHM) in lateral planes. The localization distributions were normalized by their own maximum counts. **e** Demonstration of SML imaging utilizing the photo-switchable nature of holoUnaG in various subcellular structures (left: ER, middle: vimentin filaments, right: clathrin-coated pits) in fixed Cos7 cells. **f** Close-up widefield and SML images from the yellow-boxed regions in **e**. Scale bars: 2 μm for **e**; 200 nm for **f**. Error bars: standard deviations ($n = 5$, each measurement contained more than 500 single-molecule information). CLC clathrin light chain.

geometry such as TIR-illumination (Supplementary Fig. 13). In contrast, UnaG offers the unique combination of following advantages simultaneously: (1) simple labeling method of standard genetic incorporation; (2) high photon numbers comparable to EosFP and Atto 488; (3) reversible switching between the dark and green fluorescence states; (4) fluorescence recovery by binding a natural metabolite (i.e. BR) that can be exogenously supplemented when necessary; (5) little fluorescent background under Epi illumination at micromolar ligand concentration that acts as pseudo-unlimited reservoir (Supplementary Fig. 14).

**Live-cell multicolor super-resolution imaging.** The genetically incorporated UnaG can be readily applied to live-cell SML imaging and offers fast tracking for the cellular dynamics due to the high photon numbers and easily controllable UV-free kinetics. We first investigated the possible cytotoxic effects of our imaging conditions to the live cells. As an endogenous metabolite, external BR did not induce any severe cytotoxic effects to the live cells up to 2 μM in general cell growth conditions (Supplementary Fig. 15a). However, intense light irradiation (488 nm, 300 W cm$^{-2}$) induced cell freezing with long observation time, which differed from the effect of weak 405-nm activation light (4.5 mW cm$^{-2}$) that caused cell apoptosis (Supplementary Fig. 15b–e)[53]. Interestingly, the phototoxic effect of 488-nm light was practically eliminated in the SML imaging buffer that contains both GLOX and BR (Supplementary Fig. 15f, g). Excess BR can efficiently absorb the blue light instead of cellular materials and GLOX can decrease the concentration of ROS by depleting the dissolved oxygen and the H$_2$O$_2$ (Supplementary Fig. 15h)[28]. As a result, the cells survived for more than 3 h after they were illuminated by 300 W cm$^{-2}$ of 488-nm laser for up to 15 min. PAINT with no chemical additives or primed conversion with near-infrared activation may offer more cell-friendly imaging conditions. But applications of

PAINT to live cells are limited by impermeability and chemical toxicity of the probes and primed conversion FPs convert the emission from green to red upon exposure to blue and red/infrared lasers, making simultaneous multicolor imaging near impossible[54,55].

In Fig. 5, we obtained live-cell SML images of UnaG-Sec61β on the cytosolic side of the ER membrane. The imaging buffer was supplemented with the oxygen scavenging system based on GLOX for slowing down the off-switching rate and with exogenous BR at the final concentration of 300 nM for accelerating the on-switching rate. The imaging medium with GLOX was previously used in live-cell STORM imaging[56]. As like in the case of fixed cells, GLOX and external BRs also could control the switching kinetics of UnaG expressed in a live cell (Supplementary Fig. 16a, b). Under these SML imaging conditions, UnaG could be repetitively switched on and off with slight decrement of maximum fluorescence probably due to the insufficient recovery time for the spontaneous transition to the brighter state (Supplementary Fig. 16c)[40]. The off-switching rate was constantly maintained during the switching events, whereas the on-switching rate was continuously decreased probably due to the local depletion of BR molecules in solution by direct photo-oxidation (Supplementary Fig. 16d, e). Finally, we obtained a continuous time series consisting of ten super-resolution snapshots, each took 1 s for reconstructing the super-resolution image (Fig. 5b and Supplementary Movie 1). The movie resolves dynamic remodeling of the ER tubule network. Repetitive switching of UnaG lasted more than 15 min continuously providing 1-s SML snapshots (Supplementary Fig. 17a). The number of localized molecules decreased initially, but it settled to an almost stationary state with roughly half of the initial number after the switching reaction reached the equilibrium (Supplementary Fig. 17b). UnaG also successfully resolved the ring-like projections of clathrin-coated pits in live cells with 2-s temporal resolution by increasing the excitation intensity tenfold higher ($I_{ex} \simeq 3$ kW cm$^{-2}$, Supplementary Fig. 18). Without supplementing exogenous BR, the live-cell images of clathrin-coated pits became blurred by the motion and

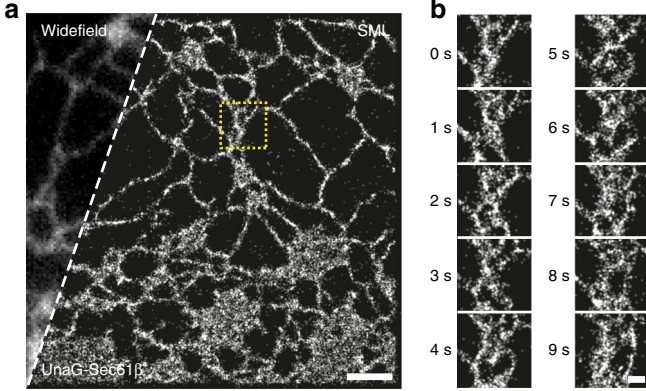

**Fig. 5 Live-cell super-resolution microscopy. a**, **b** One-second-long SML images of the ER in a live Cos7 cell transfected with UnaG-Sec61β. **a** A far view with conventional widefield and SML images. The widefield image was taken immediately before super-resolution imaging. **b** A time series of closed-up SML images from the yellow-boxed region in **a**. Scale bars: 2 μm for **a**; 500 nm for **b**.

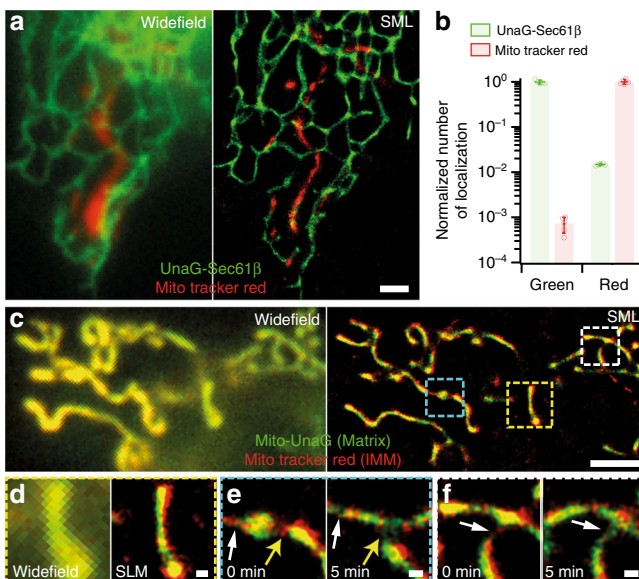

**Fig. 6 Two-color live-cell super-resolution microscopy. a** ER (green) and mitochondria (red) of a live Cos7 cell in two-color widefield (left) and SML (right) images. The cell was transfected with UnaG-Sec61β and subsequently stained with MitoTracker Red. The widefield image was taken immediately before super-resolution imaging. **b** Crosstalk analysis using singly labeled control samples imaged in the two emission channels of green and red split by a dichroic mirror and filtered by a single bandpass filter for each channel. The localizations numbers were normalized by the average count from the proper color channel. **c-f** Mitochondrial matrix (green) and inner membrane (red) in two-color SML images of a live Cos7 cell transfected with Mito-UnaG and stained by MitoTracker Red. **c** A far view comparison between widefield (left) and SML (right) images. **d** Close-up comparison between widefield (left) and SML (right) images of yellow-boxed region in **c**. **e**, **f** Zoom-ins in the cyan- and white-boxed regions in **c** at 0 min (left) and after 5 min (right) with fusion (white arrows) and fission (yellow arrows) events. Scale bars: 2 μm for **a**, **c**; 200 nm for **d**, **f**. Error bars: standard deviations (n = 5, from independent SML images).

structural evolution during the substantially longer time for data acquisition (Supplementary Fig. 19).

In addition, UnaG's narrow spectral window in green streamlines multicolor live-cell SML imaging. Since UnaG only emits green fluorescence, orange-to-red window is available for other live-cell compatible photo-switching probes. Photoconvertible FPs occupying green and red channels cannot be combined with UnaG[5,6] and only allow for far-red dyes to be simultaneously imaged[57]. FPs that switch between the dark and fluorescent state such as YFP and PAmCherry can be combined with UnaG (Supplementary Fig. 20b), but the low number of photons and activation with UV light is less preferred for live-cell SML imaging[6,58]. Instead, lipophilic cell-permeable dyes that reversibly switch between red and dark states and produce >2000 photons can be readily combined with UnaG for high-quality SML imaging[56].

With MitoTracker Red, a cationic rosamine dye, we stained the mitochondrial inner membrane of live cells transfected with UnaG-Sec61β. By separating the two fluorescence images with a dichroic mirror and filtering the two images separately with single-band bandpass filters, we simultaneously obtained two-color super-resolution images of the ER and mitochondria in a live cell with virtually no color crosstalk between the channels (Fig. 6a). The quantified color crosstalk was 1.45 ± 0.09% and 0.07 ± 0.02% from green to red and vice versa, respectively (Fig. 6b, ± values are the standard deviations from five independent measurements of >10,000 molecules per experiment). With the crosstalk-free methods, we imaged the mitochondrial matrix and inner membrane using mito-UnaG and MitoTracker Red, respectively (Fig. 6c–f). The matrix and inner membrane appear completely colocalized in conventional widefield images (Fig. 6c, d). The two bright probes with little crosstalk enabled us to resolve the outline of the membrane enclosing the matrix in the super-resolution images (Fig. 6d). In addition, thin membrane tubes lacking matrix were observed during fission/fusion events (arrows in Fig. 6e, f).

The same multicolor super-resolution imaging condition for live cells can be expanded to many different targets and probes. For instance, other photoswitching red membrane probes specific to the ER, lysosome and the plasma membrane[56] can be combined with UnaG fused to various proteins of interest, generating a large number of two-color combinations (Supplementary Fig. 20a–c). Moreover, far-red probes can be added to the two-color method for three-color live-cell super-resolution

imaging. Far-red photo-switching dyes can be targeted through self-labeling enzymes such as SnapTag and HaloTag[51,52]. In Supplementary Fig. 20d, we demonstrated the three-color imaging of UnaG-Sec61β with MitoTracker Red and Alexa Flour 647 labeled on the cellular membrane by using HaloTag. Three-color imaging did suffer from bleaching problem, but it was possible to obtain three-color image for highly expressed targets such as the plasma membrane. Since our three-color live-cell SML method combines spectrally distinct probes, it does not require advanced instrument and analysis for ratiometric distinction[14] or spectral detection[59], but can use a standard filter-based multicolor instrument that can optically minimize spectral leak through.

## Discussion

We discovered the fluorescence recovery from photobleached UnaG due to the breakage of ligand–protein interactions and subsequent displacement of photo-oxidized BR from UnaG, followed by rebinding to fresh BR in solution. The photo-oxidation process was extensively characterized, including the kinetics of both on- and off-switching directions and the reaction mechanism of the off-switching process. From the detailed understanding of the reactions, we devised ways to control the reaction rates in both directions individually and conveniently. We performed SML imaging of various subcellular structures at sub-diffraction-limited resolutions, utilizing the advantages of UnaG such as

genetic incorporation for labeling, background-free imaging under Epi illumination, controllable UV-free switching kinetics ensuring both of high contrast ratio and fast acquisition, and crosstalk-free multicolor imaging of live cells.

The switching chemistry of UnaG is distinct from structural changes occurring in the chromophores in conventional photo-convertible proteins[5,6]. In all FPs except for UnaG, the chromophores are covalently linked to the protein (Supplementary Fig. 1). In GFP and photoswitching proteins used in SML imaging, the chromophores are formed from a series of reactions between the amino acids of the proteins (Supplementary Fig. 1a, d, e). In infrared FPs, the BV ligand is covalently linked through a thioether bond to a Cys residue in the protein (Supplementary Fig. 1b)[60]. When we performed a similar experiment on IFP2.0, we could not observe switching behavior (Supplementary Fig. 21), indicating that the covalent bond between BV and IFP2.0 prevents the bleached BV from dissociation. In all the other photoactivatable and photoconvertible proteins, the chromophores undergo photo-induced reactions such as isomerization, covalent bond formation/breakage and protonation/deprotonation to switch the fluorescence on and off (Supplementary Fig. 1d, e)[5,13]. In this study, we found that the non-covalent binding interaction of the ligand and the protein allows the detachment of damaged chromophore and rebinding to a different chromophore molecule.

UnaG photo-switching chemistry is also different from all switching fluorophores including synthetic dyes[8]. The off-switching photo-oxidation reaction of UnaG is reminiscent to the photobleaching of photoactivatable proteins in PALM and light-induced adduct formation of cyanine dyes in STORM[1–3]. The on-switching reaction of UnaG related to the BR binding is similar to that of PAINT or BALM[4,61]. Unlike PAINT requiring selective, sheeted illuminations, the fluorogenic nature of BR allows for standard Epi illumination as in STORM. Nonetheless, the stochastic switching of individual UnaG molecules allows temporal separation of the single-molecule images and high-precision localization of the centroid positions, as in all the SML techniques.

UnaG may enable molecule-level colocalization analysis, thanks to the specificity and completeness of labeling a POI and the proximity to the target (<3 nm). By expressing UnaG directly attached to another spectrally distinct FP in mammalian cells, and then pulling the molecules using antibodies on glass coverslips for SML, we obtained the colocalization signals with ~13 nm center-to-center distance in average (Supplementary Fig. 22c–e). This colocalization assay can be extended to molecules in cells as we demonstrated for N- and C-termini of CLCs (Supplementary Fig. 22f). However, in the current experimental setting, we could not measure distances within individual CLC molecules due to the ~8-nm error governed by localization and alignment accuracies (Supplementary Fig. 22a, b). To apply UnaG for true molecular colocalization analysis, other technological advances should follow. First, the localization precision of SML should reach < 5 nm. MINFLUX with 1-nm resolution has high promise once multicolor MINFLUX becomes available[62]. Second, more precise method for aligning the two images for each channel is required for reducing errors in the distance estimation between two molecular species[63]. Third, UnaG has to be attached to all the target molecules in a cell for eliminating false negatives in colocalization from incomplete labeling. UnaG's small size (15.6 kDa) may facilitate genome editing to knock in the probe to the gene of interest while keeping the target-probe distance short[64]. Fourth, colocalization analysis algorithm should take into account the nanoscale resolution, target–probe distance and orientation, and labeling coverage[65]. These combined efforts will enable us to directly visualize molecular interactions in living cells.

## Methods

**Plasmids.** Genes were cloned into the specified vectors by using standard enzymatic restriction digest and ligation with T4 DNA ligase. To generate constructs where short tags (e.g. HA or Flag epitope tag) or signal sequences were appended to the protein, the tag was included in the primers used to PCR-amplify the gene. PCR products were digested with restriction enzymes and ligated into cut vectors (e.g. pcDNA3 and pDisplay). In all cases, the CMV promoter was used for expression in mammalian cells. The genetic constructs cloned and used for this study as well as the sequence information for primers are summarized in Supplementary Table 1.

**Purification of holoUnaG.** UnaG cloned into pGEX-2T was kindly provided by Dr. Miyawaki (RIKEN, Japan)[23]. To produce UnaG protein, the plasmid was transformed into *Escherichia coli* strain BL21 (DE3) cells. Proteins were expressed by induction of 0.25 mM isopropyl β-D-1-thiogalactopyranoside at 18 °C for 18 h. The cells were harvested and lysed in 25 mM sodium phosphate pH 7.8, 400 mM sodium chloride, 5 mM dithiothreitol, 1 mM phenylmethylsulfonyl fluoride at 4 °C. UnaG was purified using GST affinity chromatography, and the GST tag was removed with thrombin protease over pH 8.0 overnight at 4 °C. The protein was purified further by ion exchange (HiTrap Q HP, GE Healthcare) and size-exclusion chromatography on a Superdex 200 column (GE Healthcare). Reconstitution was carried by mixing purified ApoUnaG and BR with 1:2 stoichiometry. After reconstitution using BR, free BR and DMSO were eliminated using a HiTrap desalting column (GE healthcare) equilibrated with a PBS buffer.

**Absorption and fluorescence measurement.** The absorption and fluorescence spectra were obtained by using a UV/VIS absorption spectrometer (Lambda25, Perkin Elmer) and a fluorescence spectrometer (Quantamaster, Photon Technology International). To bleach the holoUnaG, a sample cuvette was placed in front of a blue laser (TECBL-20GC-488, World Star Tech), whereas the cuvette was placed in a dark room after adding BR during fluorescence recovery.

Pictures for fluorescence emission of purified UnaG showed in Fig. 1c were taken with a camera on cellular phone (V20, LG). An emission filter (HQ525/50m, Chroma) was attached in front of the camera to transmit only the fluorescence.

**Purification of photo-oxidation products.** The photo-oxidation products were extracted by using different solubility of the compounds between aqueous and organic solvents. We added equivalent volume of chloroform (528730, Sigma Aldrich) into bleached holoUnaG, and then strongly vortexed the mixed sample. The UnaG proteins were denatured and aggregated as a white pellet under this condition, where the organic layer was gradually turned into yellow color indicating that the UV-absorbing products were dissolved in the organic layer. We gently took the organic layer and evaporated the solvent at 60 °C with vacuum pump. The resulting yellow pellet was primarily dissolved in dimethyl sulfoxide (D8418, Sigma Aldrich), and further diluted for five times with deionized water before analysis.

**LC–MS.** LC separation was performed by a Nexera X2 ultra high performance liquid chromatography (UHPLC) system (Shimadzu Scientific Instruments) with an ACE Excel 2 C18 column (50 × 2.1 mm i.d., 2 µm, ACE). The mobile phases were (A) 0.1% formic acid in 95/5 water/acetonitrile and (B) 0.1% formic acid in acetonitrile. The solvent program (gradient) consisted of holding solvent (A/B 90:10) for 1 min, the linearly converting to solvent (A/B 60:40) for 7 min, the linearly converting to solvent (A/B 15:85) for 2 min and holding for 10 min (Supplementary Fig. 4a). The flow rate was 200 µL min$^{-1}$ and the injection volume was 10 µL. The column temperature was set to 40 °C. For LC-UV/vis detection, the LC system was coupled to a photodiode array detector (DAD, SPD-M30A, Shimadzu Scientific Instruments). For LC/MS analysis, the LC system was interfaced to a Thermo Finnigan LTQ XL linear ion trap mass spectrometer (Thermo Scientific Inc.) via Ion Max electrospray ionization (ESI) source. For ESI, the voltage was set to +4.0 kV and capillary temperature was 250 °C. Full mass spectra were recorded with the mass range of 100–2000 Da. For HRMS and HRMS/MS analyses, a Q Exactive orbitrap mass spectrometer (Thermo Scientific Inc.) was utilized. For collision-induced dissociation (CID) HRMS/MS, precursor isolation width and normalized collision energy were set to 2 Da and 35%, respectively. The resolutions of full-MS scan and data-dependent MS/MS (ddMS$^2$) modes for HRMS and HRMS/MS were set to 70,000 and 17,500, respectively.

**Cell culture, transfection, fixation and staining.** Cos7 cells (Korean Collection for Type Cultures (KCTC)) were cultured on a coverslip-bottomed 8-well chamber (155409, LAB-TEK) in Dulbecco's modified Eagle's medium (DMEM, SH30022.01, Hyclone) supplemented with 10% v/v fetal bovine serum (97068-085, VWR Life Science) and 1% v/v antibiotic–antimycotic (15240-062, Gibco). The cells were transfected at ~70% of confluency by using an electroporator (MPK5000, Invitrogen) with ~500 ng mL$^{-1}$ of plasmids. Mediums provided from the company were used during transfection. After 12–24 h post-transfection, we fixed the cells with 3% paraformaldehyde (50-980-495, Electron Microscopy Sciences) with 0.2% glutaldehyde (16020, Electron Microscopy Sciences) for 10 min at room temperature for all target structures except the ER, which was incubated in the fixation

solution for 30 min at 37 °C to preserve the delicate structure. After fixation, we washed the sample with PBS buffer rigorously.

For the live-cell multicolor imaging, MitoTracker Red (1 μM, M7512, Invitrogen), ER Tracker Red (10 μM, E34250, Invitrogen) and LysoTracker Red (1 μM, L7528, Invitrogen) were applied to the live cells for 1 min at 37 °C, and the cells were washed three times with DMEM before imaging. HaloTag staining with homemade Alexa Flour 647 labeled HaloTag ligand (Promega) was carried by incubating the live cells in 3 μM of HaloTag ligand containing growth medium for 15 min at 37 °C.

For the immunostaning of the vimentin filaments, primary (anti-vimentin, MA5-11883, Thermo Fisher Scientific) and secondary antibodies (Alexa Flour 488 labeled anti-mouseIgG, A-10680, Thermo Fisher Scientific; homemade Atto 488 labeled anti-mouseIgG) were diluted to 2 μg mL$^{-1}$ concentration in PBS, and applied to the fixed cells for 1 h at room temperature sequentially. Atto 488-NHS ester (41698, Sigma Aldrich) was linked to the bare anti-mouse IgG via click reaction with 3:1 stoichiometry, and purified by using a size exclusion column (Amicon Ultra, 10 k, UFC5010, Sigma Aldrich) to exclude the unreacted reagents.

To investigate the cellular toxicity of UnaG under SML imaging conditions, the cell viability was measured in two different ways: (1) a fluorescent live/dead assay (L3224, Thermo Fisher Scientific), (2) bright-field imaging with high magnification for cellular dynamics and morphology. Additionally, a fluorescent ROS indicator (CellROX Deep Red, C10422, Thermo Fisher Scientific) was used to check any possible ROS damage during SML imaging, by staining the cells with 5 μM of CellROX Deep Red for 30 min at 37 °C.

**Pull-down assay.** For pull-down assay, Cos7 cells were transfected with proper plasmids (UnaG-APtag + BirA, UnaG-mCherry, pSNAP-FtnA and UnaG-FtnA) at ~500 ng mL$^{-1}$ concentration and plated on the cell culture dish with 150 mm diameter. The cells grew up for 2 days in a biotin-containing medium to produce biotinylated UnaG. At >70% confluency, cells were washed with cold PBS and collected by using TrypLE (12605, Thermo Scientific Inc.). Collected cells were divided into 6 aliquots and centrifuged at 3000 g for 5 min at 4 °C. After discarding the supernatants, the resulting cell pallets were stored at −20 °C for further analysis.

Lysis of the cells was carried out in a lysis buffer (50 mM HEPES pH 7.4, 150 mM NaCl, 1 mM EDTA, 10% v/v glycerol and 1% v/v Triton X-100) supplemented with 1% v/v protease inhibitor cocktail (P8340, Sigma-Aldrich) and 1% v/v phosphatase inhibitor cocktail (P5726, Sigma-Aldrich). Cells were gently suspended and incubated at 4 °C for 30 min, with additional mixing steps for every 10 min. After incubation, the cell suspensions were centrifuged at 15,000 g for 10 min at 4 °C, and resulting supernatants were further filtered in a size-exclusion column (Amicon Ultra, 10 k, UFC5010, Sigma Aldrich).

To attach the cell extracts on the coverslip, we built a flow chamber with PEG-coated coverslips and double-holed slide glasses. The coverslips and slide glasses were sonicated in alconox, deionized water, acetone and 1 M KOH sequentially for 20 min each. Cleaned glasses were incubated in aminosilane reaction buffer (200 mL of MeOH + 10 mL of acetic acid + 2 mL of aminosilane) for 20 min, with 1-min sonication in the middle of incubation. PEG coating was carried by incubating the glasses in PEG reaction solution (320 μL of 100 mM sodium bicarbonate + 80 mg of mPEG + 1 mg of biotin-PEG) for 3 h. Prepared glasses were sandwiched by double-sided tape and sealed by epoxy. After building a flow chamber, 1.25 mg mL$^{-1}$ of streptavidin and the 1/100 diluted cell lysates were incubated for 10 min sequentially, with rigorous washing steps with PBS. Biotinylated UnaG from UnaG-APtag was attached to the surface via biotin–avidin interaction for 20 min in lysis buffer. For UnaG-mCherry, additional 1/500 diluted biotinylated anti-RFP antibody (ab34771, Abcam) was incubated for 1 h in lysis buffer, following UnaG-mCherry incubation for 20 min in lysis buffer.

FtnA clusters were immobilized by applying 1/100 diluted cell lysate on to the poly-L-lysine coated coverslip for 30 min. For SnapTag-FtnA, the immobilized lysate was blocked with 10% BSA in PBS for 30 min and treated with 0.1 μM of SNAP-Surface Alexa Flour 647 (S9136S, New England Biolabs) in staining buffer (10% BSA, 0.1 v/v Triton X-100 in PBS) for 6 h at room temperature.

**Imaging buffers.** Fixed cells were imaged in imaging buffer containing Tris pH 8 (10 mM, TRI01, LPS Solution), NaCl (50 mM, 7548–4405, Daejung), glucose (10% w/v, G7021, Sigma Aldrich), GLOX (560 μg mL$^{-1}$, G2133, Sigma Aldrich), CAT (400 μg mL$^{-1}$, C100, Sigma Aldrich) and BR (600–1,000 nM, B4126, Sigma Aldrich). To prevent the inflow of oxygen from the air, the top of the sample chamber was tightly sealed with coverslip using biologically non-toxic grease during measurement. For the live-cell imaging, the Cos7 cells were prepared on a coverslip (18 mm diameter, 0117580, Deckgläser) in the same condition as described above, and were imaged after 12–24 h post-transfection. Prepared coverslip was primarily mounted on a magnetic chamber (Chamlide CMB, Live Cell Instrument), which acts as an efficient reservoir of the buffer solution. A DMEM-based imaging buffer was introduced to keep the cells alive for a long time, which contained DMEM (no phenol red, high glucose, 21063-029, Gibco), HEPES pH 8 (75 mM, H4034, Sigma Aldrich), glucose (2% w/v), GLOX (560 μg mL$^{-1}$), CAT (400 μg mL$^{-1}$) and BR (300–1000 nM). To switch off Alexa Flour 647 for the multicolor imaging, 10 mM of β-mercaptoethylamine was supplemented to the buffer additionally.

**Fluorescence microscope and data analysis.** A home-built imaging setup with standard Epi illumination was used to acquire all imaging results, except for confocal z stack images in Supplementary Fig. 10b that were obtained by using a commercial spinning disk confocal system (DragonFly, Andor). Three laser diode light sources (488 nm: 150 mW, OBIS, Coherent; 561 nm: 150 mW, OBIS, Coherent; 647 nm: 120 mW, OBIS, Coherent) were co-aligned and coupled with a single-mode optical fiber (3.4 μm diameter, PM-488-FC, Thorlabs) for beam cleanup. The fiber output was 25 times expanded with relay lenses to enlarge the field-of-view, and delivered into a microscope body (Eclipse Ti-E, Nikon) through a lens (f = 400 mm, AC508-200-A-ML, Thorlabs) to focus the laser at the center of the back aperture of objective lens (Plan Apo TIRF, 100×, NA1.49, oil, Nikon) to achieve Epi illumination. A dichroic beam-splitter (ZT405/488/561/647rpc, Chroma) was introduced to deliver the laser lights to objective lens, where the fluorescence passed through the dichroic beam-splitter and was further filtered by an emission filter (ZET405/488/561/647m-TRF, Chroma). A perfect focus system kept the imaging plane constant during all measurements.

For single-color imaging, filtered fluorescence was directly recorded with a scientific CMOS camera (sCMOS, Prime95b, Photometrics) without any other optics except the tube lens. The camera was set to 12-bit sensitive detection mode without any pre- and post-processing, with 110 nm of pixel size. In order to obtain exact photon numbers per frame of holoUnaG, the effective photoelectrons per A/D count (gain, G) of camera was measured by using the relation, G = counts/variance(counts), giving a value of 0.44 comparable to manufacturer's information (G = 0.65). For multicolor imaging, the filtered fluorescence was delivered to a multichannel imaging system (QV2, Photometrics) to project the single-molecule signals from different dyes to different regions of the EMCCD camera (iXon Ultra 888, Andor) and record the multiple images at the same time. We used a narrow band-pass filter for the red dyes (ET605/30m, Chroma) to minimize the fluorescence leakage of UnaG, and moderate band-pass filters for green (ET525/50m, Chroma) and far-red (ET700/75m, Chroma) dyes. The EMCCD camera recorded all the channels simultaneously with 130 nm of pixel size, 30 MHz of pixel readout speed and 30–300 of EM gain.

To image the live samples, temperature of the sample holder and the objective lens were kept at 37 °C by using a temperature controller (Chamlide TC, Live Cell Instrument), if needed.

An open source imaging software, μManager[66], was used to acquire widefield and raw single-molecule images. Another freely distributed software, ThunderSTORM[67], analyzed the raw data and reconstructed the SML images. For the live-cell SML imaging with 1-s time resolution, the raw images were recorded in overactivated conditions and processed with HAWK software that enables artifact-free high-density localizations[68], before analyzed with ThunderSTORM. The SML images consisted of 5- or 10-nm pixels, and rendered by normalized Gaussian. The multicolor images recorded in different regions of the camera were aligned using calibration images of broadband fluorescent beads (TetraSpek, T14792, Invitrogen) that appeared in both channels and a home-built MatLab code that transformed the signals from red and far-red regions to corresponding green region by using second-order polynomial function.

Labeling coverage of vimentin filament was analyzed by the following steps. First, we manually traced thin, individual filaments and straightened the segmented images using ImageJ. Then, a custom MatLab code converted a straightened image into a binary image and calculated the fraction of covered area of the middle three pixels.

To analyze the completeness of labeling from FtnA cluster images, we first analyzed the photon numbers from single UnaG and Alexa Fluor 647 molecules under switching condition. Then, the fluorescence intensities from >100 FtnA clusters labeled with each fluorophore were collected at the same illumination intensity, camera settings including the exposure time and buffer conditions. Since FtnA is non-existent in mammalian cells and forms homo-oligomer consisting of 24 monomers[69], 24-times of the single-molecule intensity was used as the theoretical estimation value form the FtnA cluster, and the completeness of labeling was calculated by the percentage of measured cluster intensity to the theoretical estimation.

**Reporting summary.** Further information on research design is available in the Nature Research Reporting Summary linked to this article.

## Data availability

The generated and analyzed datasets that support the findings of this study including the raw data are available from the corresponding authors upon reasonable request.

## Code availability

The home-built data analysis software used in this study are available from corresponding authors upon reasonable request.

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

## Acknowledgements
We thank Prof. Atsushi Miyawaki (RIKEN, Japan) for the UnaG plasmid for purification and Prof. Seong Keun Kim (Seoul National University) for generously allowing us to use his facilities. We are also grateful to Myeong-Gyun Kang and Chulhwan Kwak for their intense assistance during the revision period. This work was supported by the Institute for Basic Science (IBS) program IBS-R023-D1 to J.K., M.K. and S.-H.S., and K.T.K. was supported by another IBS program (IBS-R020-D1). S.Choi and S.Cha were supported by Basic Science Research Program through the National Research Foundation of Korea (NRF) funded by the Ministry of Education (NRF-2016R1D1A1B01006576). J.-S.P. and H.-W.R. thank general support by Korea Brain Research Institute (KBRI) basic research program. H.-W.R. was supported by the Organelle Network Research Center (NRF-2017R1A5A1015366) and the NRF (NRF-2019R1A2C3008463).

## Author contributions
S.-H.S and H.-W.L. designed and supervised the research; J.K. measured and analyzed kinetics with simulation; J.-S.P. synthesized the gene constructs; M.K. prepared fixed and live cells transfected with UnaG; J.K., S.Choi and S.Cha acquired LC-HRMS chromatograms; J.K. and G.T.K. acquired SML images; J.-S.P., J.P. and C.L. purified UnaG proteins; J.K. and S.-H.S. mainly wrote the manuscript with feedback from other authors.

## Competing interests
The authors declare no competing interests.
