## [Peer Review File · Nature Communications]

Reviewers' comments:

Reviewer #1 (Remarks to the Author):

The authors detail the application of UnaG as a versatile super-resolution and single molecular tracking dye in the submitted manuscript. To this end, the authors suggest a unique mode of fluorescent switching compared to previously established dyes, based on the photo-oxidation of the UnaG ligand, bilirubin, to turn off the fluorescent signal without irreversibly damaging the UnaG protein. Based on the outlined mechanism, the authors demonstrate the versatility of UnaG in multi-colour super-resolution imaging of live cells. Overall the concept and practical application of UnaG as a super-resolution dye represent a novel approach.

Major criticisms:

1. The general mechanism of bilirubin photooxidation and the involvement of a singlet oxygen species were extensively discussed in literature (e.g. McDonagh, *Biochem. Biophys. Res. Commun.*). Nevertheless, while the singlet oxygen driven mechanism is the primary accepted mechanism, the MS and MS/MS evidence provided by the authors for the formation of the suggested products 1 and 2 is weak. The points of concern are the combination of :
 - a) low resolution MS data of the suggested oxidation product.
 - b) lack of peak peaking, especially in case of Fig. S3e and to some degree Fig S3d.
 - c) precursor isolation width of 3 Da, very low accuracy for precursor mass selection.
 - d) while carboxylic acid containing ions tend to exhibit low precursor ion intensity in MS/MS scans, it is not clear whether the complete absence of the precursor ions in the MS/MS spectra in Fig. 3C is due to complete fragmentation or due to the selected observation range of the second mass scan (not specified in the methods section).
 - e) no reference spectra are provided or cited (possibly due to unavailability).While improved mass spectral data, such as high resolution MS, or MS/MS or reference material would be highly recommended, alternatively the discussion on page 6 should more clearly emphasize the highly speculative nature of the proposed oxidation products.
2. Fragments in fig 3c should be denoted following standard MS/MS procedures, such as specification of $[M-H_2O]^+$ etc and all fragmentation points should be clearly marked in a single structure of the precursor ion. This will circumvent ambiguity or potentially misleading fragment structures, such as the proposed dehydration of the precursor ion at the carboxylic acid function giving rise to the 297.1 fragment. In general, a protonated carboxylic acid should be more likely associated with the formation of an acylium ion due to its resonance stabilization and not an enol (*Middlemiss Can. J. Chem.* 57, 1979), if the authors suggest that the carboxyl function is protonated under the ionization conditions.
3. The depicted Br oxidation in Fig. S4 by singlet oxygen to give rise to the two proposed oxidation products are in agreement with previous reports and supported by the demonstrated oxygen dependency of the observed oxidation. Unfortunately, the suggested localization of the oxidation, inside UnaG, is highly speculative and no evidence has been provided to support this notion. Based on previously published X-ray data, presence of oxygen inside the binding pocket of UnaG-Br has not been reported. Additionally, formation of a Br radical at the bridging methylene carbon would highly likely be able to form a novel covalent bond between UnaG and its ligand, rendering the protein unable to rebind a non-oxidized Br ligand effectively constituting photobleaching. To support the proposed oxidation location, the authors need to provide evidence that singlet oxygen is inside the UnaG binding pocket and explain why carbon radical formation does not lead to Br-UnaG covalent bond formation and photobleaching.
4. As the existence of damUnaG1/2 is highly unlikely and not supported by experimental evidence, the authors will need to either adopt their proposed model by excluding damUnaGs or as outlined in point 3 above, demonstrate the existence of damUnaGs. At this point it seems likely, that omission of damUnaG species will lead to a more simplified model, with equally good fitting properties.

Minor comments:

5. Please specify the amount of UnaG used for the experiment of Figure 2.
6. page 10, line 267. The data of IFP2.0 should be presented.
7. I wonder where the values of k_{on1} , k_h and k_h' shown in Figure S2a come from.

Reviewer #2 (Remarks to the Author):

The manuscript "Bright Ligand-activatable Fluorescent Protein for High-quality Multicolor Live-cell Super-resolution Microscopy" introduces and characterizes the UnaG/bilirubin system as a fluorescent labeling system for localization microscopy. The labeling system is rather small (about half the size of GFP) and can be "refueled" by the pool of available bilirubin – both are good reasons to check and evaluate it as an interesting and promising SMLM labeling option. I nevertheless believe that the system is not convincingly characterized in the current manuscript and doesn't prove the claimed SMLM quality. Therefore, I would like to ask the authors for several additional experiments and clarifications which could appease my concerns as explained below:

1) Please state for all images in the figure caption as well as in the text if the images were taken on live or fixed cells. This is an important detail.

2) Data of Figure 1 and 2: I do believe the linear dependencies on BR concentration and light intensity as shown in Figure 2. I nevertheless don't believe that they explain the statistics shown in Figure 1 and the behavior of the probe for SMLM conditions. Figure 2 was measured in the 0-80 mW/cm² 488 nm irradiation regime (actually, what were the intensities used for 2c and e – I believe e) is the same blue curve as in a) thus 80 mW/cm²?), Figure 1 is measured with 300 W/cm², an increase in laser dose of 3750-fold and in a changed buffer (with/without oxygen). I strongly believe that at these really high intensities used for STORM-like imaging, different mechanisms and transitions occur as compared to the low laser intensity regime and that these different switching characteristics will not show a simple linear dependence extending from the graphs shown in Fig.2. This can also be simply estimated from the authors data as the product of +BR-OS photon count peaking about ~130 and locs per 1000 frames with 300 is about 39.000 and photon peak of +BR+OS with 190 times ~ 500 locs/1000 fr. is 95.000  the ratio of this is only a 2x increase in total number of detected photons not matching the prediction using the rates of figure 2.

Thus, please i) measure the rates and kinetics at relevant settings (thus high intensities and corresponding buffers, e.g. prove the predicted 13 1/s off-rate!). ii) Please proof that the off-switching under STORM conditions is pure photo-oxidation and no additional transitions like e.g. some fraction in a reversible dark state making them blinking in a STORM-like manner is involved (hint already measured in Ref 43). iii) please measure the dissociation rate of the photo-oxidized product?

3) Phototoxicity: The authors mention several times that UV light is phototoxic (introduction, live-cell results, discussion). With this I generally agree but I highly believe that 300 W/cm² of 488 nm light as used by the authors is actually much more phototoxic than the usual mW/cm² regime of 405 nm light used for photoactivation of fluorescent proteins. Please proof that e.g. imaging the biological targets of Fig. 4 by the UnaG system is causing less cell death than imaging the same targets (at similar SR image quality) by e.g. mEos3.2 fusions (mEos2 is oligomerising at higher densities and thus not very useful to image dense structures as e.g. vimentin (see publ. Zhuang group, Wang et al PNAS).

4) Are biological targets disturbed by UnaG labels and BR? E.g. steric hinderance, oligomerization artefacts of UnaG, possible ROS damage by the BR radicals in photo-oxidation processes, possible metabolic reactions on oxygen depletion, effects of non-physiological concentrations of ~ 1 μ M BV (for the latter: 1 μ M BV control data is not shown and 2h incubation is not long enough for visual cell shape read-out), please characterize the BR uptake for living cells etc.? Authors need to control ALL of this, measure and characterize ALL the effects and to do comparative experiments with common labels to show that the UnaG system yields better compromises than other labeling solutions.

5) Please do comparative photon counts for Eos and Atto488 on your system, these heavily depend on exact setup parameters.

6) FIRE values are highly doubtful. I don't believe the stated 18 nm. FIRE values have to be corrected for self-blinking effects, e.g. caused by highly blinking dyes or by several localizations on adjacent frames – please revisit original publication (“Q value”). Also, please do calibration experiments on standards, e.g. single-molecule surfaces with UnaG yielding FWHM of single protein localization distributions or use other common standards (NPC, oligomers, origamis etc.). Compare those to other labels!

In this context, also Figure 5 and 6 don't look convincing. My guesstimate by eye would rather be ~100 nm resolution, which, on these targets can be reached by deconvolution. In this sense, SMLM fitting here acts as a background removal tool and thus results in contrast improvement and not so much in a resolution improvement.

7) Live-cell imaging: How does GLOX on the outside of cells help the labels in the inside? Ref. 50 imaged proteins ON the membrane. How do ON-rates change for live cells due to the needed uptake of BR (intracellular concentration is not sufficient!).

8) P.9, II.245-247: “UnaG can be combined with a large number of targets for dual-color imaging”. If so, please show different targets! Concerning proposed three-color (from lines 249!): I believe it will suffer from bleaching problems, if not – please show!

9) Please show the mentioned IFP2.0 experiments!

10) Please show a colocalization sample that can be used for quality control, e.g. using Glycine receptors (stable 2 β :3 α subunit mix) or origamis.

11) Supplemental video: 1s SR-snapshots is really fast – can the authors please provide the original tif-stack? Imaging speed is not provided! What was the exact frame rate?

Other comments:

1) The introduction in my opinion has a wrong focus. Instead of introducing colocalization (not covered in the manuscript) current labeling systems should be introduced and discussed as this is the topic of the manuscript by introducing a new option. Especially fluorogenic systems with a focus on the class of proteins holding an external chromophore should be introduced. Thus, authors should introduce and discuss e.g. FIAsh, SNAP, CLIP, Halo, PYP, CRABP1, BC2, FAST, IFPs with BV etc. already here, not only IFPs with BV in the discussion.

2) Introduction, discussion multi-color imaging: Next to spectrally distinct FPs or by demixing overlapping FPs, one can also use correlative approaches, e.g. combination with PAINT or STORM (e.g. pubs.acs.org/doi/pdf/10.1021/nl304071h or [10.1039/c5ib00077g](https://pubs.acs.org/doi/pdf/10.1039/c5ib00077g)), different activation doses (<https://onlinelibrary.wiley.com/doi/full/10.1002/anie.201509649>), reverse activation mechanisms (<https://www.nature.com/articles/nbt.1493>) or different activation mechanisms (<http://www.mdpi.com/1422-0067/18/7/1524>). I believe that all these ideas are worth mentioning when it comes to dual color approaches.

3) P.11, II.287-289: Working with aberration-free methods circumvents this nicely and is commonly done in recent studies!

Minor:

4) P. 4 II. 90 -96: “dim and bright images” are not appropriate descriptions for reconstructed images, which are at best intensity-mimicking histograms of localizations. I had to read this sentence several times. Also, when mixed-up with the notion that oxygen scavenging indeed raises the photons per imaged spot, it is even more difficult to entangle this.

Reviewer #3 (Remarks to the Author):

In their manuscript Kwon and coworkers elegantly exploit chromophore destruction and release as well as subsequent rebinding of a fresh molecule of bilirubin (BR) to the protein UnaG to achieve fluorescence on/off transitions needed for single molecule localization super resolution techniques. While the manuscript presents no “new development” per se it includes a very nice characterization of the mechanisms of this phenomenon and the demonstrated application in super-resolution imaging. The described use of UnaG is a noteworthy addition to the field of existing dyes and proteins for super-resolution imaging. The manuscript is soundly written and the made experiments are appropriate for the general characterization as well as for the mechanism of UnaG-BR binding/release. Nonetheless, I would request the authors to make some additional explanations and experiments to judge the applicability of the present version of UnaG for different imaging scenarios. Together with such experiments, I can recommend the work for publication in Nature Communications.

In particular:

- The spectra in Fig. 1b show a slight decrease in absorbance of the rebound UnaG vs. the “fresh” UnaG. Further switching is only demonstrated as images (Fig. 1c). This small decrease stays uncommented. It would be great if the authors could provide switching trajectories of > 10 cycles on the same protein solution to judge i) potential damage on the protein with repeated switching and ii) quantification of the depletion of the BR reservoir.
- The same experiment would be interesting in live cells to judge the potential for longitudinal imaging. E.g. how long can a cell be imaged until some form of photo-fatigue can be observed? And does the reported BR additions to the buffer last quasi unlimited or is a depletion eventually noticeable and requires renewal of the buffer?
- The authors could add a basic in vitro analysis of dependence of the kinetics on cellular factors like pH and ROS levels.
- The authors allude to different compartments and their respective BR concentrations or potential for uptake. It would be nice if they could extend their set of application examples to another compartment e.g. the nucleus. To demonstrate that BR accessibility poses no limitation to imaging basically every cellular structure.
- Along the lines of the longitudinal experiments the authors should demonstrated multiple time step measurements with all their samples and added BR concentrations (and produced photo oxidation products). Data for prolonged BR incubation is stated but not shown. This data is important to judge any perturbation of the imaging by addition of BR. Because this is clearly the main difference to fully light modulatable labels, that they have to add a molecule and produce “waste” in the cell. Consequently, by providing information on the ranges of BR addition, oxygen quencher concentrations and incubation time the authors should demonstrated the boundaries of this method. How toxic the light of the other methods eventually is, is another topic.

Point-by-point Responses to Reviewer's Comments for *Nature Communications* Manuscript

Reviewer #1 (Remarks to the Author):

The authors detail the application of UnaG as a versatile super-resolution and single molecular tracking dye in the submitted manuscript. To this end, the authors suggest a unique mode of fluorescent switching compared to previously established dyes, based on the photo-oxidation of the UnaG ligand, bilirubin, to turn off the fluorescent signal without irreversibly damaging the UnaG protein. Based on the outlined mechanism, the authors demonstrate the versatility of UnaG in multi-colour super-resolution imaging of live cells. Overall the concept and practical application of UnaG as a super-resolution dye represent a novel approach.

Response: We thank that the reviewer acknowledged novelty of our work. We also appreciate his constructive comments on the photo-oxidation process of bilirubin (BR) bound on UnaG protein, including the analysis of the photo-oxidation products, the photo-oxidation mechanism, and the fluorescence switching model. Following the reviewer's suggestion on increasing the resolution of MS analysis, we performed new experiments (high-resolution MS) and added new figures (Fig. 3b and S3e,f). The new results helped us to clarify the mechanism of oxidation reaction. Details are as below.

Major criticisms:

1. The general mechanism of bilirubin photooxidation and the involvement of a singlet oxygen species were extensively discussed in literature (e.g. McDonagh, *Biochem. Biophys. Res. Commun.*). Nevertheless, while the singlet oxygen driven mechanism is the primary accepted mechanism, the MS and MS/MS evidence provided by the authors for the formation of the suggested products 1 and 2 is weak. The points of concern are the combination of

- a) low resolution MS data of the suggested oxidation product.
- b) lack of peak peaking, especially in case of Fig. S3e and to some degree Fig S3d.
- c) precursor isolation width of 3 Da, very low accuracy for precursor mass selection.
- d) while carboxylic acid containing ions tend to exhibit low precursor ion intensity in MS/MS scans, it is not clear whether the complete absence of the precursor ions in the MS/MS spectra in Fig. 3C is due to complete fragmentation or due to the selected observation range of the second mass scan (not specified in the methods section).
- e) no reference spectra are provided or cited (possibly due to unavailability).

While improved mass spectral data, such as high resolution MS, or MS/MS or reference material would be highly recommended, alternatively the discussion on page 6 should more clearly emphasize the highly speculative nature of the proposed oxidation products.

Response: The reviewer mainly pointed out that our previous MS and MS/MS resolutions and data presentation were not sufficient to figure out the photo-oxidation products. To address the problems, as suggested by the reviewer, we re-measured the MS(/MS) results with high-resolution instruments.

We performed new experiments using a separate high-resolution orbitrap mass spectrometer in order to improve the quality of MS results, and thus to accurately predict the structures. Now we achieve 70,000 of resolution for full-MS scan and 17,500 of resolution for ddMS² mode. The precursor isolation width is also reduced from 3 Da to 2 Da, which provides better accuracy for the mass selection. With these experimental conditions, we clearly confirmed that the *product1* is the most abundant species among the photo-oxidation products (Fig. 3b and Fig. S3e,f). We focused on the peak at 315.1336 in Fig. 3b for identifying the most abundant product of the photo-oxidation reaction. On the other hand, the significance of the *product2* is doubtful, since a very small amount of the *product2* was observed in the revision experiments repetitively. Thus, in the revised manuscript, we focus only on one major oxidation product, *product1*, which is already found in a previous study by Matteis et al. (*Rapid Commun. Mass Spectrom.*, 2006). Now we call *product1* as OxBR in the revised manuscript. The changes in manuscript related to MS are summarized below.

- 1) The major photo-oxidation product is called 'OxBR' rather than *product1*.
- 2) The *product2* and related reaction mechanism are deleted from the main text and supplementary information.
- 3) Full-MS scan and ddMS² mass spectra for OxBR are replaced with high-resolution results (in Fig. 3b and S3e,f).
- 4) MS/MS result now clearly shows the precursor ion (in Fig. S3e).
- 5) Experimental details for high-resolution MS (HRMS) and HRMS/MS are added in the method section.

2. Fragments in fig 3c should be denoted following standard MS/MS procedures, such as specification of [M-H₂O]⁺ etc and all fragmentation points should be clearly marked in a single structure of the precursor ion. This will circumvent ambiguity or potentially misleading fragment structures, such as the proposed dehydration of the precursor ion at the carboxylic acid function giving rise to the 297.1 fragment. In general, a protonated carboxylic acid should be more

likely associated with the formation of an acylium ion due to its resonance stabilization and not an enol (Middlemiss *Can. J. Chem.* 57, 1979), if the authors suggest that the carboxyl function is protonated under the ionization conditions.

We assigned the fragment ions observed in the high-resolution MS/MS data for OxBR, which have >10% of relative abundance to the most abundant fragment, with conventional notation as the reviewer suggested, and summarized the result in Fig. S3e, f.

3. The depicted Br oxidation in Fig. S4 by singlet oxygen to give rise to the two proposed oxidation products are in agreement with previous reports and supported by the demonstrated oxygen dependency of the observed oxidation. Unfortunately, the suggested localization of the oxidation, inside UnaG, is highly speculative and no evidence has been provided to support this notion. Based on previously published X-ray data, presence of oxygen inside the binding pocket of UnaG-Br has not been reported. Additionally, formation of a Br radical at the bridging methylene carbon would highly likely be able to form a novel covalent bond between UnaG and its ligand, rendering the protein unable to rebind a non-oxidized Br ligand effectively constituting photobleaching. To support the proposed oxidation location, the authors need to provide evidence that singlet oxygen is inside the UnaG binding pocket and explain why carbon radical formation does not lead to Br-UnaG covalent bond formation and photobleaching.

Response: Following the reviewer's request on additional supporting data for the photo-oxidation mechanism specific to holoUnaG, we experimentally addressed two main questions (singlet oxygen inside UnaG and formation of BR-UnaG covalent bond) by kinetic measurement and MALDI-TOF analysis (Fig-referee 1).

In order to provide an evidence for generating reactive oxygen species (ROS) inside the binding pocket of UnaG, we measured the off-switching rate in various ROS conditions. We used sodium L-ascorbate as an antioxidant effectively quenching the radicals (A. E. Wagner et al., *J. Agric. Food Chem.*, 2008), and catalase-free GLOX as a ROS generating system (M. K. Weibel et al., *J. Biol. Chem.*, 1971). Both of different ROS conditions did not significantly affect the off-switching rates (Fig. 2c). From these results indicating that the singlet oxygen species outside the protein do not affect the photo-oxidation rate, we postulate that the photo-excited BR in holoUnaG reacts with O₂ inside the protein and generates ¹O₂, which is immediately used in further oxidation processes before escaping from the binding pocket.

To check whether the carbon radicals produced during photo-oxidation reaction react with amino acids, we carried out photoreaction on the mixed sample of free BR and free amino acids (amino acid standard, AAS18, Sigma Aldrich). Resulting product was analyzed by MALDI-TOF. We could not detect BR adducts covalently bonded to any amino acid, indicated in the figure below. Based on the observation, we conclude that UnaG photoreaction mainly oxidizes the BR rather than produces covalently linked BR adducts.

Fig-Referee 1. MALDI-TOF analysis of photo-oxidation products of bilirubin (BR) and amino acid mixture. Mass spectra of amino acid standards (blue) and photo-oxidation products (red) are shown for the m/z windows of the total range (top), 0-300 for amino acids (middle) and 550-850 for free BR and BR-adducts (bottom).

4. As the existence of damUnaG1/2 is highly unlikely and not supported by experimental evidence, the authors will need to either adopt their proposed model by excluding damUnaGs or as outlined in point 3 above, demonstrate the existence of damUnaGs. At this point it seems likely, that omission of damUnaG species will lead to a more simplified model, with equally good fitting properties.

As the reviewer suggested, we simplified our on-switching model with the consideration of new experimental results. According to the comment #2 of Reviewer #2, we re-measured the on- and off-switching rates in a completely different experiment of single-molecule pull-down assay on the same microscope that we performed STORM imaging. In this new model, the photo-oxidation of holoUnaG1 and 2 produces a single product, damUnaG - the complex of UnaG and the oxidation product of BR (OxBR). Since our new experimental results require the existence of damUnaG, we cannot omit it from our model. In our simplified model, to recover the fluorescence again, the damUnaG must release the OxBR for emptying the binding pocket. After that, the UnaG with free binding pocket, apoUnaG, can capture a freely diffusing BR molecules to form holoUnaG1, which spontaneously transform to the brighter state, holoUnaG2. We obtained the analytical solution for this model to fit the experimental results (Fig. 2e and S2), and finally could derive the dissociation rate of OxBR, as well as the on-switching rate.

Minor comments:

5. Please specify the amount of UnaG used for the experiment of Figure 2.

150 nM of holoUnaG was used to the ensemble kinetics for the Fig. 2 in the first submitted version of manuscript. Please note that we totally replaced the figure with new on-microscope measurements. In this scheme, we measured the on- and off-switching rates from UnaGs from lysates of transiently expressing mammalian cells, thus we unfortunately cannot specify the amount of UnaG used.

6. page 10, line 267. The data of IFP2.0 should be presented.

Now we present the IFP2.0 experiment as a supplementary figure (Fig. S13). We expressed Sec61 β -IFP2.0 in Cos7 cells and bleached it with intense excitation laser at 647 nm. After bleaching, we added external biliverdin ligands to

see if IFP2.0 can restore its fluorescence again. After 10 mins, no fluorescence for IFP2.0 was recovered. On the other hand, Sec61 β -labeled UnaG and the external BRs clearly recover the fluorescence with the similar experimental procedure.

7. I wonder where the values of k_{on1} , k_h and k_h' shown in Figure S2a come from.

The kinetic rates used for numerical simulation came from a previous study of UnaG-BR complex (Y. Shitashima et al., *Biophys. J.*, 2017). The values were $7.05 \times 10^6 \text{ s}^{-1}$, $1.72 \times 10^{-3} \text{ s}^{-1}$ and $2.62 \times 10^{-3} \text{ s}^{-1}$ for k_{ahl} (= k_{on1}), k_h and k_h' , respectively. In the revised theoretical and experimental kinetic study, we do not use any previously reported kinetic rates for the analysis.

Reviewer #2 (Remarks to the Author):

The manuscript “Bright Ligand-activatable Fluorescent Protein for High-quality Multicolor Live-cell Super-resolution Microscopy” introduces and characterizes the UnaG/bilirubin system as a fluorescent labeling system for localization microscopy. The labeling system is rather small (about half the size of GFP) and can be “refueled” by the pool of available bilirubin – both are good reasons to check and evaluate it as an interesting and promising SMLM labeling option. I nevertheless believe that the system is not convincingly characterized in the current manuscript and doesn’t prove the claimed SMLM quality. Therefore, I would like to ask the authors for several additional experiments and clarifications, which could appease my concerns as explained below:

Response: We appreciate that the reviewer found a number of weak points in our previous manuscript mainly on the labeling issues and SML imaging, including cytotoxicity problem, appropriate kinetic measurements, multicolor and live-cell imaging capacities, as well as more suitable contents to be introduced. All of the comments greatly assisted to reinforce our manuscript. We tried to experimentally address all the concerns as point-by-point responses below and included the additional results in new figures (Fig. 2,4,5 and S2,5,6,9-14) and texts (introduction, kinetics, live-cell multicolor imaging and discussion).

1) Please state for all images in the figure caption as well as in the text if the images were taken on live or fixed cells. This is an important detail.

Following the reviewer's suggestion, we added the statements for the cell conditions used in each experiment in the figure captions, as well as in the main text.

2) Data of Figure 1 and 2: I do believe the linear dependencies on BR concentration and light intensity as shown in Figure 2. I nevertheless don’t believe that they explain the statistics shown in Figure 1 and the behavior of the probe for SMLM conditions. Figure 2 was measured in the 0-80 mW/cm² 488 nm irradiation regime (actually, what were the intensities used for 2c and e – I believe e) is the same blue curve as in a) thus 80 mW/cm²?), Figure 1 is measured with 300 W/cm², an increase in laser dose of 3750-fold and in a changed buffer (with/without oxygen). I strongly believe that at these really high intensities used for STORM-like imaging, different mechanisms and transitions occur as compared to the low laser intensity regime and that these different switching characteristics will not show a simple linear dependence extending from the graphs shown in Fig.2. This can also be simply estimated from the authors data as the product of +BR-OS photon count peaking about ~130 and locs per 1000 frames with 300 is about 39.000 and photon peak of +BR+OS with 190 times ~ 500 locs/1000 fr. is 95.000  the ratio of this is only a 2x increase in total number of detected photons not matching the prediction using the rates of figure 2. Thus, please i) measure the rates and kinetics at relevant settings (thus high intensities and corresponding buffers, e.g. prove the predicted 13 1/s off-rate!). ii) Please proof that the off-switching under STORM conditions is pure photo-oxidation and no additional transitions like e.g. some fraction in a reversible dark state making them blinking in a STORM-like manner is involved (hint already measured in Ref 43). iii) please measure the dissociation rate of the photo-oxidized product?

Response: Here the reviewer's concerns were mainly based on the difference of conditions between the kinetic measurements and the SML imaging. We totally re-measured the kinetic rates under SML imaging conditions to provide more appropriate rate constants for UnaG photoswitching.

Instead of our previous fluorometer experiments that we could not increase the irradiation intensity due to instrumental restrictions, we measured the on- and off-switching rates in the same STORM microscope by using single-molecule pull-down assay of UnaG-Aptag proteins extracted from transiently expressing Cos7 cells. We replaced Fig. 2 with a completely new figure constructed from the new STORM-relevant measurements. At high light intensities and fast time resolutions that are used in STORM imaging, we found that there actually exists a nonlinear property on the off-switching rate, where additional faster off-switching event appears, giving bi-exponential behavior rather than single-exponential decaying (Fig. 2a). Hereafter, we call the slower and faster off-switching rates as k_{off1} and k_{off2} , respectively. The off-switching rates linearly depended on the irradiation intensity, and the dependency of k_{off1} was well corresponded to our previous result giving ~6.4 s⁻¹ of off-switching rate at 300 W/cm². We concluded that both of the two off-switching events are related to the photo-oxidation of bilirubin (BR) ligand for the following reasons: (1) If we

did not supply external BR, no fluorescence was recovered (Fig. S5, The experiment also supported that there are no other fluorescence transitions mediated by metastable dark states); (2) Depleting the dissolved oxygen with GLOX reduced both of off-switching rates ~3.5-times to the reference values, as the reviewer correctly estimated from the photon peaking results shown in Fig. 1; (3) Both rates were unaffected by conditions that influence intersystem crossing (MEA and KI in Fig. 2c) and the concentration of reactive oxygen species [Ascb and GLOX (-CAT) in Fig. 2c]. The results with MEA and KI also indicate that the off-switching reactions do not involve in reversible dark states such as thiol adduct and triplet state in conventional STORM conditions.

The enhanced time resolution for on-switching rate measurement also revealed much more complex behavior than previously observed single-exponential rising (Fig. 2e). To address the complicated fluorescence recovery, we revisited the switching model. With the derivations described in the section titled as "Derivation of Analytical Solutions for On-switching Kinetics" in SI, we obtained the analytical solution for the fluorescence recovery that fitted well to the experimental results (Fig. 2c and S2e). The analytical solution in Eq. 14 that is used to fit the experimental data was obtained with the assumption that the dissociation of oxidized BR is infinite. Even the solution in Eq. 13 without the assumption of infinite dissociation rate could reproduce the bi-exponential rise observed in the experiment only when the dissociation rate is much faster than the frame rate. With these modeling results, we conclude that the dissociation rate is too fast to measure under our experimental conditions.

3) Phototoxicity: The authors mention several times that UV light is phototoxic (introduction, live-cell results, discussion). With this I generally agree but I highly believe that 300 W/cm² of 488 nm light as used by the authors is actually much more phototoxic than the usual mW/cm² regime of 405 nm light used for photoactivation of fluorescent proteins. Please proof that e.g. imaging the biological targets of Fig. 4 by the UnaG system is causing less cell death than imaging the same targets (at similar SR image quality) by e.g. mEos3.2 fusions (mEos2 is oligomerising at higher densities and thus not very useful to image dense structures as e.g. vimentin (see publ. Zhuang group, Wang et al PNAS).

Response: As the reviewer mentioned, high-intensity 488-nm laser is known to induce phototoxic effect to live cells. Thus, it is important to measure the phototoxicity of our STORM imaging conditions to inform the readers interested in live-cell imaging. For this purpose, we performed long-term bright-field imaging of UnaG-expressing cells under high 488-nm illumination and in buffer supplements used in STORM imaging.

Under high 488-nm intensities, when no supplements such as bilirubin and oxygen scavenging system was used, some cells become frozen (S. Wäldchen et al., *Sci. Rep.*, 2015) due to photo-crosslinking (Fig. S10b). In contrast, the usual levels of 561- and 405-nm lights that are used in EosFP STORM imaging caused apoptosis in a fraction of cells, indicating DNA damage by near-UV light (Fig. S10c). These results indicate different mechanisms of cytotoxicity under different illumination wavelengths.

Interestingly, under the 488-nm illumination condition with only the oxygen scavenging system (GLOX) added to the solution, cell freezing was retarded (Fig. S10d). In contrast, when only bilirubin is added, cell freezing was accelerated (Fig. S10e). When both bilirubin and GLOX are supplemented, the freezing phenomena was nearly eliminated (Fig. S10f). These results indicate that the supplemented BR acted as antioxidant (R. Stocker et al., *Science*, 1987) prevented photo-crosslinking (N. Kilian et al., *Nat Methods*, 2018) under low oxygen levels. But, under high oxygen levels, the reactive oxygen species (ROS) produced during photo-oxidation of BR (Bonnert et al., *J. Chem. Soc., Perkin Trans. 1*, 1975) can promote the photo-crosslinking and cell freezing. Indeed, the ROS levels measured by fluorescent ROS indicator supported the above-mentioned toxicity mechanism (Fig. S10h).

Overall, we proved that the illumination and buffer condition for UnaG STORM imaging resulted in slightly better cell survival than the condition for EosFP (Fig. S10g). Now, we added the new toxicity results in the first paragraph in "Live-cell Multicolor Super-resolution Imaging" section in page 9 and explicitly compare the cytotoxicity of STORM imaging conditions of UnaG and EosFP.

4) Are biological targets disturbed by UnaG labels and BR? E.g. steric hinderance, oligomerization artefacts of UnaG, possible ROS damage by the BR radicals in photo-oxidation processes, possible metabolic reactions on oxygen depletion, effects of non-physiological concentrations of ~ 1 μM BV (for the latter: 1μM BV control data is not shown and 2h incubation is not long enough for visual cell shape read-out), please characterize the BR uptake for living cells etc.? Authors need to control ALL of this, measure and characterize ALL the effects and to do comparative experiments with common labels to show that the UnaG system yields better compromises than other labeling solutions.

Response: As the reviewer pointed out, it is important to characterize side effects of UnaG expression or high bilirubin concentration in live cells because fluorescent proteins and external reagents often perturb cell's conditions in deleterious way. We experimentally addressed all the issues that the reviewer mentioned as below.

First, we examined the brightfield images of live cells expressing various UnaG-fused proteins (Fig. S9a) and found that the expression of UnaG did not induce any significant change in morphology. Consistent to the previous work which indicates monomeric state of holoUnaG (Y. Shitashima et al., *Biophys. J.*, 2017), UnaG-labeled vimentin showed quite similar structure to EGFP- and mMaple3-labeled vimentin. We noted that all the cells transiently expressing vimentin fused with fluorescent proteins (EGFP, mMaple3, and UnaG in Fig. S9b) showed slightly different pattern

from antibody-stained vimentin (Fig. S10b), implying that the subtle artifact stems from overexpression. Note that overexpression artifact can be eliminated by generating knock-in cell lines.

Second, BR radicals produced from photo-oxidation process during SML imaging may induce ROS damage to the cell. We used a fluorescent ROS indicator to check whether the intracellular [ROS] increases under SLM imaging conditions (Fig. S10h). As the response to the comment #3, the intracellular [ROS] showed little difference before and after the light illumination under SML imaging conditions (+GLOX, +BR).

Third, when cells are placed in oxygen-depleted conditions for a long time, the cells can fall into hypoxic condition. But transient depletion of the oxygen induces little noticeable damage on cellular metabolism as demonstrated in a previous study that characterized the effect of GLOX on live-cell dynamics up to 20 minutes (S. A. Jones et al., *Nat. Methods*, 2011), which is clearly longer than the acquisition times used in this study. In fact, a recent study (N. Kilian et al., *Nat Methods*, 2018) showed that oxygen scavenging can help reduce short-term photodamage by reducing ROS generated from excitation of fluorophores. Our experimental results also showed that the cells can survive more than 3 hours in the oxygen-depleted condition (Fig. S10).

Fourth, the possible harmful effects from external BR at high concentration were monitored by using fluorescent live-dead assay (Fig. S10a). We incubated the cells in different concentration of BR up to 3 μM and checked the viability after 24 hours. No significant differences in viability were observed in all cases.

For the BR uptake by live cells, please refer our response to the comment #7 below.

Additionally, a previous study that used UnaG as a sensor for intracellular [BR] distribution reported no noticeable distortions from the UnaG labeling in various sites in the cell (J.-S. Park et al., *ACS Chem. Biol.*, 2016).

5) Please do comparative photon counts for Eos and Atto488 on your system, these heavily depend on exact setup parameters.

We agreed to the reviewer's opinion and measured the photon counts of blue-absorbing fluorescent probes on our experimental setup for fair comparison (Fig. 4c). We used mMaple3 instead of Eos because mMaple3 is as bright as Eos but induces less oligomerization. Under the same experimental conditions that UnaG gave 1,220 photons per switching cycle, the photon counts of mMaple3 and Atto 488 were measured as 980 and 1,450, respectively.

6) FIRE values are highly doubtful. I don't believe the stated 18 nm. FIRE values have to be corrected for self-blinking effects, e.g. caused by highly blinking dyes or by several localizations on adjacent frames – please revisit original publication (“Q value”). Also, please do calibration experiments on standards, e.g. single-molecule surfaces with UnaG yielding FWHM of single protein localization distributions or use other common standards (NPC, oligomers, origamis etc.). Compare those to other labels! In this context, also Figure 5 and 6 don't look convincing. My guesstimate by eye would rather be ~100 nm resolution, which, on these targets can be reached by deconvolution. In this sense, SMLM fitting here acts as a background removal tool and thus results in contrast improvement and not so much in a resolution improvement.

To present the super-resolution imaging performance of UnaG appropriately, we avoided doubtful FRC analysis and added more commonly used repetitive localization distribution from single UnaG molecules (Fig. 4d). The mean localization precision was measured as 16.9 nm, which corresponds to ~40 nm of resolution.

Fig. 5 and 6 demonstrated live-cell imaging of UnaG-labeled ER and mitochondria, whose widths are few tens of nanometers for ER tubules and several hundreds of nanometers for mitochondria. Cellular motions during data acquisition can blur the SML images, worsening the overall imaging resolution. However, we found that the FWHM of single ER tubules in certain areas in Fig. 5 and 6 is 50-60 nm, suggesting the resolution is certainly better than 100 nm (see the figure below).

Fig-Referee 2. Transverse profiles of ER tubules imaged in live cells in Fig. 5a (left) and Fig. 6a (right). The average FWHM (full width at half maximum) from 5 tubules are denoted at the top-right corner of each plot.

7) Live-cell imaging: How does GLOX on the outside of cells help the labels in the inside? Ref. 50 imaged proteins ON the membrane. How do ON-rates change for live cells due to the needed uptake of BR (intracellular concentration is not sufficient!).

It is true that the intracellular [O₂] and [BR] are not easily controllable with external reagents. However, even with lower efficiency than *in vitro*, addition of GLOX or external [BR] in imaging buffer can reduce the oxygen level or

increase the intracellular [BR] in certain degrees to help the super-resolution imaging. To prove this effect, we measured the on- and off-switching rates inside the live cells, which were perturbed by GLOX and BR (Fig. S11a,b). Also, the dramatic difference on cellular phototoxicity of 488-nm illumination with and without BR and/or GLOX supports cellular uptake of exogenous BR and influence of extracellular oxygen scavenging on the intracellular oxygen concentration (Fig. S10c-f).

8) P.9, ll.245-247: “UnaG can be combined with a large number of targets for dual-color imaging”. If so, please show different targets! Concerning proposed three-color (from lines 249!): I believe it will suffer from bleaching problems, if not – please show!

We now included various two- and three-color live-cell images in Fig. S12. Two-color images of UnaG-Sec61 β and LysoTracker Red (Fig. S12a) and Mito-UnaG and ER-tracker Red (Fig. S12c) are added as additional examples for red membrane probes. As an example of red photoactivatable fluorescent protein, we added 2-color image of CLC-UnaG and TfR-PAmCherry (Fig S12b). Three-color image (Fig. S12d) was obtained from UnaG-Sec61 β , MitoTracker Red and HaloTag targeted to the extracellular side of the plasma membrane (TM-HaloTag) labeled with Alexa Flour 647. The PAmCherry and HaloTag examples suggest that genetic incorporation of PAmCherry and HaloTag to various targets can vastly increase the number of targets for multi-color imaging. Three-color imaging did suffer from bleaching problem, but it was possible to obtain 3-color image for highly expressed targets such as the plasma membrane (Fig. S12d).

9) Please show the mentioned IFP2.0 experiments!

We showed the non-switching nature of IFP2.0 as a supplementary figure (Fig. S13). Please refer our response to reviewer #1's comment (#6) for the details.

10) Please show a colocalization sample that can be used for quality control, e.g. using Glycine receptors (stable 2 β :3 α subunit mix) or origamis.

In Fig. S14, we imaged surface-immobilized UnaG-mCherry complex in single-molecule pull-down assay as a colocalization standard. This assay gave \sim 13 nm of mean distance between the centers of two fluorophores (Fig. S14c-e). Although mCherry does not photoswitchable characteristics, the center position can be precisely measured in low molecular density which ensures intermolecular distance to be longer than the diffraction limit. Here, the channel alignment error was measured as \sim 8 nm from fluorescent beads (Fig. S14a,b). Furthermore, to demonstrate the molecular colocalization in cells, we introduced CLC-UnaG-HaloTag construct to locate two fluorophores at the N- and C-termini of the same clathrin light chain (CLC). By labeling HaloTag with Alexa Flour 647, we performed two-color SMLM that yielded colocalized images (Fig. S14f). However, due to the \sim 13 nm colocalization error, we could not obtain distances between N- and C-termini of individual CLC molecules. Since technical development for reducing the colocalization error is out of the scope of the current work, we suggested potential technical advances in the discussion.

11) Supplemental video: 1s SR-snapshots is really fast – can the authors please provide the original tif-stack? Imaging speed is not provided! What was the exact frame rate?

We apologize the unintended mistakes in the figure and the caption. We revisited the raw data and found that the SR-snapshots in the video were taken for 10 sec. We also found that the slow speed and low localization density was due to high density of single molecules in the raw movie that resulted in overlapped images of single fluorophores and false averaging of analyzed positions. These artifacts were hard to notice due to irregular morphologies of ER tubules and sheets. During the review period, a new analysis software that separates overlapped images in different sub-frames, called HAWK (Marsh et al., *Nat. Methods*, 2018), were introduced. We used the software and reanalyzed the same raw movie. The newly analyzed STORM images contained significantly more localizations and allowed us to produce 1-sec snapshots in Fig. 5 and Supplementary Movie S1. For comparison of all the image reconstruction conditions, please see the following Fig-Referee 3.

Fig-Referee 3. Comparison of SML images for a Sec61 β -UnaG expressing live Cos7 cell in different time resolution and data processing. In first submitted manuscript, we displayed 10 s time resolution image by mistake (left). First 100 frames of raw data were used to reconstruct the 1 s time resolution image (middle) with low localization density and incorrect positions. Recently reported HAWK analysis helped to improve the image quality by separating the individual molecules from the overactivated raw data (right).

Other comments:

1) The introduction in my opinion has a wrong focus. Instead of introducing colocalization (not covered in the manuscript) current labeling systems should be introduced and discussed as this is the topic of the manuscript by introducing a new option. Especially fluorogenic systems with a focus on the class of proteins holding an external chromophore should be introduced. Thus, authors should introduce and discuss e.g. FIAsh, SNAP, CLIP, Halo, PYP, CRABPII, BC2, FAST, IFPs with BV etc. already here, not only IFPs with BV in the discussion.

We agree with the reviewer and completely re-wrote the introduction. The new introduction now focuses on labeling systems, following the reviewer's suggestion. We also included fluorogenic systems when introducing synthetic dye labeling.

2) Introduction, discussion multi-color imaging: Next to spectrally distinct FPs or by demixing overlapping FPs, one can also use correlative approaches, e.g. combination with PAINT or STORM (e.g. pubs.acs.org/doi/pdf/10.1021/nl304071h or [10.1039/c5ib00077g](https://doi.org/10.1039/c5ib00077g)), different activation doses (<https://onlinelibrary.wiley.com/doi/full/10.1002/anie.201509649>), reverse activation mechanisms (<https://www.nature.com/articles/nbt.1493>) or different activation mechanisms (<http://www.mdpi.com/1422-0067/18/7/1524>). I believe that all these ideas are worth mentioning when it comes to dual color approaches.

We added the references that the reviewer suggested in the introduction (page 3) and multicolor imaging section (pages 9-10).

3) P.11, ll.287-289: Working with aberration-free methods circumvents this nicely and is commonly done in recent studies!

We added the reference in the last paragraph of the discussion (page 12).

Minor:

4) P. 4 ll. 90 -96: "dim and bright images" are not appropriate descriptions for reconstructed images, which are at best intensity-mimicking histograms of localizations. I had to read this sentence several times. Also, when mixed-up with the notion that oxygen scavenging indeed raises the photons per imaged spot, it is even more difficult to entangle this.

"Dim and bright images" refer to raw, unprocessed fluorescence images of single molecules rather than reconstructed super-resolution images. To clarify this point, we change the phrase "the single-molecule images of UnaG were dim" to "because the fluorescence of many single molecules per each camera frame was too weak to detect above the signal-to-noise threshold (Fig. 1e)".

Reviewer #3 (Remarks to the Author):

In their manuscript Kwon and coworkers elegantly exploit chromophore destruction and release as well as subsequent rebinding of a fresh molecule of bilirubin (BR) to the protein UnaG to achieve fluorescence on/off transitions needed for single molecule localization super resolution techniques. While the manuscript presents no "new development" per se it includes a very nice characterization of the mechanisms of this phenomenon and the demonstrated application in super-resolution imaging. The described use of UnaG is a noteworthy addition to the field of existing dyes and proteins for super-resolution imaging. The manuscript is soundly written and the made experiments are appropriate for the general characterization as well as for the mechanism of UnaG-BR binding/release. Nonetheless, I would request the authors to make some additional explanations and experiments to judge the applicability of the present version of UnaG for different imaging scenarios. Together with such experiments, I can recommend the work for publication in Nature Communications.

Response: We thank the reviewer for the incisive comments particularly on live cell SML imaging. To address the reviewer's comments in the following, we characterized cytotoxicity and switching properties in living cells. We also performed all control experiments requested. Details on how we addressed the reviewer's comments are as below.

In particular:

- The spectra in Fig. 1b show a slight decrease in absorbance of the rebound UnaG vs. the "fresh" UnaG. Further switching is only demonstrated as images (Fig. 1c). This small decrease stays unmentioned. It would be great if the

authors could provide switching trajectories of > 10 cycles on the same protein solution to judge i) potential damage on the protein with repeated switching and ii) quantification of the depletion of the BR reservoir.

We agreed with the reviewer and measured the fluorescence switching of holoUnaG for 10 cycles of repetitive photobleaching and recovery in live cells, especially in actual SLM imaging conditions (Fig. S11c). The maximum fluorescence intensity gradually decreased, possibly due to the following reasons.

(1) The transition rate between holoUnaG1 and holoUnaG2 ($\sim 0.001 \text{ s}^{-1}$) may require more time for complete fluorescence recovery. In the measurement in Fig. S11c, we waited 5 minutes for each cycle of fluorescence recovery to finish the entire measurement before the pH of the imaging buffer drops down significantly due to the oxygen scavenging system.

(2) Under the imaging conditions with high intensity of 488-nm laser light, the local concentration of free BR around the imaging area could be decreased due to the photo-oxidation of free BR. This effect appeared in the gradual decrement of k_{on} (the binding rate of BR to UnaG) in the same experiment (Fig. S11e).

We suppose that the protein damage is insignificant regarding the nearly identical off-switching behavior for each of the 10 switching cycles (Fig. S11d).

- The same experiment would be interesting in live cells to judge the potential for longitudinal imaging. E.g. how long can a cell be imaged until some form of photo-fatigue can be observed? And does the reported BR additions to the buffer last quasi unlimited or is a depletion eventually noticeable and requires renewal of the buffer?

Under the SML imaging conditions, where the imaging buffer contains both of exogenous BR and GLOX, live cells showed no severe photo-fatigue after <15 min illumination of intense (300 W/cm^2) 488-nm laser (Fig. S10d-f). Notably, the viability strongly depended on the composition of imaging buffer (Fig. S10g). Please refer our response to the Reviewer #2's comment (#3) for the details.

For the concentration of external BR, we suppose that the used concentration ($1 \mu\text{M}$) is not enough to unlimitedly recover the fluorescence of UnaG, since the intense irradiation induced direct photo-oxidation of external BR significantly as shown in Fig. S11e.

- The authors could add a basic in vitro analysis of dependence of the kinetics on cellular factors like pH and ROS levels.

During the revision period, we re-measured the whole kinetics under actual SMLM conditions as suggested by the reviewer #2 (Fig. 2). In this new setting, we measured the off-switching rates in various pH and ROS conditions as the reviewer suggested (Fig. 2c,d).

- The authors allude to different compartments and their respective BR concentrations or potential for uptake. It would be nice if they could extend their set of application examples to another compartment e.g. the nucleus. To demonstrate that BR accessibility poses no limitation to imaging basically every cellular structure.

We demonstrated the SML imaging of UnaG-fused lamins inside of the nucleus (Fig. 4e,f and S8a). Our demonstration of SML imaging covered almost all the subcellular locations, from inside the nucleus (lamin) to cell membrane (clathrin).

- Along the lines of the longitudinal experiments the authors should demonstrated multiple time step measurements with all their samples and added BR concentrations (and produced photo oxidation products). Data for prolonged BR incubation is stated but not shown. This data is important to judge any perturbation of the imaging by addition of BR. Because this is clearly the main difference to fully light modulatable labels that they have to add a molecule and produce "waste" in the cell. Consequently, by providing information on the ranges of BR addition, oxygen quencher concentrations and incubation time the authors should demonstrated the boundaries of this method. How toxic the light of the other methods eventually is, is another topic.

As the reviewer mentioned, it is important to characterize the toxicity of our technique for longitudinal live cell imaging. We measured the cytotoxicity of BR, potential ROS damages from BR photo-oxidation, and the phototoxicity due to intense 488-nm irradiation (Fig. S11). Please refer to our response to the Reviewer #2's comment (#3 and 4) for the details.

Reviewers' comments:

Reviewer #1 (Remarks to the Author):

The authors have answered my comments seriously. I think that the manuscript has been substantially improved.

Reviewer #2 (Remarks to the Author):

I first of all would like to thank the authors for their detailed revision which significantly changed and strengthened the manuscript by many extra experiments and work conducted during the revision phase. Nevertheless, one important aspect of my last review remains on all scales: I asked for all those further experiments and validations so that the authors proof their claim that their new method is not "yet another method" but actually has some clear advantages over existing methods and produces convincing SMLM results. I am not at all convinced by all SMLM images given (see comments below).

Even with all the extra work I honestly still doubt all points the authors claim in this sentence (II. 228-231):

"In contrast, UnaG offers unique advantages: (1) simple labeling method of standard genetic incorporation; (2) high photon numbers comparable to EosFP and Atto 488; (3) reversible switching between the dark and green fluorescence states; (4) fluorescence recovery by binding an endogenous metabolite (i.e. bilirubin)."

1) Simple labeling:

a. For me, there is no proof that the genetic tagging using UnaG is less interfering with biological structures than other tags. I am alerted when the authors state that the tagged structures are different than the native ones.

- Please proof that the tag is not interfering with biology

- I don't believe that the artifacts observed stem from over-expression and can be resolved by stable cell lines. Please proof!

b. Phototoxicity of approach

- Is this method less phototoxic than PAINT approaches or primed conversion? Please proof! (both need no additives as Glox or bilirubin, no UV light etc. like the comparative condition of 405 and 561 in SI figure 10g – so maybe use those for comparison instead UV.)

2) High photon numbers:

a. Localization precisions of 17 to 25 nm are really not the state of the art.

- With EosFp, better results are published (in the 10-15 nm range).

- Drift correction (as estimated from the coloc value of 13 nm) is really bad. Why not better?

- I am alerted when bilirubin signals are largely too weak as a reliable single-molecule fluorophore without oxygen scavenger systems (esp. when already imaged at really slow imaging rates of 200 ms per image). high quality SMLM needs high quality signals.

3) Reversible switching:

- 10s acquisitions are not very indicative for high-quality reversible switching. Please proof that switching can be maintained for LONG experiments, e.g. following structural dynamics over the course of minutes, 10 to 15 min is standard for live cell SMLM applications (see literature).

- Please provide a graph that shows the fluorescent spot density of an imaged area over time and how constant it is (so that resupply works reliably and at a stable rate and oxBR does not interfere)

- Even though I believe in the oxidation reaction the authors propose, I am missing an important experiment. I am wondering if large amounts of photobleached bilirubin interfere with the binding of fresh bilirubin to the UnaG molecules. Can the authors perform a competition experiment? E.g. in Figure 2, how do the dynamics look like when adding e.g. 50% photobleached bilirubin and 50% fresh bilirubin together in contrast to 100% fresh bilirubin (choose your favorite concentration).

- How to avoid the bleaching of bilirubin molecules in the solution (before being read-out being bound to UnaG)?

- Control of photoswitching: recording movies in "overactivated conditions" doesn't seem like a decent control of photophysics to me. Is it possible to record movies at lower, real single-molecule spot density?

4) Endogenous bilirubin

“endogenous bilirubin” is not really used for this technique. It is bleached away to get to a single molecule level. Then extra bilirubin is supplied to refill the UnaG. The endogenous level can't provide this.

- Please change this in the whole text (introduction, discussion), currently the text suggests that this method works easily in live cells with endogenous metabolites and doesn't need any uptake of any component.

- You can add that the advantage of endogenous bilirubin is that you have a fluorescent structure at the start which you can selected for, focus on, etc., then you bleach this one away and start the SMLM experiment using the newly supplied bilirubin.

Final SMLM image quality: I am not at all convinced by the SMLM image qualities of Figure 4 e and f, Fig. 6 and the supplemental dual and triple color images. Judging these images and the localization precisions given by the authors, this method cannot compete with other current labelling and SMLM imaging methods. Please investigate not only photon numbers but a) reversibility of switching, b) control over photophysics (seems really bad to me) c) labeling coverage, d) false positive rates, false negative rates etc., e.g. using standards as nuclear pore complexes which should give 16 or 32 clear spots. The images of a new technique must be highly convincing and significantly better than images obtained by the current methods in the field (taken from the literature). With such bad SMLM images like presented currently, it is only “yet another technique” that doesn't perform better and thus will not find application. Widths of ER tubules and vimentin in the 50 nm range are too high! If you would like to use cytoskeletal structures (and not nuclear pore complexes), then maybe use microtubules. These should be hollow in the middle, so give a double profile (e.g. see <https://www.nature.com/articles/nmeth.2214>).

Additional major comments:

200 ms is “fast switching” (p.9?) – no, image acquisitions in the 20-50 ms range are the state-of-the-art of being “fast” SMLM.

Why are the fluorescent spots (see Fig.4 d) asymmetric in x and y (by 10 nm?).

Minor comments:

- Reference 19 is a dye not a FP, I believe the authors would like to change “FPs” into “fluorophores” in their argumentation.

- The authors should exchange BC2 and PYP in their introduction. BC2 is a peptide tag and thus similar to the enzyme tags in the list, PYP is a fluorogenic tag (whereas Bc2 is not).

- Citation of the Minflux paper missing.

- Comment on the bleaching in the 3-color approach.

Reviewer #3 (Remarks to the Author):

All my concerns have been fully addressed.

Minor comment: Maybe it is a misunderstanding on my side, but the times for “on” and “off” in figure s11c state “20s” and “5min”, respectively. In the corresponding figures s11d and s11e however data is shown for 3s and 100s. This is, at least for me, confusing.

The length of the movie of the ER-remodeling is for 5s. If the authors acquired more images showing maxed out imaging time it would be nice to include them. In line with the their statements on the possibility for long imaging made in the toxicity studies.

However, overall I like the manuscript and the improvements in the revision process and see it fit for publications with Nat. comm.

Point-by-point Responses to Reviewer's Comments for *Nature Communications* Manuscript

Reviewer #1 (Remarks to the Author):

The authors have answered my comments seriously. I think that the manuscript has been substantially improved.

Response: We greatly appreciate the reviewer for approving our revised manuscript. We hope the reviewer was sufficiently satisfied with our previous responses.

Reviewer #2 (Remarks to the Author):

I first of all would like to thank the authors for their detailed revision, which significantly changed and strengthened the manuscript by many extra experiments and work conducted during the revision phase. Nevertheless, one important aspect of my last review remains on all scales: I asked for all those further experiments and validations so that the authors proof their claim that their new method is not “yet another method” but actually has some clear advantages over existing methods and produces convincing SMLM results. I am not at all convinced by all SMLM images given (see comments below). Even with all the extra work I honestly still doubt all points the authors claim in this sentence (ll. 228-231):

“In contrast, UnaG offers unique advantages: (1) simple labeling method of standard genetic incorporation; (2) high photon numbers comparable to EosFP and Atto 488; (3) reversible switching between the dark and green fluorescence states; (4) fluorescence recovery by binding an endogenous metabolite (i.e. bilirubin).”

Response: We thank the reviewer for admitting our efforts in the last revision period, and for additional professional advices that help us to improve our manuscript. The reviewer raises doubt mainly on the novelty of UnaG as a photoswitchable fluorophore and asks for noticeable advantages of UnaG for SMLM compared to other fluorophores. We agree that there are other fluorophores offering each of the above-mentioned advantages separately. However, there is no such fluorophore that simultaneously gives all the advantages of genetic incorporation, high photon numbers, and reversible photoswitching with chemical activation, all together. This combined set of benefits makes UnaG as a powerful fluorescent tag for the SML imaging. To emphasize the combinatorial advantage of UnaG more clearly, we edited mentioned expression as below.

“In contrast, UnaG offers the unique combination of following advantages simultaneously: (1) ...”

Also, we addressed each of the reviewer’s comments below by adding new discussions on the existing data (in the sections of “Super-resolution imaging of various subcellular structures” and “Live-cell multicolor super-resolution imaging”) and new experimental results (in Fig. 4d and Fig. S12). We detail the changes following the reviewer’s comments as below.

1) Simple labeling:

- a. For me, there is no proof that the genetic tagging using UnaG is less interfering with biological structures than other tags. I am alerted when the authors state that the tagged structures are different than the native ones.
 - Please proof that the tag is not interfering with biology.
 - I don’t believe that the artifacts observed stem from over-expression and can be resolved by stable cell lines. Please proof!

Response: We do not intend to argue that UnaG is much better than other FPs in terms of interference in biology. Instead, we provided evidences that UnaG’s perturbation is comparable to other least perturbing FPs. For instance, the structure of vimentin filaments labeled with UnaG (Fig. S9b) showed minimal structural distortions comparable to those of mMaple3 that is considered as minimally perturbing FP for SMLM [Wang et al., PNAS, 2014]. Also, there was no severe interference with the overall cell morphology (Fig. S9a). Moreover, there are a number of previous studies using UnaG as a common fluorescent tag for live cells [Erapaneedi et al., EMBO, 2016; To et al., Protein Sci., 2016; Navarro et al., ACS Chem. Biol., 2016; Park et al., ACS Chem. Biol., 2016; Hu et al, Free Radic. Biol. Med., 2018; and others]. Especially, Park et al. labeled ten subcellular structures with UnaG and proved that UnaG can be located in many cellular loci without morphological interferences to the target structures. Thus, we consider that UnaG can be regarded as one of the least interfering FPs.

We apologize that our discussion on biological interference in the previous manuscript failed to deliver the message and cause misinterpretation. To better convey our intension, we changed the discussion in the main text on Figure S9 in the new revised manuscript as follows.

“The expression of subcellular structures labeled with UnaG did not noticeably perturb the cellular morphology observed by a bright field microscope and the cytoskeletal distributions were similar to those labeled with other FPs that are known to minimally perturb the structure under the same condition (Supplementary Fig. S9).”

Also, we changed the related discussion in the caption of Figure S9 as follows.

“UnaG-labeled vimentin showed mostly thin filaments observed in the immunostained cells, with slight bundling comparable to those observed in other FPs with minimal dimerization characters such as EGFP and mMaple3.”

Note that in the two revised sentences above, we used the conservative terms of “similar” and “comparable”. However, the UnaG image in Fig. S9 showed less bundling than the EGFP and mMaple3 images in Fig. S9. EGFP is considered as intrinsically monomeric and mMaple3 has minimal dimerization character among SMLM FPs [Wang et al., PNAS, 2014]. We did not stress on the lesser amount of bundling in the UnaG image because the extent of bundling could be related to the expression level.

As suggested by the reviewer, creation of stable cell lines with endogenous expression level will help to better compare the bundling effects. However, creation of such stable mammalian cells requires a long period of months for generation, selection, propagation and characterization that is substantially longer than a typical second revision period. Since we spent many months for our first revision, we are afraid of further substantial delay on publication if we generate stable cell lines. We plan such comparison for a separate study after publishing the current work.

b. Phototoxicity of approach:

- Is this method less phototoxic than PAINT approaches or primed conversion? Please proof! (both need no additives as Glox or bilirubin, no UV light etc. like the comparative condition of 405 and 561 in SI figure 10g – so maybe use those for comparison instead UV).

Response: We agree that no chemical additive or no UV light will be less toxic than the current UnaG SMLM condition. However, the current versions of PAINT or primed conversion have critical disadvantages that intrinsically limit the live-cell applications.

PAINT utilizes binding and unbinding of fluorescent probes that are often incapable of penetrating through the cell membrane and/or can be chemically toxic to live cells. In fact, PAINT probes can be considered as chemical additives since the probes should be present in solution throughout the imaging period. Even when these critical issues for live cells are resolved, the binding and unbinding kinetics are often too slow to reconstruct SMLM images of live cells in reasonable time resolution. Also, the high background from unbound probes requires special illumination geometry such as total internal reflection [Schnitzbauer et al., Nat. Protoc., 2017] or lattice light sheet [Legant et al., Nat. Methods, 2016]. Due to these disadvantages, publications on live-cell PAINT are rare and limited to surface targets [Nieves et al., Genes, 2018].

Primed conversion clearly demonstrated less phototoxicity compared to common UV-activated FPs because it uses two visible lights (e.g. 488 nm + 730 nm) for photoconversion [Turkowsky et al., Angew. Chem. Int. Ed., 2017]. However, the most recent FPs for primed conversion produce far less photons per switching cycle (120-180 photons) than UnaG [Turkowsky et al., Fig. S8]. Also, it complicates the imaging instrument, especially with unusual light source at 730 nm and filtering optics.

For readers who wonders about PAINT and primed conversion as the reviewer did, we now added the related discussions at the end of the first paragraph for the section “*Live-cell multicolor super-resolution imaging*” that reads as follows.

“PAINT with no chemical additives or primed conversion with near-infrared activation may offer more cell-friendly imaging conditions, but their applications to live cells are limited by impermeability and chemical toxicity of the PAINT probes and other photophysical characteristics such as slow kinetics, high background (PAINT) or low photon numbers (primed conversion).”

2) High photon numbers:

a. Localization precisions of 17 to 25 nm are really not the state of the art.

- With EosFp, better results are published (in the 10-15 nm range).
- Drift correction (as estimated from the coloc value of 13 nm) is really bad. Why not better?
- I am alerted when bilirubin signals are largely too weak as a reliable single-molecule fluorophore without oxygen scavenger systems (esp. when already imaged at really slow imaging rates of 200 ms per image). high quality SMLM needs high quality signals.

Response: We agree to the reviewer's opinion about the values of the localization precision. To improve it, we reanalyzed the single-molecule localization data to better align multiple localization distributions into one. Mainly, we changed the center-finding algorithm for better alignment of individual UnaG molecule. Previously, we assumed that a single UnaG molecule is located at the average position of the multiple position values from a cluster of localizations. The average position was calculated from simple numerical means. In the revised version, we fitted the localization distribution with two-dimensional Gaussian function and the molecular position is assigned as the centroid of the Gaussian fit. This new center estimation procedure helped to reduce the error from combining multiple localization

distributions. Also, we added more data set for reducing the effect of insufficient sampling. The larger, better combined data set yielded localization precision of ~ 12 nm corresponds to ~ 29 nm in FWHM with better symmetry. These values are comparable to the 10-15 nm ranges with EosFP, as the reviewer pointed out. We replaced Fig. 4d with the new results.

With the new data with reasonable values, we rule out drift correction as the cause of low localization precision. The ~ 13 nm of uncertainty (Fig. S15e) that the reviewer mentioned mainly come from the miss-alignment of two different color channels and the localization uncertainties of the two channels. It can be reduced to ~ 8 nm with fluorescent beads which offers much larger number of photons, hence better localization precision (Fig. S15b). We aligned two color channels by applying a transform matrix with second-order polynomial function, where this strategy usually gives ~ 10 nm of alignment error [Lehmann et al., J. Biophotonics, 2016]. For the drift correction, we used image correlation process built in the ThunderSTORM, which is widely accepted [Marsh et al., Nat. Methods, 2018].

The third comment is related to the low photon number of UnaG without the oxygen scavenging system. We used the oxygen scavenging system because we aimed for the highest quality possible. Even though the faster off-switching reaction without the oxygen scavenging system (Fig. 2 and 3) resulted in the lower photon number per switching cycle (Fig. 1e), we could still obtain some single-molecule signals from UnaG that could be localized (Fig. 1d) to reconstruct a SML image if we consume enough acquisition time in that condition. We anticipate that the mechanism of photo-oxidation that we propose in this study help other researchers in protein engineering to develop new UnaG versions that do not require oxygen scavenging in the future.

3) Reversible switching:

- 10s acquisitions are not very indicative for high-quality reversible switching. Please proof that switching can be maintained for LONG experiments, e.g. following structural dynamics over the course of minutes, 10 to 15 min is standard for live cell SMLM applications (see literature).
- Please provide a graph that shows the fluorescent spot density of an imaged area over time and how constant it is (so that resupply works reliably and at a stable rate and oxBR does not interfere)
- Even though I believe in the oxidation reaction the authors propose, I am missing an important experiment. I am wondering if large amounts of photobleached bilirubin interfere with the binding of fresh bilirubin to the UnaG molecules. Can the authors perform a competition experiment? E.g. in Figure 2, how do the dynamics look like when adding e.g. 50% photobleached bilirubin and 50% fresh bilirubin together in contrast to 100% fresh bilirubin (choose your favorite concentration).
- How to avoid the bleaching of bilirubin molecules in the solution (before being read-out being bound to UnaG)?
- Control of photoswitching: recording movies in “overactivated conditions” doesn’t seem like a decent control of photophysics to me. Is it possible to record movies at lower, real single-molecule spot density?

Response: As the reviewer suggested, we provided further data on long-term live-cell imaging capacity in Fig. S12, with spot density vs time graph, which lasted more than 15 min. The data is from the same experiment to construct the images in Fig. 5 in the original manuscript. Previously, we only included the first 1-s snapshots with the highest image quality because the spot density was rapidly decreased, as now shown in Fig. S12. In Fig. S12b, the localization number plateaus after ~ 5 mins to roughly half of the initial value. Therefore, even though the spatial or temporal resolution may slightly decrease, reasonable SMLM images could be reconstructed for 15 mins.

In our proposed photoswitching mechanism of UnaG, photo-oxidized bilirubin immediately dissociates from the protein due to insufficient binding affinity (Fig. 3, S1, S3 and S4). In this perspective, we can easily deduce that the photo-oxidized bilirubin does not interfere the binding kinetics of fresh bilirubin regardless of its amounts. To prove that, we mixed a large portion of pre-bleached bilirubin to the fresh bilirubin and observed the on-switching behavior of UnaG in a fixed cell as the reviewer suggested (see Fig. Reviewer 1 below). Here, $0.2 \mu\text{M}$ of fresh bilirubin showed almost identical behavior with or without $0.8 \mu\text{M}$ of pre-bleached bilirubin (green and blue lines). This is consistent to the *in vitro* result in Fig. 2e in the previous revision that damaged/bleached bilirubin induce virtually no fluorescence recovery (purple dashed line).

Fig. Reviewer 1 Fluorescence recovery of UnaG under different buffer compositions. Pre-bleached bilirubin (dmBR) did not significantly affect to the recovery rate regardless of its amount.

It is important to prevent the bleaching of free bilirubin by direct photo-oxidation that results in decreasing the on-switching rate of UnaG proteins. To reduce the chance of photo-absorption of bilirubin, we can introduce total internal reflection illumination, which selectively excites only bilirubin molecules near the glass surface. But, the special illumination geometry will limit the imaging depth. Light source with longer wavelength also would help this problem since free bilirubin showed blue-shifted absorption spectra than holoUnaG [Kumagai et al., Cell, 2013], but this approach should worsen the multicolor imaging capability in terms of crosstalks.

For the live-cell imaging, we intentionally acquired the raw images under "overactivated conditions" to reduce the temporal resolution that plays a crucial role in the studies of live-cell dynamics. The new HAWK analysis helped to reduce the overactivation artifacts and produced well-resolved ER structures in Fig. 5. On the other hand, all the raw images of fixed cells were recorded with much less activated molecules ensuring the single-molecule spot density and the SML images were reconstructed with standard ThunderSTORM analysis without HAWK. Therefore, Fig. 1d, 4e, 4f and S6-8 obtained from fixed cells offer data set for controlled photophysics.

4) Endogenous bilirubin:

- a. "endogenous bilirubin" is not really used for this technique. It is bleached away to get to a single molecule level. Then extra bilirubin is supplied to refill the UnaG. The endogenous level can't provide this.
 - Please change this in the whole text (introduction, discussion), currently the text suggests that this method works easily in live cells with endogenous metabolites and doesn't need any uptake of any component.
 - You can add that the advantage of endogenous bilirubin is that you have a fluorescent structure at the start which you can selected for, focus on, etc., then you bleach this one away and start the SMLM experiment using the newly supplied bilirubin.

Response: We respectively disagree that the endogenous bilirubin is not used for SMLM imaging. The UnaG-bound bilirubin molecules, not free bilirubin molecule, should be bleached to get to a single-molecule level of UnaG. Photooxidation of free, unbound bilirubin in solution can occur with less efficiency than that of UnaG-bound bilirubin since the absorption maximum of bilirubin at 450 nm is blue-shifted by 48 nm from that of UnaG [Kumagai et al., Cell, 2013]. In fact, Fig. 1d(ii) that was obtained without supplementing exogenous bilirubin demonstrates that endogenous level of bilirubin can produce SMLM images albeit the lower image quality. From this observation, we could tell that SMLM is possible with endogenous level of bilirubin without any supplemental bilirubin. To obtain high-quality SML images, we added exogenous bilirubin as a supplement to boost the on-switching rate.

It is noteworthy that this exogenously added bilirubin has very low chemical toxicity to cells because the exogenous bilirubin that we used is chemically identical to endogenous bilirubin, which is naturally generated in the cells. This could provide much benefit compared to other SMLM probes, which require addition of synthetic fluorophores (e.g. PAINT probes or HaloTag/SNAPtag ligands) whose cellular toxicities have not been extensively characterized yet.

To better convey the above-mentioned message, we now have changed the phrase of "(4) fluorescence recovery by binding an endogenous metabolite (i.e. bilirubin)" in page 9 to "(4) fluorescence recovery by binding a natural metabolite (i.e. bilirubin) that can be exogenously supplemented when necessary."

Also, to prevent misinterpretation, we changed the first sentence of the section titled as "Super-resolution imaging of various subcellular structures" to "No fluorescence recovery without external BR of UnaG proteins *in vitro* and in fixed cells indicates that the repetitive binding of BR to the protein mainly causes the reversible photoswitching of UnaG (Fig. 2e and Supplementary Fig. S5)."

We thank the reviewer to point out that UnaG photophysics allows for conventional widefield imaging prior to SML to select for, focus on a view field of interest. Therefore, we added a new sentence in the second paragraph in page 8 that reads "Before SML imaging, we could obtain conventional widefield images with a low excitation intensity."

Final SMLM image quality:

I am not at all convinced by the SMLM image qualities of Figure 4 e and f, Fig. 6 and the supplemental dual and triple color images. Judging these images and the localization precisions given by the authors, this method cannot compete with other current labeling and SMLM imaging methods. Please investigate not only photon numbers but a) reversibility of switching, b) control over photophysics (seems really bad to me), c) labeling coverage, d) false positive rates, false negative rates etc., e.g. using standards as nuclear pore complexes which should give 16 or 32 clear spots. The images of a new technique must be highly convincing and significantly better than images obtained by the current methods in the field (taken from the literature). With such bad SMLM images like presented currently, it is only "yet another technique" that doesn't perform better and thus will not find application. Widths of ER tubules and vimentin in the 50 nm range are too high! If you would like to use cytoskeletal structures (and not nuclear pore complexes), then maybe use microtubules. These should be hollow in the middle, so give a double profile (e.g. see <https://www.nature.com/articles/nmeth.2214>).

Response: We would like to emphasize that UnaG offers the best image quality among fluorescent proteins that are workhorses for live-cell imaging. It is true that our SML images are not as good as other SML images from fixed cells labeled with far-red organic dyes, which provides much more photon numbers. Many of such high-photon conditions

cannot be applied to live cells. For instance, the double microtubule profile in [Vaughan et al, Nat. Methods, 2012] that the reviewer mentioned was obtained from cell-impermeable Cy3B or Alexa Fluor 647 dyes reduced with harsh chemical treatment of NaBH₄ that were imaged for hours to reconstruct one SML image. High-photon far-red dyes can be targeted to live cells with advanced, non-standard labeling method and cannot recover from photobleaching.

Our SML images offer high qualities in the field of live-cell SML imaging utilizing FPs whose photon numbers are comparable to UnaG. In the Fig. Reviewer 2 below, we compare our images to previously reported SML images with FPs (mEos2, mMaple3 and PAmCherry) and Atto 488, which is well known to give the highest photon numbers among blue-absorbing dyes. Please note that even Atto 488 dye could not resolve the hollow structure of the microtubule and could not reconstruct the ring-like shape of a clathrin coated pit perfectly.

Figures below:

Reprinted by permission from Springer Nature: Nature, Nature Methods. A bright and photostable photoconvertible fluorescent protein, McKinney et al., © 2019 Springer Nature Limited (2009)

Wang et al., Characterization and development of photoactivatable fluorescent proteins for single-molecule-based superresolution imaging, PNAS. 111(23):8452-7 (2014)

Reprinted by permission from Springer Nature: Nature, Nature Methods. Photoactivatable mCherry for high-resolution two-color fluorescence microscopy, Subach et al., © 2019 Springer Nature Limited (2009)

Reprinted by permission from Springer Nature: Nature, Nature Methods. Evaluation of fluorophores for optimal performance in localization-based super-resolution imaging,

Dempsey et al., © 2019 Springer Nature Limited (2011)

Fig. Reviewer 2 Examples of SML imaging performances for FPs providing high photon numbers (mEos2, mMaple3, PAmCherry1 and UnaG), and for green organic dye, Atto488.

The reversibility of switching can be found in several places in our manuscript. For example, Fig. 1d compared the switching properties under different conditions in the same field of view, which cannot be achieved with irreversibly switchable fluorophores. Moreover, Fig. S11c also showed the repetitive reversible switching in the live cells.

SML imaging property over controlling the photophysics was also displayed in Fig. 1d. Here we obtained the SML images under different buffer conditions that control the on-switching rate (bilirubin) or the off-switching rate (GLOX).

In terms of labeling coverage, we imaged six targets (ER, mitochondria, vimentin, peroxisome, lamin and clathrin, Fig. 4 and Fig. S6-8) with fixed cells, and showed no notable distortions in the structures of the labeled targets and the morphology of UnaG-expressing cells (Fig. S9a). In a previous study ([Park et al., ACS Chem. Biol., 2016]), UnaG could label 10 protein targets in various sites including NLS (nuclear localization signal), NES (nuclear exclusion signal), IMS (intermembrane space of the mitochondria), and PM (plasma membrane).

False positive and negative rates are important to the final SML image quality. It requires a standard sample that has well-known structure such as NPCs. We tried to image the NPCs with UnaG to get such information, even in the previous revision period, but we failed to label NPCs with UnaG due to poor parent plasmids for NPCs that we obtained from AddGene. Alternatively, we used vimentin filaments and clathrin coated pits as structural standards, as previously used for systematically comparing the SMLM image qualities of various organic dyes [Dempsey et al, Nat. Methods, 2011] and fluorescent proteins [Wang et al, PNAS, 2014].

Additional major comments:

200 ms is “fast switching” (p.9?) – no, image acquisitions in the 20-50 ms range are the state-of-the-art of being “fast” SMLM. Why are the fluorescent spots (see Fig.4 d) asymmetric in x and y (by 10 nm)?

Response: We agree that 200 ms camera frame rate is not the state-of-art. The off-switching rate might be accelerated with higher illumination intensity. Despite the low camera frame rate, the final reconstructed temporal resolution of 1 sec in Fig.5 is quite fast as the reviewer pointed out in the previous review. Following the reviewer’s comment, we modified the sentence as below.

"The genetically incorporated UnaG can be readily applied to live-cell SML imaging and offers fast tracking for the cellular dynamics due to the high photon numbers and easily controllable UV-free kinetics."

Asymmetric resolution may come from not enough number of samples. Now we provide more symmetric resolution from additional experiment that showed ~12 nm of localization precision in both x and y directions (Fig. 4d).

Minor comments:

- Reference 19 is a dye not a FP, I believe the authors would like to change "FPs" into "fluorophores" in their argumentation.
- The authors should exchange BC2 and PYP in their introduction. BC2 is a peptide tag and thus similar to the enzyme tags in the list, PYP is a fluorogenic tag (whereas Bc2 is not).
- Citation of the Minflux paper missing.
- Comment on the bleaching in the 3-color approach.

Response: We thank the reviewer to sharply point out our mistakes from the manuscript. All the concerns are applied in our revised manuscript.

i) We removed Ref. 19 from the mentioned part because we discussed about only FPs in that paragraph (the reference still exists on the list since it is mentioned in another part).

ii) BC2 and PYP (and also their references) are now exchanged to be placed on correct positions.

iii) We also cited MINFLUX.

iv) Possible bleaching problems are mentioned in "*Live-cell multicolor super-resolution imaging*" part.

"Three-color imaging did suffer from bleaching problem, but it was possible to obtain three-color image for highly expressed targets such as the plasma membrane."

Reviewer #3 (Remarks to the Author):

All my concerns have been fully addressed.

Minor comment:

Maybe it is a misunderstanding on my side, but the times for "on" and "off" in figure s11c state "20s" and "5min", respectively. In the corresponding figures s11d and s11e however data is shown for 3s and 100s. This is, at least for me, confusing. The length of the movie of the ER-remodeling is for 5s. If the authors acquired more images showing maxed out imaging time it would be nice to include them. In line with the their statements on the possibility for long imaging made in the toxicity studies.

However, overall I like the manuscript and the improvements in the revision process and see it fit for publications with Nat. comm.

Response: We thank the reviewer very much for the supportive comment on the publication of our revised manuscript. We also addressed newly arisen issues from the reviewer as below, to further reinforce the manuscript.

i) We added more detailed description in the caption of Fig. S11d and e to avoid such confusion.

"**Supplementary Fig. S11** ... **d** Off-switching rates measured in each switching cycle in **c** (displayed first 300 ms of total 20-s of illumination). ... **e** On-switching rates measured in each switching cycle in **c** (displayed first 100 s of total 300-s of recovery). ..."

ii) Now we displayed long-term SML snapshots for >15 min with 1-s temporal resolution, and also added a graph for spot density versus frames to show the long-term imaging capability (Fig. S12).

Reviewers' comments:

Reviewer #2 (Remarks to the Author):

At this stage I would like to ask the editor for his/her decision as the authors and me have very different opinions on the novelty and utility of the UnaG approach as a SMLM labelling technique. I would like to clearly state that, in my opinion, the experimental evidence does not show sufficient SMLM data quality - which is needed for this approach to be a useful tool. But I am happy to accept other opinions – In case the manuscript is accepted, I would like to kindly ask to publish the rebuttal files with the manuscript for full transparency. To give a bit more details why I keep my opinion after the last revision phase:

1. To be honest, this answer to my last review sparks the essence of my criticism: "In fact, Fig. 1d(ii) that was obtained without supplementing exogenous bilirubin demonstrates that endogenous level of bilirubin can produce SMLM images albeit the lower image quality."

For me, this is not an SMLM image at all. Nobody should use this tag for SMLM if this is its intrinsic quality. This image has nothing to do with a SMLM image that actually resolves something. There are more holes in the vimentin structure than blobs which were fitted (which are at the denser areas, a typical fitting result for not well controlled photophysics of the probe). Nobody will profit from an image like this – here people should stay with a good confocal but diffraction-limited image to deduce their biological answers. To be publishable, the technique should provide something new and an application which cannot be done with other methods until now. Which one would this be?

2. UnaG imaging in this manuscript with 200ms integration times is as slow as PAINT. So, please proof experimentally that you are faster (at high quality, thus without "overactivation conditions"). I furthermore don't think that UnaG performs better than PAINT concerning high background. Like in PAINT, in this technique the probes are added. The added metabolites cause background and the technique needs to make use of special imaging geometries such as TIRF. Thus, the criticism towards PAINT should be tuned down or the authors have to proof that they perform better. PAINT manuscripts are in the 3 nm resolution range, UnaG seems to me at least 5 times worse.

3. Better photon numbers than primed conversion: Please proof experimentally that you get more photons and a better resolution in vivo using a reference structure and image it both, with normal FPs, UnaG and primed conversion FPs.

4. I don't see any experimental proof for this sentence "We would like to emphasize that UnaG offers the best image quality among fluorescent proteins that are workhorses for live-cell imaging" in the manuscript. Even if there is a photon numbers measurement, the higher photon counts do not show in the images. So, please proof the utility of the tool on a reference standard in living cells, e.g. nuclear pore, microtubule hollow structure (giving a profile with a dip in the middle), FtnA oligomers, Glycine receptor and show that you obtain better image quality. Just an image of vimentin does not proof anything. There are high quality images on reference standards. Using reference standards is independent of the setup used or the ability of the experimenter – thus please perform this experiment. This is needed to introduce a new tool, e.g. the recent BC2 nanobody paper performed FtnA standard measurements vs. SNAP tag, the just uploaded back-to-back preprints from the Ries and Jungmann groups (<https://www.biorxiv.org/content/10.1101/582668v1>, <https://www.biorxiv.org/content/10.1101/579961v2>) show nuclear pore imaging with protein tags, dSTORM, GFP nanobodies, DNA-PAINT and FPs at very high quality. This is the current standard of the field the authors have to benchmark against, not images from publications more than 5 years ago.

5. I honestly currently see no advantage of UnaG in SMLM imaging quality in comparison to mMaple3 or mEos3.2. Please proof with an in vivo cell reference structure. The only advantage I see is something the authors don't show: All FPs need oxygen for maturation – UnaG works without and could thus be an ideal tag for anaerobic cultures. Thus, an application of this technique for exploring new biology in e.g. anaerobe archaea would be novel and cannot be done with FP tags.

6. I highly appreciate the fixed cell experiment with 0.2 μM of fresh bilirubin and 0.8 μM of pre-bleached bilirubin. I would recommend to add this figure to the SI.

7. This sentence is not experimentally shown, is it? "The genetically incorporated UnaG can be readily applied to live-cell SML imaging and offers fast tracking for the cellular dynamics due to the high photon numbers and easily controllable UV-free kinetics." At least not at the quality I would expect, I know of the 1s overreacting conditions experiment but this is not what I am looking for.

Reviewer's comments:

Reviewer #2 (Remarks to the Author):

At this stage I would like to ask the editor for his/her decision as the authors and me have very different opinions on the novelty and utility of the UnaG approach as a SMLM labelling technique. I would like to clearly state that, in my opinion, the experimental evidence does not show sufficient SMLM data quality - which is needed for this approach to be a useful tool. But I am happy to accept other opinions – In case the manuscript is accepted, I would like to kindly ask to publish the rebuttal files with the manuscript for full transparency. To give a bit more details why I keep my opinion after the last revision phase:

In the first review, the reviewer recognized the benefits of our method and said, “*the labeling system is rather small (about half the size of GFP) and can be ‘refueled’ by the pool of available bilirubin – both are good reasons to check and evaluate it as an interesting and promising SMLM labeling option.*”. This statement indicates that the reviewer acknowledged the novelty and utility of our work in very similar perspectives to ours. In our previous revisions, we may have not succeeded in convincing the reviewer with our experimental results and discussions. In this revision round, we performed additional experiments such as (1) SML imaging on reference structures benchmarking a recent paper [Virant et al., *Nat. Commun.* **9**, 930 (2018)] following the reviewer’s suggestion, (2) direct, quantitative comparisons of UnaG and mMaple3 in labeling coverage and long-term imaging capacity and (3) direct comparisons of background in UnaG and PAINT imaging under Epi and TIR illuminations, (4) SMLM imaging at particular conditions such as endogenous level of bilirubin and non-overactivation conditions, and so on. Needless to say, we would be more than happy to expose all the material including the rebuttals of the previous and current revision rounds.

1. To be honest, this answer to my last review sparks the essence of my criticism: “In fact, Fig. 1d(ii) that was obtained without supplementing exogenous bilirubin demonstrates that endogenous level of bilirubin can produce SMLM images albeit the lower image quality.” For me, this is not an SMLM image at all. Nobody should use this tag for SMLM if this is its intrinsic quality. This image has nothing to do with a SMLM image that actually resolves something.

We apologize for giving an inappropriate description. It is better to call Fig. 1d(ii) as a partial, undersampled copy of an SMLM image in Fig. 1d(iv). The exact same UnaG proteins targeted to the ER in the same cell used for Fig. 1d(ii) produced the SMLM image in Fig. 1d(iv) that resolved the ER tubules. By using Fig. 1d(i)-(iv), we intended to emphasize the consequences of various buffer conditions and to visually guide the readers to the switching mechanism. If we have collected much more frames under the condition for Fig. 1d(ii) (i.e. with endogenous level of bilirubin), we should have been able to collect much more localizations in total albeit at the low on-switching rate limited by the bilirubin concentration. Therefore, if we extend the imaging session for Fig. 1d(ii) sufficiently to accumulate localizations above the density level required for resolving the ER tubules, we should have been able to reconstruct a proper SMLM image as in Fig. 1d(iv).

To best answer the reviewer’s original question of whether the endogenous level of bilirubin can produce SMLM images, we here present the SMLM images of clathrin-coated pits (CCP) without supplementing exogenous bilirubin (Fig. R1). In a fixed cell, we could resolve the ring-like projections of CCPs with endogenous level of bilirubin (Fig. R1a) by reconstructing the SMLM image from 20,000 frames of slowly switching UnaG. In a live cell, we could reconstruct a SMLM image with just 1,000 frames (Fig. R1b). It is because live cells actively retain intracellular bilirubin concentration. In contrast, in fixed cells, cell-permeable bilirubin can be lost during washing steps with bilirubin-free solutions. However, the live-cell SMLM image in Fig. R1b is blurred due to motion and structure evolution in the live cell during the acquisition time of 50 seconds, which was required for reconstructing the SMLM image at the endogenous bilirubin concentration that still limited the on-switching rate. The blurred image is consistent with the previous result [Huang et al., *Nat. Methods* **10**, 653 (2013)]. When necessary to accelerate the acquisition speed, we can simply increase the concentration of bilirubin in the imaging buffer up to micromolar level without increasing the background under Epi illumination as demonstrated in Fig. R3 (for more details about the minimal background with Epi geometry, please see our response to the reviewer’s point #2 below). Indeed, as in our response to the point #7, we performed clathrin imaging in live cells with 1 μ M exogenous bilirubin and obtained 2-sec snapshots that resolved the ring-like projections (Fig. R8). Note that the 2-sec SMLM image was obtained under non-

overactivation condition (i.e. the raw diffraction-limited images of individual fluorophores largely do not overlap and can be fitted with single Gaussians). It was possible to achieve such condition by increasing the light intensity by ten times. Then, the off-switching rate becomes sufficiently high for lowering the on-off duty cycle to the level that ensures only one molecule to turn on at a given frame within a CCP whose dimension is below the diffraction limit.

Fig. R1 SMLM image of clathrin-coated pits (CCP) with UnaG-CLC with endogenous concentration of bilirubin without supplementing bilirubin in the imaging buffer. **a** CCPs in a fixed Cos7 cell whose ring shapes are resolved. **b** CCPs in a live Cos7 cell. The image is reconstructed from 1,000 frames acquired at 50 ms exposure time that took total 50 s. The acquisition time is long enough for the pits to move and to change shape in live cells resulted in blurred SMLM image as previously observed in [Huang et al., *Nat. Methods* **10**, 653 (2013)]. (Adapted by permission from Springer Nature: Nature, Nature Methods. Video-rate nanoscopy using sCMOS camera-specific single-molecule localization algorithms, Huang et al., © 2019 Springer Nature Limited (2013))

There are more holes in the vimentin structure than blobs which were fitted (which are at the denser areas, a typical fitting result for not well controlled photophysics of the probe). Nobody will profit from an image like this – here people should stay with a good confocal but diffraction-limited image to deduce their biological answers. To be publishable, the technique should provide something new and an application which cannot be done with other methods until now. Which one would this be?

We replaced the vimentin SMLM image in Fig. 4e,f with a new image in which thin, peripheral fibrils are better resolved in a similar manner to the recently published BC2 nanobody images of vimentin that the reviewer mentioned in his/her point #3 [Virant et al., *Nat. Commun.* **9**, 930 (2018)]. In Fig. R2, we compare the new vimentin image with UnaG (Figs. R2c) as well as the published results of mMaple3 (Fig. R2b), PAmCherry (Fig. R2a, left), GFP nanobody (Fig. R2a, middle) and BC2 nanobody (Fig. R2a, right) conjugated with Alexa Fluor 647. The vimentin fibrils labeled with PAmCherry were either highly aggregated or sparsely labeled (Fig. R2a, left). The mMaple3 image (Fig. R2b) show more thin fibrils with better labeling coverage than the PAmCherry result [Wang et al., *Proc. Natl. Acad. Sci.* **111**, 8452 (2014)]. For direct comparison of UnaG and mMaple3, we conducted SMLM imaging of mMaple3-labeled vimentin filaments and placed the images with UnaG and mMaple3 side by side in Fig. R6 (see the reviewer's point #4 for more details). Following the method in Virant et al., we also quantified the lengthwise labeling coverage from these images for thin filaments that showed the largest differences in labeling coverage of various labeling methods in Virant et al. (Fig. R6d). The labeling coverage for UnaG (76%) was better than mMaple3 (56%) and comparable to the reported value for BC2 nanobody (~80% for fibers below 75 nm). When comparing the nanobody images (middle and right panels in Fig. R2a) and UnaG image (Fig. R2c), the lower photon number of UnaG than that of Alexa Fluor 647 is reflected in the thicker filament widths. However, UnaG-labeled vimentin fibrils achieved similar level of lengthwise labeling coverage to the BC2 results probably due to UnaG's small size. In the BC2 nanobody paper, the higher labeling coverage of BC2 nanobody than that of GFP nanobody was attributed to the small size of the tag. Likewise, the difference between UnaG and mMaple3 can be attributed to the difference in molecular weight, as we discussed in more detail in response to the point #4. Therefore, the vimentin images serve to illustrate the benefits of UnaG's small size for SMLM that the reviewer recognized in his first review round.

Reproduced from Wang et al., Characterization and development of photoactivatable fluorescent proteins for single-molecule-based superresolution imaging, PNAS. 111(23):8452-7 (2014)

Fig. R2 Comparison of the SMLM images of vimentin labeled with different fluorophores and methods. **a** PAmCherry (left) and nanobodies conjugated with Alexa 647 for GFP (middle) and BC2 tag (right) from [Virant et al., *Nat. Commun.* **9**, 930 (2018)]. **b** mMaple3 from [Wang et al., *Proc. Natl. Acad. Sci.* **111**, 8452 (2014)]; **c** UnaG. The contrasts of the zoom-ins were re-adjusted for better representing the specific views. UnaG provided better labeling coverage compared to other fluorescent proteins including GFP and comparable coverage to that of BC2 nanobody conjugated with Alexa 647 dye, due to the small size.

We'd like to emphasize that UnaG and Alexa Flour 647 have complementary properties. While Alexa Flour 647 offer high photon outputs, live-cell labeling of the organic dye involves with multiple steps including genetic incorporation of a tag and delivery of dye-conjugated tag-binders such as nanobody that are impermeable to cells. For instance, Virant et al. used lipid-based protein transfection protocol to deliver the cell-impermeable nanobodies conjugated with Alexa Flour 647 to intracellular targets inside live cells. In contrast, UnaG labeling follows standard genetic incorporation method of fluorescent proteins and does not require perturbative means for delivery of the ligand. Bilirubin, an endogenous metabolite, is permeable to cell membrane. Thus, by simply adding bilirubin in the medium, we could increase the on-switching rate when necessary. Also, exogenous bilirubin can be omitted when the endogenous level is sufficient, as we demonstrated in Fig. R1. Although the photon output of UnaG is lower than that of the organic dye, it offers the highest photon counts among fluorescent proteins. In addition, UnaG is unique among fluorescent proteins in terms of the small size (about half), the ability to refuel the fluorescence and the narrow spectral window. These benefits are what the reviewer previously recognized and what we again try to provide experimental validations in our responses to the point #5.

To further demonstrate for which cannot be done with conventional fluorescent proteins, we obtained long-term time-lapse movies of UnaG and mMaple3 that are expressed in the cytosol. The time traces of localization number are in Fig. R7 in response to the reviewer's point #5. By the end of the 25-min movie of mMaple3, the localization number diminishes to nearly zero probably because photobleaching permanently damages the chromophore composed of several amino acids of mMaple3. In contrast, UnaG localization number is nearly unchanged within a similar 25-min-

long movie probably due to the replacement of the bleached bilirubin by fresh bilirubin in solution. This comparison is aimed at supporting the reviewer's previous comment in the first review round that recognized the 'refueling' capability, which cannot be done with conventional FPs. The permanent consequence of photobleaching is common in most fluorescent proteins in which the chromophore is covalently bonded to the protein (Fig. S1). In contrast, UnaG is the only FP with purely noncovalent interactions between the chromophore and protein that make it possible to detach the damaged chromophore and to re-bind with a new, fresh one. Inspired by this unique chemical nature of UnaG, we recently published a computational chemistry paper on relating the composition and dynamics of hydrogen-bonding network connecting the chromophore, protein and water molecules to the fluorescence quantum yields of UnaG variants [Lee et al., *Chem. Sci.* **9**, 8325 (2018)].

2. UnaG imaging in this manuscript with 200ms integration times is as slow as PAINT. So, please proof experimentally that you are faster (at high quality, thus without "overactivation conditions"). I furthermore don't think that UnaG performs better than PAINT concerning high background. Like in PAINT, in this technique the probes are added. The added metabolites cause background and the technique needs to make use of special imaging geometries such as TIRF. Thus, the criticism towards PAINT should be tuned down or the authors have to proof that they perform better. PAINT manuscripts are in the 3 nm resolution range, UnaG seems to me at least 5 times worse.

We agree that UnaG's photon number is much lower than PAINT. But, the background from bilirubin is nearly non-existent and does not require special imaging geometries. Bilirubin is fluorogenic: free bilirubin in solution is non-fluorescent due to the free dihedral rotations at the central carbon atoms; non-fluorescent bilirubin becomes fluorescent when binding to UnaG protein that restricts the rotation [Kumagai et al., *Cell* **153**, 1602 (2013)]. We recently published a paper on relating the rotational degree of freedom to the fluorescence quantum yield by using computational chemistry simulations [Lee et al., *Chem. Sci.* **9**, 8325 (2018)]. Owing to the non-fluorescent nature of the ligand, there is no need to implement special illumination geometry such as TIR, which we have never used throughout the current work. All of our images have been obtained with standard Epi illumination. To highlight the benefit, we added Fig. R3 that shows the raw diffraction-limited images of individual molecules of UnaG in which the fluorescence background is virtually unchanged up to 1 micromolar concentration of bilirubin under Epi illumination. In contrast, even 1.0 nM PAINT probe increased the fluorescent background significantly under Epi illumination and required TIR to suppress the background (Fig. R3b,c).

Fig. R3 Comparison of the raw images of DNA-PAINT and UnaG SMLM images under different illumination geometries. **a** In UnaG SMLM imaging, in which the unbound ligand of bilirubin is non-fluorescent, the background under Epi illumination geometry were virtually unchanged up to 1,000 nM concentration of exogenously supplemented bilirubin. **b** In PAINT imaging, even 1.0 nM elevated the background significantly in Epi illumination. **c** Total internal reflection illumination (TIR) was required to decrease the fluorescence background of 1.0 nM PAINT probes.

From the initial submission to the current revision rounds, all of our UnaG SMLM images in the manuscript were obtained with standard Epi illumination. We apologize that our method section in the manuscript did not clearly state this fact. Now, we included additional descriptions on the non-fluorescent nature of free bilirubin and the illumination geometry in the manuscript and placed Fig. R3 as Fig. S13 along with a brief discussion to prevent the readers from

making false assumption as the reviewer did. We also added this unique benefit to the list of UnaG's advantages, which now reads:

“UnaG offers the unique combination of following advantages simultaneously: (1) simple labeling method of standard genetic incorporation; (2) high photon numbers comparable to EosFP and Atto 488; (3) reversible switching between the dark and green fluorescence states; (4) fluorescence recovery by binding a natural metabolite (i.e. bilirubin) that can be exogenously supplemented when necessary. (5) little fluorescent background under Epi illumination at micromolar ligand concentration that acts as pseudo-unlimited reservoir (Supplementary Fig. S14).”

Thanks to the low background (Fig. R3) and easily controllable kinetics (Fig. 2), UnaG imaging can be faster than PAINT. In Fig. R8 in our response to the point #7, we obtained 2-sec SMLM image with 10-ms exposure time without overactivation condition by increasing the 488-nm intensity by ten times. The 10-fold increase in light intensity linearly accelerated the off-switching rate as we demonstrated in Fig. 2b. The increased off-switching rate allowed us to use the short exposure time of 10 ms and to avoid overactivation even when using 1 μ M exogenous bilirubin by lowering the on-off duty cycle. Again, the non-fluorescent nature of bilirubin allowed us to use the high probe concentration without worrying about the background even in Epi illumination. In contrast, PAINT imaging speed is limited by the chemical nature of the probe. The unbinding rate limits the exposure time in PAINT imaging in the range of hundreds of milliseconds, in general. It may be possible to increase the unbinding rate by changing the chemical nature of the probe, but the chemical change can deteriorate the binding affinity because the dissociation constant (i.e. the equilibrium constant of the dissociation reaction) is basically the unbinding rate constant divided by the binding rate constant. Also, background from fluorescent probes in solution restricts the on-switching rate and the probe concentration is usually in nanomolar level in PAINT.

3. Better photon numbers than primed conversion: Please proof experimentally that you get more photons and a better resolution in vivo using a reference structure and image it both, with normal FPs, UnaG and primed conversion FPs.

Previously, we reported \sim 1,000 photons for mMaple3 and \sim 1,200 photons for UnaG in Fig. 4a-c. Among primed conversion FPs, mMaple3 offers the largest number of photons, 180 photons under primed conversion (Fig. R4a) [Turkowyd et al., *Angew. Chem. Int. Ed.* **56**, 11634 (2017)]. Also, the measured fluorescence intensity of mMaple3 under primed conversion is at least 5-times lower than conventional conversion mediated by 405 nm at the physiological conditions, even when the primed conversion rate is faster than UV-activation rate (Turkowyd et al., Fig. R4b), indicating single-molecule photon number notably decreased with primed conversion. mEos2 also showed lower photon numbers under primed conversion than UV-activation (Fig. R4c from [Mohr et al., *Angew. Chem. Int. Ed.* **56**, 11628 (2017)]). Since each comparison was performed under identical experimental settings, it is rational to conclude that mMaple3 provides lower photons under primed conversion than UnaG and/or 405-activated mMaple3, by accepting previously published results of three data sets from two independent papers. The photon number comparisons of UnaG and mMaple3 in Fig. 4c and SMLM images of vimentin in Fig. R6 would serve as the experimental proofs for UnaG and 405-activated mMaple3.

Figure below:

Panels A and B reproduced with permissions from Turkowyd et al., A General Mechanism of Photoconversion of Green-to-Red Fluorescent Proteins Based on Blue and Infrared Light Reduces Phototoxicity in Live-Cell Single-Molecule Imaging, *Angew. Chem. Int. Ed.* (John Wiley and Sons). 56(38):11634-11639 (2017) Copyright © 1999-2019 John Wiley & Sons, Inc. All rights reserved

Panel C reproduced with permissions from Mohr et al., Rational Engineering of Photoconvertible Fluorescent Proteins for Dual-Color Fluorescence Nanoscopy Enabled by a Triplet-State Mechanism of Primed Conversion, *Angew. Chem. Int. Ed.* (John Wiley and Sons). 56(38):11628-11633 (2017) Copyright © 1999-2019 John Wiley & Sons, Inc. All rights reserved

Fig. R4 Comparison of the brightness of primed conversion FPs. **a** Single-molecule photon numbers of fluorescent proteins under primed conversion. The figure was adopted from Supplementary Figure 8 of [Turkowyd et al., *Angew. Chem. Int. Ed.* **56**, 11634 (2017)]. **b** Relative brightness (red) of 405-mediated mMaple3 (upper panel) and primed converting mMaple3 (lower panel). The figure was adopted from Supplementary Figure 3c(iii) of Turkowyd et al. again. **c** Single-molecule photon numbers

of mEos2 under 405-nm-mediated activation and primed conversion. Primed conversion mEos2 showed lower photon counts than 405-activated mEos2. The figure was adopted from Figure 3c of [Mohr et al., *Angew. Chem. Int. Ed.* **56**, 11628 (2017)].

In addition to the photon numbers, UnaG offers several benefits compared to mMaple3, one of the brightest primed conversion FPs (pcFPs), as we emphasize through all the manuscript and response letter, in terms of small size (about half of GFP) that leads into higher labeling coverage (Fig. R6), refueling property for long-term imaging (Fig. R7), dark-to-green transition for streamlined, high-performance multicolor live-cell imaging (Figs.6, S18 and S20). All pcFPs, including mMaple3, share common molecular features such as molecular weight comparable to GFP, covalently linked chromophores, and green-to-red transition. Therefore, many of our experimental comparisons of mMaple3's performance in SMLM in comparison to UnaG intrinsically can be applied to other pcFPs. Also, we can deduce other pcFPs' properties by referencing the mMaple3 results. For instance, Dendra2 is the most commonly used pcFP for SMLM, but some of its SMLM performance such as labeling efficiency [Wang et al., *Proc. Natl. Acad. Sci.* **111**, 8452 (2014)] and photon numbers under primed conversion [Turkowsky et al., *Angew. Chem. Int. Ed.* **56**, 11634 (2017)] are reported as inferior to mMaple3. Thus, we can deduce Dendra2's labeling efficiency and photon numbers to be lower than UnaG.

Nonetheless, we found that the broad spectral window of pcFP (green-to-red emission as well as dual conversion lights in blue and red/infrared) is better to highlight than the low photons or other disadvantages because it makes pcFPs incapable of multicolor imaging modalities that UnaG can easily achieve. The pre-conversion emission in green overlaps with dark-to-green FPs and the post-conversion emission in red overlaps with dark-to-red FPs. By sequential primed conversion followed by UV-activation, a pcFP can be sequentially distinguished from a photoactivatable FP that cannot be activated by primed conversion such as PAmCherry [Mohr et al., *Angew. Chem. Int. Ed.* **56**, 11628 (2017); Virant et al., *Int. J. Mol. Sci.* **18**, 1524 (2017)]. Also, red PAINT probes were distinguished sequentially after imaging and bleaching pcFPs in Virant et al. However, these approaches of sequential multicolor imaging can neither observe different structures in a live cell at the same time nor collect a time-lapse movie composed of multiple SMLM frames in multiple color channels.

Far-red dyes may allow for simultaneous SMLM imaging with pcFPs, albeit the complication in labeling as we discussed in the point #1. However, the red excitation laser for far-red dyes can serve as efficient light source of primed conversion [Klementieva et al., *Chem. Commun.* **52**, 13144 (2016)]. Klementieva et al. used 250 W/cm² of 640 nm for SMLM imaging of pcFPs, which is 1-2 orders of magnitude lower than the excitation intensity for live-cell SMLM imaging of Alexa Four 647. Therefore, the red laser for exciting a far-red dye can induce primed conversion of pcFPs at a significantly higher rate than needed for proper SMLM imaging, and therefore overly activate pcFPs. Or, the red laser for proper primed conversion rate can be insufficient for high-performance SMLM imaging of far-red dyes. Overall, it is highly challenging to find laser intensities, particularly for red laser, that give optimal on-switching rate of a pcFP and optimal off-switching rate of a far-red dye at the same time. Probably because of this reason, simultaneous multicolor imaging of pcFPs has not been demonstrated yet, to the best of our knowledge.

In contrast, we demonstrated simultaneous multicolor imaging capacities of UnaG with simple filter-based approach, by utilizing the background-free dark-to-green transition of UnaG. In our previous revision, we demonstrated (1) two-color imaging with bright red membrane probes (Fig. 6), (2) FP-only two-color imaging with PAmCherry (Fig. S18b), (3) three-color SMLM imaging with Alexa Flour 647 (Fig. S18d), (4) simultaneous observation of structural evolution of mitochondrial matrix and inner membrane during division and fusion (Fig. 6e,f). All of these simultaneous multicolor imaging applications are near impossible for pcFPs to carry out due to the broad emission window in green-to-red and multiple conversion light sources in blue and red/infrared. Moreover, Virant et al. used 4 kW/cm² of 488-nm light for bleaching the residual Dendra2, which is ten times more intense than the excitation intensity of UnaG imaging. As a result, primed conversion multicolor SMLM can be more toxic to cells than UnaG imaging.

Finally, to highlight the more prominent advantage of UnaG that we already demonstrated experimentally in various combinations of colors, probes and applications in Fig. 6, S18 and S20, we deleted the phrase of mentioning low photons as major disadvantage of pcFPs in the main text and instead inserted a sentence that discusses the broad spectral window and the limit in multicolor capability that reads, "*primed conversion FPs convert the emission from green to red upon exposure to blue and red/infrared lasers, making simultaneous multicolor imaging near impossible*".

4. I don't see any experimental proof for this sentence "We would like to emphasize that UnaG offers the best image quality among fluorescent proteins that are workhorses for live-cell imaging" in the manuscript. Even if there is a

photon numbers measurement, the higher photon counts do not show in the images. So, please proof the utility of the tool on a reference standard in living cells, e.g. nuclear pore, microtubule hollow structure (giving a profile with a dip in the middle), FtnA oligomers, Glycine receptor and show that you obtain better image quality. Just an image of vimentin does not proof anything. There are high quality images on reference standards. Using reference standards is independent of the setup used or the ability of the experimenter – thus please perform this experiment. This is needed to introduce a new tool, e.g. the recent BC2 nanobody paper performed FtnA standard measurements vs. SNAP tag, the just uploaded back-to-back preprints from the Ries and Jungmann groups (<https://www.biorxiv.org/content/10.1101/582668v1>, <https://www.biorxiv.org/content/10.1101/579961v2>) show nuclear pore imaging with protein tags, dSTORM, GFP nanobodies, DNA-PAINT and FPs at very high quality. This is the current standard of the field the authors have to benchmark against, not images from publications more than 5 years ago.

Among the reviewer's suggested experiments, we benchmarked the BC2 nanobody paper [Virant et al., *Nat. Commun.* **9**, 930 (2018)] since it is published recently in the same journal. Following the reviewer's suggestion, we generated UnaG-FtnA and performed measurement of UnaG-FtnA along with SNAP-FtnA (Fig. R5). For SNAP-FtnA, we obtained 72.5% of completeness of labeling that is consistent to the published value in [Finan et al., *Angew. Chem. Int. Ed.* **54**, 12049 (2015)]. UnaG-FtnA measurement yielded 94.6% of completeness of labeling, consistent to near complete labeling of fluorescent proteins as shown for YFP-FtnA in Finan et al. This is consistent to the high vimentin labeling coverage in Fig. R6. Note that the completeness of labeling for Alexa Flour 647 when using SNAP-tag or BC2 nanobody is lower than fluorescent proteins such as YFP and UnaG probably because of incomplete conjugation reaction of the ligand or incomplete binding of the nanobody to the tag. UnaG's near complete labeling is likely due to the high binding affinity of bilirubin to UnaG ($K_d = 98$ pM). Therefore, despite the lower photon count of UnaG, UnaG has advantage in labeling efficiency in SMLM. We now added the FtnA result as Fig. S12 in the supplementary material.

Fig. R5 Completeness of labeling for UnaG and SNAP-tag labeling systems using FtnA-oligomers of 24 subunits. Mean values of FtnA-24mer fluorescence intensities are presented as percentage of theoretical estimation. **a** Representative image of FtnA-24mer cluster labeled with UnaG. **b** Single-molecule photon number distribution of UnaG in measurement conditions, presented with the mean value. **c** Intensity distribution of FtnA-UnaG clusters. The mean value was used to calculate the completeness of labeling in **g**. **d-f** Analogous data set with **a-c** for SNAP-tag labeling system. **g** Measured completeness of labeling of UnaG and SNAP-tag that yielded 94.6% and 72.5% for UnaG and SNAP-tag, respectively.

In addition, the BC2 nanobody paper used vimentin as a major reference target. Fig. 1,2 of the paper among the total four images in the main text used vimentin labeled with various methods including PAmCherry, GFP and BC2 nanobodies. Therefore, vimentin SML images also serve as good benchmark for us. As we discussed in detail in the point #2 and #5, the labeling coverage of UnaG-labeled vimentin was better than mMaple3 and comparable to the BC2 nanobody results, indicating that the small size of UnaG helps to improve SMLM image quality in terms of labeling coverage.

5. I honestly currently see no advantage of UnaG in SMLM imaging quality in comparison to mMaple3 or mEos3.2. Please proof with an in vivo cell reference structure. The only advantage I see is something the authors don't show: All FPs need oxygen for maturation – UnaG works without and could thus be an ideal tag for anaerobic cultures. Thus, an application of this technique for exploring new biology in e.g. anaerobe archaea would be novel and cannot be done with FP tags.

To directly compare UnaG and mMaple3, we received the mMaple3-Vim plasmids from the authors of [Wang et al., *Proc. Natl. Acad. Sci.* **111**, 8452 (2014)] and performed mMaple3 imaging of vimentin structures (Fig. R6). Also, the image published in Wang et al. is included in Fig. R2b, next to the new UnaG image of vimentin in Fig. R2c. Following the quantitative comparison of SMLM images of vimentin labeled with various tags and fluorophores in [Virant et al., *Nat. Commun.* **9**, 930 (2018)], we measured the lengthwise labeling coverages of thin filaments in SMLM images obtained with UnaG and mMaple3. The average coverage was 76% for UnaG and 56% for mMaple3. The qualitative difference of more continuous fibrils in the UnaG image (Fig. R6a,b) is consistent to the quantitative difference in the values of labeling coverage (Fig. R6c). As we discussed in detail in the point #2 along with additional comparison to PAmCherry and BC2 nanobody, UnaG-labeled vimentin showed higher labeling coverage than mMaple3 probably due to the smaller size. Note that Wang et al. reported the labeling efficiency of mEos3.2 to be much lower than that of mMaple3. We added the labeling coverage comparison of mMaple3 and UnaG as Fig. S11 in the supplementary material.

Fig. R6 Direct comparison of SMLM images of vimentin fibers fused with UnaG and mMaple3 that are imaged in our laboratory. **a** SMLM images of vimentin labeled UnaG and mMaple3. Note that the two images were analyzed and rendered in the exact same way. Specifically, we rendered each image with normalized Gaussian with 10-nm pixel size in ThunderSTORM. **b** Traced and straightened vimentin fibers labeled with UnaG (top) and mMaple3 (bottom), obtained from **a** by using manual segmentation and straighten selection tools in ImageJ. **c** Lengthwise labeling coverage histograms obtained from 88 fibrils for UnaG and 52 fibrils for mMaple3 as exemplified in **b**. We followed the analysis method in [Virant et al., *Nat. Commun.* **9**, 930 (2018)]. Briefly, we manually traced thin, individual filaments and straightened the segmented images using ImageJ. Then, a custom MatLab code converted a straightened image into a binary image and calculated the fraction of covered area of the middle 3 pixels.

As an additional benefit of UnaG to mMaple3, we performed a 25-min-long time-lapse imaging of cytosolic UnaG and mMaple3 in Fig. R7. The UnaG localization numbers were virtually unchanged for more than 25 min probably because of the reversible fluorescence recovery of UnaG by the near infinite pool of free, non-fluorescent ligands at high concentration of 1 μ M (blue line in Fig. R7). In contrast, mMaple3 localizations were exponentially (red line in Fig. R7) decreased possibly due to irreversible photobleaching of the chromophore that is covalently attached to the protein (Fig. S1). The mechanism of reversible switching of UnaG is quite unique because it stems from noncovalent, highly specific binding of the ligand to the protein (Fig. S1). In conventional FPs in SMLM including mMaple3, the photoconversion or photoswitching occur through the chemical change of the chromophore such as photo-isomerization or photo-oxidation. In fact, UnaG is the first and only FP with noncovalently bound chromophore [Kumagai et al., *Cell* **153**, 1602 (2013); Lee et al., *Chem. Sci.* **9**, 8325 (2018)]. Purely noncovalent interactions between the chromophore and the protein along with the use of highly concentrated pool of nonfluorescent ligands without complication of delivery or notable cell toxicity, both of which are quite unique for UnaG, contributed to negate the effect of photobleaching in long-term SMLM imaging as in Fig. R7. We include the time traces of localization numbers in the new Supplementary Fig. S14.

Fig. R7 Number of switching events (i.e. localizations) of UnaG and mMaple3 in extended periods of SML imaging. Localization numbers of cytosolic UnaG (blue) and mMaple3 (red) expressed in fixed Cos7 cells imaged at an exposure time of 40 ms. The UnaG samples were supplemented with 1 μM bilirubin in solution. While the switching events of mMaple3 diminished due to irreversible photobleaching, UnaG continued to produce switched-on spots thanks to the fluorescence recovery by the unbound bilirubin ligands in solution.

Another benefit to point out is that UnaG allow for crosstalk-free, high-resolution, three-color imaging in live cells. We experimentally demonstrated simultaneous three-color live-cell imaging in Fig. S18d by combining UnaG, MitoTracker Red and Alexa Fluor 647, each of which offer the most photon outputs in its color panel of green, red or far-red. It was possible because UnaG fluorescence switch between the nonfluorescent dark state and green fluorescent state while giving out the largest photon outputs among green switchable fluorophores. This application would be impossible with mMaple3 or mEos3.2 that photoconvert from green to red, covering two channels among the three color windows.

In summary, we experimentally showed UnaG's benefits in SMLM imaging in comparison to mMaple3 or mEos3.2. as follows. (1) The high labeling coverage of vimentin SMLM image would have come from the small size (Fig. R6). (2) During a long-term imaging session in which mMaple3 localization numbers diminished significantly due to the irreversible photobleaching, UnaG localization number stayed virtually unchanged owing to the reversible fluorescence and the nonfluorescent ligand (Fig. R7). (3) Simultaneous three-color live-cell imaging with high photons and little crosstalk is possible with UnaG (Fig. S18d), but not with mMaple3 or mEos3.2.

We appreciate the reviewer's suggestion for an application to anaerobic cultures. This would indeed highlight that UnaG's fluorescence does not rely on oxidation of amino acids constituting the chromophore that occurs in most fluorescent proteins. Also, the suggested application would demonstrate potential for opening a window for new biology. However, we intend to highlight UnaG's benefits for more general applications for appealing to more general public. As we listed above, there are a number of advantages of UnaG that would be useful for standard SMLM imaging, in particular for live-cell and multicolor applications.

6. I highly appreciate the fixed cell experiment with 0.2 μM of fresh bilirubin and 0.8 μM of pre-bleached bilirubin. I would recommend to add this figure to the SI.

We appreciate the reviewer's suggestion on the experiment that helped to eliminate the possibility of competitive inhibition by bleached bilirubin. We added the result in SI (Fig. S3) as the reviewer suggested.

7. This sentence is not experimentally shown, is it? "The genetically incorporated UnaG can be readily applied to live-cell SML imaging and offers fast tracking for the cellular dynamics due to the high photon numbers and easily controllable UV-free kinetics." At least not at the quality I would expect, I know of the 1s overreacting conditions experiment but this is not what I am looking for.

As an experimental proof, we obtained 2-sec snapshots of UnaG-labeled clathrin-coated pits as shown in Fig. R8 under non-overactivation condition. To accelerate the SMLM imaging speed, we increased the excitation intensity by ten times for promoting the off-switching rate fit for 10-ms exposure time while using 1 μM bilirubin for high on-switching rate. The 10-fold increase in excitation also helped to avoid overactivation (i.e. overlaps of single-molecule images in a camera frame). Since the on-off duty cycle can be represented as $1/[1+k_{\text{off}}/k_{\text{on}}]$, the 10-fold higher laser intensity reduced the on-off duty cycle by ten times and therefore the probability of image overlaps by ten times. As a consequence, we could resolve the ring-like projections of clathrin-coated pits with single-emitter fitting algorithm (Fig. R8).

Fig. R8 2-sec snapshot of UnaG-labeled clathrin-coated pits in live cell without using overactivation condition. The SMLM image was reconstructed from 200 frames at 10 ms exposure time that are analyzed with standard single-emitter fitting algorithm without further processing.

However, we would like to point that overactivation condition can be useful in live-cell imaging because the imaging speed can be accelerated without increasing the light dose that can induce cell stress. Indeed, “overactivation” was used for massively accelerating the imaging speed by using many state-of-the-art analysis software packages including deep learning algorithms [Marsh et al., *Nat. Methods* **15**, 689 (2018); Ouyang et al., *Nat. Biotech.* **36**, 460 (2018)]. These algorithms successfully demonstrated to reconstruct proper SML images from raw data containing highly overlapped single-molecule fluorescence images, and therefore allowed to significantly reduce the number of camera frames per SML image. Note that PAINT would face the challenge of high background to achieve overactivation condition because high concentration of PAINT probe for increasing the on-switching rate would result in high background as demonstrated in Fig. R2c. Even with TIR illumination, the exponentially decreasing evanescent field can still excite a small fraction of fluorescent probes in solution. This resulted in a noticeable background in Fig. R2c with 1 nM PAINT probes under TIR excitation. In contrast, we could use 1 μ M bilirubin and Epi illumination to achieve background-free overactivated condition for reconstructing 1-sec SML snapshots of ER in Fig. 5 with HAWK, a recently published and publicly available software for artifact-free high-density localization analysis [Marsh et al, *Nat. Methods* **15**, 689 (2018)].

Comments to the authors 3rd revision (in green):

At this stage I would like to ask the editor for his/her decision as the authors and me have very different opinions on the novelty and utility of the UnaG approach as a SMLM labelling technique. I would like to clearly state that, in my opinion, the experimental evidence does not show sufficient SMLM data quality - which is needed for this approach to be a useful tool. But I am happy to accept other opinions – In case the manuscript is accepted, I would like to kindly ask to publish the rebuttal files with the manuscript for full transparency. To give a bit more details why I keep my opinion after the last revision phase:

In the first review, the reviewer recognized the benefits of our method and said, “the labeling system is rather small (about half the size of GFP) and can be ‘refueled’ by the pool of available bilirubin – both are good reasons to check and evaluate it as an interesting and promising SMLM labeling option.”. This statement indicates that the reviewer acknowledged the novelty and utility of our work in very similar perspectives to ours. In our previous revisions, we may have not succeeded in convincing the reviewer with our experimental results and discussions. In this revision round, we performed additional experiments such as (1) SMLM imaging on reference structures benchmarking a recent paper [Virant et al., *Nat. Commun.* **9**, 930 (2018)] following the reviewer’s suggestion, (2) direct, quantitative comparisons of UnaG and mMaple3 in labeling coverage and long-term imaging capacity and (3) direct comparisons of background in UnaG and PAINT imaging under Epi and TIR illuminations, (4) SMLM imaging at particular conditions such as endogenous level of bilirubin and non-overactivation conditions, and so on. Needless to say, we would be more than happy to expose all the material including the rebuttals of the previous and current revision rounds.

I thank the authors for all the additional experiments done and their consent to publish the review and rebuttal materials. Comments are below.

1. To be honest, this answer to my last review sparks the essence of my criticism: “In fact, Fig. 1d(ii) that was obtained without supplementing exogenous bilirubin demonstrates that endogenous level of bilirubin can produce SMLM images albeit the lower image quality.” For me, this is not an SMLM image at all. Nobody should use this tag for SMLM if this is its intrinsic quality. This image has nothing to do with a SMLM image that actually resolves something.

We apologize for giving an inappropriate description. It is better to call Fig. 1d(ii) as a partial, undersampled copy of an SMLM image in Fig. 1d(iv). The exact same UnaG proteins targeted to the ER in the same cell used for Fig. 1d(ii) produced the SMLM image in Fig. 1d(iv) that resolved the ER tubules. By using Fig. 1d(i)-(iv), we intended to emphasize the consequences of various buffer conditions and to visually guide the readers to the switching mechanism. If we have collected much more frames under the condition for Fig. 1d(ii) (i.e. with endogenous level of bilirubin), we should have been able to collect much more localizations in total albeit at the low on-switching rate limited by the bilirubin concentration. Therefore, if we extend the imaging session for Fig. 1d(ii) sufficiently to accumulate localizations above the density level required for resolving the ER tubules, we should have been able to reconstruct a proper SMLM image as in Fig. 1d(iv).

I can follow the reasoning of the authors. Nevertheless, their argumentation remains hypothetical. To prove this, they would have to record ER tubules and not CCPs (as shown below) using only endogenous levels of bilirubin. CCPs should be easier to resolve than ER as they require less staining (less abundant and smaller structure). Thus, for resolving CCPs, most likely less bilirubin is needed than for SMLM images of the ER at high quality. In my opinion, the authors should stay with their initial biological system (the ER) and show that the endogenous levels of bilirubin are sufficient for this one to obtain high quality SMLM images. Otherwise, I would recommend to change the text and would not call Fig. 1d(ii) a super-resolved image and would not state that this is a technique that generally works without external supply.

To best answer the reviewer’s original question of whether the endogenous level of bilirubin can produce SMLM images, we here present the SMLM images of clathrin-coated pits (CCP) without supplementing exogenous bilirubin (Fig. R1). In a fixed cell, we could resolve the ring-like projections of CCPs with endogenous level of bilirubin (Fig. R1a) by reconstructing the SMLM image from 20,000 frames of slowly switching UnaG. In a live cell, we could reconstruct a SMLM image with just 1,000 frames (Fig. R1b). It is because live cells actively retain intracellular bilirubin concentration. In contrast, in fixed cells, cell-permeable bilirubin can be lost during washing steps with bilirubin-free solutions. However, the live-cell SMLM image in Fig. R1b is blurred due to motion and structure evolution in the live cell during the acquisition time of 50 seconds, which was required for reconstructing the SMLM image at the endogenous bilirubin concentration

that still limited the on-switching rate. The blurred image is consistent with the previous result [Huang et al., *Nat. Methods* **10**, 653 (2013)]. When necessary to accelerate the acquisition speed, we can simply increase the concentration of bilirubin in the imaging buffer up to micromolar level without increasing the background under Epi illumination as demonstrated in Fig. R3 (for more details about the minimal background with Epi geometry, please see our response to the reviewer's point #2 below). Indeed, as in our response to the point #7, we performed clathrin imaging in live cells with 1 μM exogenous bilirubin and obtained 2-sec snapshots that resolved the ring-like projections (Fig. R8). Note that the 2-sec SMLM image was obtained under non-overactivation condition (i.e. the raw diffraction-limited images of individual fluorophores largely do not overlap and can be fitted with single Gaussians). It was possible to achieve such condition by increasing the light intensity by ten times. Then, the off-switching rate becomes sufficiently high for lowering the on-off duty cycle to the level that ensures only one molecule to turn on at a given frame within a CCP whose dimension is below the diffraction limit.

Fig. R1 SMLM image of clathrin-coated pits (CCP) with UnaG-CLC with endogenous concentration of bilirubin without supplementing bilirubin in the imaging buffer. **a** CCPs in a fixed Cos7 cell whose ring shapes are resolved. **b** CCPs in a live Cos7 cell. The image is reconstructed from 1,000 frames acquired at 50 ms exposure time that took total 50 s. The acquisition time is long enough for the pits to move and to change shape in live cells resulted in blurred SMLM image as previously observed in [Huang et al., *Nat. Methods* **10**, 653 (2013)]. (Adapted by permission from Springer Nature: Nature, Nature Methods. Video-rate nanoscopy using sCMOS camera-specific single-molecule localization algorithms, Huang et al., © 2019 Springer Nature Limited (2013))

I would like to thank the authors for their extra work. I would further ask the authors to include the live cell image of CCPs into the manuscript (in the supplemental I did find some CCPs data, but only fixed as far as I understand?). To show that the blurriness of b) is indeed motion blur, I would like to request that they – instead of a b/w image – color-code the localizations in the image in time (which e.g. should yield a clearly visible “motion-rainbow” per CCP).

There are more holes in the vimentin structure than blobs which were fitted (which are at the denser areas, a typical fitting result for not well controlled photophysics of the probe). Nobody will profit from an image like this – here people should stay with a good confocal but diffraction-limited image to deduce their biological answers. To be publishable, the technique should provide something new and an application which cannot be done with other methods until now. Which one would this be?

We replaced the vimentin SMLM image in Fig. 4e,f with a new image in which thin, peripheral fibrils are better resolved in a similar manner to the recently published BC2 nanobody images of vimentin that the reviewer mentioned in his/her point #3 [Virant et al., *Nat. Commun.* **9**, 930 (2018)]. In Fig. R2, we compare the new vimentin image with UnaG (Figs. R2c) as well as the published results of mMaple3 (Fig. R2b), PAmCherry (Fig. R2a, left), GFP nanobody (Fig. R2a, middle) and BC2 nanobody (Fig. R2a, right) conjugated with Alexa Fluor 647. The vimentin fibrils labeled with PAmCherry were either highly aggregated or sparsely labeled (Fig. R2a, left). The mMaple3 image (Fig. R2b) show more thin fibrils with better labeling coverage than the PAmCherry result [Wang et al., *Proc. Natl. Acad. Sci.* **111**, 8452 (2014)]. For direct comparison of UnaG and mMaple3, we conducted SMLM imaging of mMaple3-labeled vimentin filaments and placed the images with UnaG and mMaple3 side by side in Fig. R6 (see the reviewer's point #4 for more details). Following the method in Virant et al., we also quantified the lengthwise labeling coverage from these images for thin filaments that showed the largest differences in labeling coverage of various labeling methods in Virant et al. (Fig. R6d). The labeling coverage for UnaG (76%) was better than mMaple3 (56%) and comparable to the reported value for BC2 nanobody (~80% for fibers below 75 nm). When comparing the nanobody images (middle and right panels in Fig. R2a) and UnaG image (Fig. R2c), the lower photon number of UnaG than that of Alexa Fluor 647 is reflected in the thicker filament widths. However, UnaG-labeled vimentin fibrils achieved similar level of lengthwise labeling coverage to the BC2 results probably due to UnaG's small size. In the BC2 nanobody paper, the higher labeling coverage of BC2 nanobody than that of GFP nanobody was attributed to the small size of the tag. Likewise, the difference between UnaG and mMaple3 can be attributed to the difference in molecular weight, as we discussed in more detail in response to the

point #4. Therefore, the vimentin images serve to illustrate the benefits of UnaG's small size for SMLM that the reviewer recognized in his first review round.

Reproduced from Wang et al., Characterization and development of photoactivatable fluorescent proteins for single-molecule-based superresolution imaging, PNAS. 111(23):8452-7 (2014)

Fig. R2 Comparison of the SMLM images of vimentin labeled with different fluorophores and methods. **a** PAmCherry (left) and nanobodies conjugated with Alexa 647 for GFP (middle) and BC2 tag (right) from [Virant et al., *Nat. Commun.* **9**, 930 (2018)]. **b** mMaple3 from [Wang et al., *Proc. Natl. Acad. Sci.* **111**, 8452 (2014)]; **c** UnaG. The contrasts of the zoom-ins were re-adjusted for better representing the specific views. UnaG provided better labeling coverage compared to other fluorescent proteins including GFP and comparable coverage to that of BC2 nanobody conjugated with Alexa 647 dye, due to the small size.

We'd like to emphasize that UnaG and Alexa Flour 647 have complementary properties. While Alexa Flour 647 offer high photon outputs, live-cell labeling of the organic dye involves with multiple steps including genetic incorporation of a tag and delivery of dye-conjugated tag-binders such as nanobody that are impermeable to cells. For instance, Virant et al. used lipid-based protein transfection protocol to deliver the cell-impermeable nanobodies conjugated with Alexa Flour 647 to intracellular targets inside live cells. In contrast, UnaG labeling follows standard genetic incorporation method of fluorescent proteins and does not require perturbative means for delivery of the ligand. Bilirubin, an endogenous metabolite, is permeable to cell membrane. Thus, by simply adding bilirubin in the medium, we could increase the on-switching rate when necessary. Also, exogenous bilirubin can be omitted when the endogenous level is sufficient, as we demonstrated in Fig. R1. Although the photon output of UnaG is lower than that of the organic dye, it offers the highest photon counts among fluorescent proteins. In addition, UnaG is unique among fluorescent proteins in terms of the small size (about half), the ability to refuel the fluorescence and the narrow spectral window. These benefits are what the reviewer previously recognized and what we again try to provide experimental validations in our responses to the point #5.

To further demonstrate for which cannot be done with conventional fluorescent proteins, we obtained long-term time-lapse movies of UnaG and mMaple3 that are expressed in the cytosol. The time traces of localization number are in Fig.

R7 in response to the reviewer's point #5. By the end of the 25-min movie of mMaple3, the localization number diminishes to nearly zero probably because photobleaching permanently damages the chromophore composed of several amino acids of mMaple3. In contrast, UnaG localization number is nearly unchanged within a similar 25-min-long movie probably due to the replacement of the bleached bilirubin by fresh bilirubin in solution. This comparison is aimed at supporting the reviewer's previous comment in the first review round that recognized the 'refueling' capability, which cannot be done with conventional FPs. The permanent consequence of photobleaching is common in most fluorescent proteins in which the chromophore is covalently bonded to the protein (Fig. S1). In contrast, UnaG is the only FP with purely noncovalent interactions between the chromophore and protein that make it possible to detach the damaged chromophore and to re-bind with a new, fresh one. Inspired by this unique chemical nature of UnaG, we recently published a computational chemistry paper on relating the composition and dynamics of hydrogen-bonding network connecting the chromophore, protein and water molecules to the fluorescence quantum yields of UnaG variants [Lee et al., *Chem. Sci.* **9**, 8325 (2018)].

Thanks for all this extra work. Concerning the 25 min movie: Can you extend the supplementary figure 14 showing the number of localizations vs time by the benefits that might follow due to this long imaging possibility? For this, the measurement has to be redone with a structural target (e.g. filaments) and cannot be cytosolic. Then, the quality of the corresponding SMLM images could be evaluated. How do the images look like in case of mMaple vs. UnaG? E.g. in resolution, in filament coverage (if filaments were imaged)? Does the resolution and coverage get better for UnaG over time?

2. UnaG imaging in this manuscript with 200ms integration times is as slow as PAINT. So, please proof experimentally that you are faster (at high quality, thus without "overactivation conditions"). I furthermore don't think that UnaG performs better than PAINT concerning high background. Like in PAINT, in this technique the probes are added. The added metabolites cause background and the technique needs to make use of special imaging geometries such as TIRF. Thus, the criticism towards PAINT should be tuned down or the authors have to proof that they perform better. PAINT manuscripts are in the 3 nm resolution range, UnaG seems to me at least 5 times worse.

We agree that UnaG's photon number is much lower than PAINT. But, the background from bilirubin is nearly non-existent and does not require special imaging geometries. Bilirubin is fluorogenic: free bilirubin in solution is non-fluorescent due to the free dihedral rotations at the central carbon atoms; non-fluorescent bilirubin becomes fluorescent when binding to UnaG protein that restricts the rotation [Kumagai et al., *Cell* **153**, 1602 (2013)]. We recently published a paper on relating the rotational degree of freedom to the fluorescence quantum yield by using computational chemistry simulations [Lee et al., *Chem. Sci.* **9**, 8325 (2018)]. Owing to the non-fluorescent nature of the ligand, there is no need to implement special illumination geometry such as TIR, which we have never used throughout the current work. All of our images have been obtained with standard Epi illumination. To highlight the benefit, we added Fig. R3 that shows the raw diffraction-limited images of individual molecules of UnaG in which the fluorescence background is virtually unchanged up to 1 micromolar concentration of bilirubin under Epi illumination. In contrast, even 1.0 nM PAINT probe increased the fluorescent background significantly under Epi illumination and required TIR to suppress the background (Fig. R3b,c).

Fig. R3 Comparison of the raw images of DNA-PAINT and UnaG SMLM images under different illumination geometries. **a** In UnaG SMLM imaging, in which the unbound ligand of bilirubin is non-fluorescent, the background under Epi illumination geometry were virtually unchanged up to 1,000 nM concentration of exogenously supplemented bilirubin. **b** In PAINT imaging, even 1.0 nM elevated the background significantly in Epi illumination. **c** Total internal reflection illumination (TIR) was required to decrease the fluorescence background of 1.0 nM PAINT probes.

From the initial submission to the current revision rounds, all of our UnaG SMLM images in the manuscript were obtained with standard Epi illumination. We apologize that our method section in the manuscript did not clearly state this fact. Now, we included additional descriptions on the non-fluorescent nature of free bilirubin and the illumination geometry in the manuscript and placed Fig. R3 as Fig. S13 along with a brief discussion to prevent the readers from making false assumption as the reviewer did. We also added this unique benefit to the list of UnaG's advantages, which now reads:

“UnaG offers the unique combination of following advantages simultaneously: (1) simple labeling method of standard genetic incorporation; (2) high photon numbers comparable to EosFP and Atto 488; (3) reversible switching between the dark and green fluorescence states; (4) fluorescence recovery by binding a natural metabolite (i.e. bilirubin) that can be exogenously supplemented when necessary. (5) little fluorescent background under Epi illumination at micromolar ligand concentration that acts as pseudo-unlimited reservoir (Supplementary Fig. S14).”

Thanks to the low background (Fig. R3) and easily controllable kinetics (Fig. 2), UnaG imaging can be faster than PAINT. In Fig. R8 in our response to the point #7, we obtained 2-sec SMLM image with 10-ms exposure time without overactivation condition by increasing the 488-nm intensity by ten times. The 10-fold increase in light intensity linearly accelerated the off-switching rate as we demonstrated in Fig. 2b. The increased off-switching rate allowed us to use the short exposure time of 10 ms and to avoid overactivation even when using 1 μ M exogenous bilirubin by lowering the on-off duty cycle. Again, the non-fluorescent nature of bilirubin allowed us to use the high probe concentration without worrying about the background even in Epi illumination. In contrast, PAINT imaging speed is limited by the chemical nature of the probe. The unbinding rate limits the exposure time in PAINT imaging in the range of hundreds of milliseconds, in general. It may be possible to increase the unbinding rate by changing the chemical nature of the probe, but the chemical change can deteriorate the binding affinity because the dissociation constant (i.e. the equilibrium constant of the dissociation reaction) is basically the unbinding rate constant divided by the binding rate constant. Also, background from fluorescent probes in solution restricts the on-switching rate and the probe concentration is usually in nanomolar level in PAINT.

I am sorry that I missed the fact that all was imaged in EPI-mode and not in TIRF mode and would like to apologize for this mistake. I also would like to thank the authors for clarifying this and for the comparative work added to the manuscript.

3. Better photon numbers than primed conversion: Please proof experimentally that you get more photons and a better resolution in vivo using a reference structure and image it both, with normal FPs, UnaG and primed conversion FPs.

Previously, we reported ~1,000 photons for mMaple3 and ~1,200 photons for UnaG in Fig. 4a-c. Among primed conversion FPs, mMaple3 offers the largest number of photons, 180 photons under primed conversion (Fig. R4a) [Turkowsky et al., *Angew. Chem. Int. Ed.* **56**, 11634 (2017)]. Also, the measured fluorescence intensity of mMaple3 under primed conversion is at least 5-times lower than conventional conversion mediated by 405 nm at the physiological conditions, even when the primed conversion rate is faster than UV-activation rate (Turkowsky et al., Fig. R4b), indicating single-molecule photon number notably decreased with primed conversion. mEos2 also showed lower photon numbers under primed conversion than UV-activation (Fig. R4c from [Mohr et al., *Angew. Chem. Int. Ed.* **56**, 11628 (2017)]). Since each comparison was performed under identical experimental settings, it is rational to conclude that mMaple3 provides lower photons under primed conversion than UnaG and/or 405-activated mMaple3, by accepting previously published results of three data sets from two independent papers. The photon number comparisons of UnaG and mMaple3 in Fig. 4c and SMLM images of vimentin in Fig. R6 would serve as the experimental proofs for UnaG and 405-activated mMaple3.

I do not agree with the argumentation which contains factual errors:

- a) The referenced measurements of mMaple3 with primed conversion and UV are ensemble and NOT single molecule measurements and thus do not reflect on individual molecule fluorescence. I would argue that the difference in brightness in the referenced data is most likely caused by the efficiency of conversion which leads to less converted molecules and thus lower brightness.

- b) pr-mEos2 is a different fluorophore than mEos2. It has the arginine to threonine mutation at position 69 which causes loss in QY (see e.g. Berardozi et al., doi.org/10.1021/jacs.5b09923, who were the first to characterize and report on this mutation in green-to-red convertible Anthozoans).
- c) the individual brightness of single fluorescent spots depends on the individual setup. A measurement on the authors setup with 1000 counts cannot be simply compared to a literature value of 180 counts. Only comparative measurements on the same setup (e.g. same camera, same filters etc.) can show how different fluorophores compare (and still is then biased to the exact setup, e.g. a fluorophore which is brighter but imaged with a non-optimal filter can appear dimmer than a fluorophore with lower QY but measured under optimal settings).

Figure below:

Panels A and B reproduced with permissions from Turkowyd et al., A General Mechanism of Photoconversion of Green-to-Red Fluorescent Proteins Based on Blue and Infrared Light Reduces Phototoxicity in Live-Cell Single-Molecule Imaging, *Angew. Chem. Int. Ed.* (John Wiley and Sons). 56(38):11634-11639 (2017) Copyright © 1999-2019 John Wiley & Sons, Inc. All rights reserved

Panel C reproduced with permissions from Mohr et al., Rational Engineering of Photoconvertible Fluorescent Proteins for Dual-Color Fluorescence Nanoscopy Enabled by a Triplet-State Mechanism of Primed Conversion, *Angew. Chem. Int. Ed.* (John Wiley and Sons). 56(38):11628-11633 (2017) Copyright © 1999-2019 John Wiley & Sons, Inc. All rights reserved

Fig. R4 Comparison of the brightness of primed conversion FPs. **a** Single-molecule photon numbers of fluorescent proteins under primed conversion. The figure was adopted from Supplementary Figure 8 of [Turkowyd et al., *Angew. Chem. Int. Ed.* **56**, 11634 (2017)]. **b** Relative brightness (red) of 405-mediated mMaple3 (upper panel) and primed converting mMaple3 (lower panel). The figure was adopted from Supplementary Figure 3c(iii) of Turkowyd et al. again. **c** Single-molecule photon numbers of mEos2 under 405-nm-mediated activation and primed conversion. Primed conversion mEos2 showed lower photon counts than 405-activated mEos2. The figure was adopted from Figure 3c of [Mohr et al., *Angew. Chem. Int. Ed.* **56**, 11628 (2017)].

In addition to the photon numbers, UnaG offers several benefits compared to mMaple3, one of the brightest primed conversion FPs (pcFPs), as we emphasize through all the manuscript and response letter, in terms of small size (about half of GFP) that leads into higher labeling coverage (Fig. R6), refueling property for long-term imaging (Fig. R7), dark-to-green transition for streamlined, high-performance multicolor live-cell imaging (Figs.6, S18 and S20). All pcFPs, including mMaple3, share common molecular features such as molecular weight comparable to GFP, covalently linked chromophores, and green-to-red transition. Therefore, many of our experimental comparisons of mMaple3's performance in SMLM in comparison to UnaG intrinsically can be applied to other pcFPs. Also, we can deduce other pcFPs' properties by referencing the mMaple3 results. For instance, Dendra2 is the most commonly used pcFP for SMLM, but some of its SMLM performance such as labeling efficiency [Wang et al., *Proc. Natl. Acad. Sci.* **111**, 8452 (2014)] and photon numbers under primed conversion [Turkowyd et al., *Angew. Chem. Int. Ed.* **56**, 11634 (2017)] are reported as inferior to mMaple3. Thus, we can deduce Dendra2's labeling efficiency and photon numbers to be lower than UnaG.

Nonetheless, we found that the broad spectral window of pcFP (green-to-red emission as well as dual conversion lights in blue and red/infrared) is better to highlight than the low photons or other disadvantages because it makes pcFPs incapable of multicolor imaging modalities that UnaG can easily achieve. The pre-conversion emission in green overlaps with dark-to-green FPs and the post-conversion emission in red overlaps with dark-to-red FPs. By sequential primed conversion followed by UV-activation, a pcFP can be sequentially distinguished from a photoactivatable FP that cannot be activated by primed conversion such as PAmCherry [Mohr et al., *Angew. Chem. Int. Ed.* **56**, 11628 (2017); Virant et al., *Int. J. Mol. Sci.* **18**, 1524 (2017)]. Also, red PAINT probes were distinguished sequentially after imaging and bleaching pcFPs in Virant et al. However, these approaches of sequential multicolor imaging can neither observe different structures in a live cell at the same time nor collect a time-lapse movie composed of multiple SMLM frames in multiple color channels.

Far-red dyes may allow for simultaneous SMLM imaging with pcFPs, albeit the complication in labeling as we discussed in the point #1.

However, the red excitation laser for far-red dyes can serve as efficient light source of primed conversion [Klementieva et al., *Chem. Commun.* **52**, 13144 (2016)]. Klementieva et al. used 250 W/cm² of 640 nm for SMLM imaging of pcFPs, which is 1-2 orders of magnitude lower than the excitation intensity for live-cell SMLM imaging of Alexa Four 647.

Therefore, the red laser for exciting a far-red dye can induce primed conversion of pcFPs at a significantly higher rate than needed for proper SMLM imaging, and therefore overly activate pcFPs. Or, the red laser for proper primed conversion rate can be insufficient for high-performance SMLM imaging of far-red dyes. Overall, it is highly challenging to find laser intensities, particularly for red laser, that give optimal on-switching rate of a pcFP and optimal off-switching rate of a far-red dye at the same time. Probably because of this reason, simultaneous multicolor imaging of pcFPs has not been demonstrated yet, to the best of our knowledge.

Parallel read-out of a green-to-red converting FP and a dark-red fluorophore has been done before and is a standard technique for structural SMLM studies, see e.g. this extensive SMLM work done in yeast: [https://www.cell.com/cell/fulltext/S0092-8674\(18\)30800-6](https://www.cell.com/cell/fulltext/S0092-8674(18)30800-6).

In contrast, we demonstrated simultaneous multicolor imaging capacities of UnaG with simple filter-based approach, by utilizing the background-free dark-to-green transition of UnaG. In our previous revision, we demonstrated (1) two-color imaging with bright red membrane probes (Fig. 6), (2) FP-only two-color imaging with PAmCherry (Fig. S18b), (3) three-color SMLM imaging with Alexa Flour 647 (Fig. S18d), (4) simultaneous observation of structural evolution of mitochondrial matrix and inner membrane during division and fusion (Fig. 6e,f). All of these simultaneous multicolor imaging applications are near impossible for pcFPs to carry out due to the broad emission window in green-to-red and multiple conversion light sources in blue and red/infrared. Moreover, Virant et al. used 4 kW/cm² of 488-nm light for bleaching the residual Dendra2, which is ten times more intense than the excitation intensity of UnaG imaging. As a result, primed conversion multicolor SMLM can be more toxic to cells than UnaG imaging.

Finally, to highlight the more prominent advantage of UnaG that we already demonstrated experimentally in various combinations of colors, probes and applications in Fig. 6, S18 and S20, we deleted the phrase of mentioning low photons as major disadvantage of pcFPs in the main text and instead inserted a sentence that discusses the broad spectral window and the limit in multicolor capability that reads, “*primed conversion FPs convert the emission from green to red upon exposure to blue and red/infrared lasers, making simultaneous multicolor imaging near impossible*”.

I would like to explicitly acknowledge the authors work on simultaneous multi-color read-out. Simultaneous read-out is rare in multi-color SMLM works, most are sequential. UnaG is an excellent candidate for simultaneous multi-color SMLM imaging as discussed above. Nevertheless, multi-color imaging was not the scope of my original comment. Here I was interested in the quality of (single-color) UnaG images compared to other fluorophores. I was interested to know which probe gives me the highest quality for single-color SMLM images. That’s why I suggested the nuclear pore as a reference structure to compare fluorophores. A discussion, which combination of probes is best for multi-color imaging is a completely different (but also nevertheless important) discussion. But also here it would be important to know (when choosing UnaG as one of the two colors), what image quality one can expect for this color channel.

4. I don’t see any experimental proof for this sentence “We would like to emphasize that UnaG offers the best image quality among fluorescent proteins that are workhorses for live-cell imaging” in the manuscript. Even if there is a photon numbers measurement, the higher photon counts do not show in the images. So, please proof the utility of the tool on a reference standard in living cells, e.g. nuclear pore, microtubule hollow structure (giving a profile with a dip in the middle), FtnA oligomers, Glycine receptor and show that you obtain better image quality. Just an image of vimentin does not proof anything. There are high quality images on reference standards. Using reference standards is independent of the setup used or the ability of the experimenter – thus please perform this experiment. This is needed to introduce a new tool, e.g. the recent BC2 nanobody paper performed FtnA standard measurements vs. SNAP tag, the just uploaded back-to-back preprints from the Ries and Jungmann groups (<https://www.biorxiv.org/content/10.1101/582668v1>, <https://www.biorxiv.org/content/10.1101/579961v2>) show nuclear pore imaging with protein tags, dSTORM, GFP nanobodies, DNA-PAINT and FPs at very high quality. This is the current standard of the field the authors have to benchmark against, not images from publications more than 5 years ago.

Among the reviewer’s suggested experiments, we benchmarked the BC2 nanobody paper [Virant et al., *Nat. Commun.* **9**, 930 (2018)] since it is published recently in the same journal. Following the reviewer’s suggestion, we generated UnaG-FtnA and performed measurement of UnaG-FtnA along with SNAP-FtnA (Fig. R5). For SNAP-FtnA, we obtained 72.5% of completeness of labeling that is consistent to the published value in [Finan et al., *Angew. Chem. Int. Ed.* **54**, 12049

(2015)]. UnaG-FtnA measurement yielded 94.6% of completeness of labeling, consistent to near complete labeling of fluorescent proteins as shown for YFP-FtnA in Finan et al. This is consistent to the high vimentin labeling coverage in Fig. R6. Note that the completeness of labeling for Alexa Flour 647 when using SNAP-tag or BC2 nanobody is lower than fluorescent proteins such as YFP and UnaG probably because of incomplete conjugation reaction of the ligand or incomplete binding of the nanobody to the tag. UnaG's near complete labeling is likely due to the high binding affinity of bilirubin to UnaG ($K_d = 98$ pM). Therefore, despite the lower photon count of UnaG, UnaG has advantage in labeling efficiency in SMLM. We now added the FtnA result as Fig. S12 in the supplementary material.

Fig. R5 Completeness of labeling for UnaG and SNAP-tag labeling systems using FtnA-oligomers of 24 subunits. Mean values of FtnA-24mer fluorescence intensities are presented as percentage of theoretical estimation. **a** Representative image of FtnA-24mer cluster labeled with UnaG. **b** Single-molecule photon number distribution of UnaG in measurement conditions, presented with the mean value. **c** Intensity distribution of FtnA-UnaG clusters. The mean value was used to calculate the completeness of labeling in **g**. **d-f** Analogous data set with **a-c** for SNAP-tag labeling system. **g** Measured completeness of labeling of UnaG and SNAP-tag that yielded 94.6% and 72.5% for UnaG and SNAP-tag, respectively.

In addition, the BC2 nanobody paper used vimentin as a major reference target. Fig. 1,2 of the paper among the total four images in the main text used vimentin labeled with various methods including PamCherry, GFP and BC2 nanobodies. Therefore, vimentin SMLM images also serve as good benchmark for us. As we discussed in detail in the point #2 and #5, the labeling coverage of UnaG-labeled vimentin was better than mMaple3 and comparable to the BC2 nanobody results, indicating that the small size of UnaG helps to improve SMLM image quality in terms of labeling coverage.

I would like to thank the authors for this extra work which nicely quantitates the performance of the UnaG labeling and read-out. I am convinced that UnaG labeling performs better than the SNAP-tag as shown in this data. Maybe just one comment on individual spot brightness (adding to the previous comments above): The authors themselves have measured 1200 photons per UnaG. Here they show 222 AD-counts per UnaG which hints at an even lower photon number (e.g. the conversion in our setup is 1000 AD counts ~ 150 photons). Assuming that all measurements are done on the same setup, how can the authors explain these drastically different photon counts? Do they now maybe agree with me that spot photon counts are not easily comparable (see comment 3)?

5. I honestly currently see no advantage of UnaG in SMLM imaging quality in comparison to mMaple3 or mEos3.2. Please proof with an in vivo cell reference structure. The only advantage I see is something the authors don't show: All FPs need oxygen for maturation – UnaG works without and could thus be an ideal tag for anaerobic cultures. Thus, an application of this technique for exploring new biology in e.g. anaerobe archaea would be novel and cannot be done with FP tags.

To directly compare UnaG and mMaple3, we received the mMaple3-Vim plasmids from the authors of [Wang et al., *Proc. Natl. Acad. Sci.* **111**, 8452 (2014)] and performed mMaple3 imaging of vimentin structures (Fig. R6). Also, the image published in Wang et al. is included in Fig. R2b, next to the new UnaG image of vimentin in Fig. R2c. Following the

quantitative comparison of SMLM images of vimentin labeled with various tags and fluorophores in [Virant et al., *Nat. Commun.* **9**, 930 (2018)], we measured the lengthwise labeling coverages of thin filaments in SMLM images obtained with UnaG and mMaple3. The average coverage was 76% for UnaG and 56% for mMaple3. The qualitative difference of more continuous fibrils in the UnaG image (Fig. R6a,b) is consistent to the quantitative difference in the values of labeling coverage (Fig. R6c). As we discussed in detail in the point #2 along with additional comparison to PAmCherry and BC2 nanobody, UnaG-labeled vimentin showed higher labeling coverage than mMaple3 probably due to the smaller size. Note that Wang et al. reported the labeling efficiency of mEos3.2 to be much lower than that of mMaple3. We added the labeling coverage comparison of mMaple3 and UnaG as Fig. S11 in the supplementary material.

Fig. R6 Direct comparison of SMLM images of vimentin fibers fused with UnaG and mMaple3 that are imaged in our laboratory. **a** SMLM images of vimentin labeled UnaG and mMaple3. Note that the two images were analyzed and rendered in the exact same way. Specifically, we rendered each image with normalized Gaussian with 10-nm pixel size in ThunderSTORM. **b** Traced and straightened vimentin fibers labeled with UnaG (top) and mMaple3 (bottom), obtained from **a** by using manual segmentation and straighten selection tools in imageJ. **c** Lengthwise labeling coverage histograms obtained from 88 fibrils for UnaG and 52 fibrils for mMaple3 as exemplified in **b**. We followed the analysis method in [Virant et al., *Nat. Commun.* **9**, 930 (2018)]. Briefly, we manually traced thin, individual filaments and straightened the segmented images using ImageJ. Then, a custom MatLab code converted a straightened image into a binary image and calculated the fraction of covered area of the middle 3 pixels.

As an additional benefit of UnaG to mMaple3, we performed a 25-min-long time-lapse imaging of cytosolic UnaG and mMaple3 in Fig. R7. The UnaG localization numbers were virtually unchanged for more than 25 min probably because of the reversible fluorescence recovery of UnaG by the near infinite pool of free, non-fluorescent ligands at high concentration of 1 μM (blue line in Fig. R7). In contrast, mMaple3 localizations were exponentially (red line in Fig. R7) decreased possibly due to irreversible photobleaching of the chromophore that is covalently attached to the protein (Fig. S1). The mechanism of reversible switching of UnaG is quite unique because it stems from noncovalent, highly specific binding of the ligand to the protein (Fig. S1). In conventional FPs in SMLM including mMaple3, the photoconversion or photoswitching occur through the chemical change of the chromophore such as photo-isomerization or photo-oxidation. In fact, UnaG is the first and only FP with noncovalently bound chromophore [Kumagai et al., *Cell* **153**, 1602 (2013); Lee et al., *Chem. Sci.* **9**, 8325 (2018)]. Purely noncovalent interactions between the chromophore and the protein along with the use of highly concentrated pool of nonfluorescent ligands without complication of delivery or notable cell toxicity, both of which are quite unique for UnaG, contributed to negate the effect of photobleaching in long-term SML imaging as in Fig. R7. We include the time traces of localization numbers in the new Supplementary Fig. S14.

Fig. R7 Number of switching events (i.e. localizations) of UnaG and mMaple3 in extended periods of SML imaging. Localization numbers of cytosolic UnaG (blue) and mMaple3 (red) expressed in fixed Cos7 cells imaged at an exposure time of 40 ms. The UnaG samples were supplemented with 1 μM bilirubin in solution. While the switching events of mMaple3 diminished due to irreversible photobleaching, UnaG continued to produce switched-on spots thanks to the fluorescence recovery by the unbound bilirubin ligands in solution.

Being able to record for a long imaging period like in the authors 25 min movie is indeed an advantage. Nevertheless, being able to record for a long time is not the only parameter that determines image quality. I would like to recommend that the authors repeat the experiment of the supplemental figure 14 with mMaple3 and UnaG fused to a reference structure (e.g. vimentin) to be able to show comparative images of both approaches (see also previous comment on this). Here, UnaG should produce the denser and better resolved image compared to mMaple3 after 25 min, whereas when only using the localizations of the first 5 min both should be rather equal (judging from similar photon counts).

Another benefit to point out is that UnaG allow for crosstalk-free, high-resolution, three-color imaging in live cells. We experimentally demonstrated simultaneous three-color live-cell imaging in Fig. S18d by combining UnaG, MitoTracker Red and Alexa Fluor 647, each of which offer the most photon outputs in its color panel of green, red or far-red. It was possible because UnaG fluorescence switch between the nonfluorescent dark state and green fluorescent state while giving out the largest photon outputs among green switchable fluorophores. This application would be impossible with mMaple3 or mEos3.2 that photoconvert from green to red, covering two channels among the three color windows.

In summary, we experimentally showed UnaG's benefits in SMLM imaging in comparison to mMaple3 or mEos3.2. as follows. (1) The high labeling coverage of vimentin SMLM image would have come from the small size (Fig. R6). (2) During a long-term imaging session in which mMaple3 localization numbers diminished significantly due to the irreversible photobleaching, UnaG localization number stayed virtually unchanged owing to the reversible fluorescence and the nonfluorescent ligand (Fig. R7). (3) Simultaneous three-color live-cell imaging with high photons and little crosstalk is possible with UnaG (Fig. S18d), but not with mMaple3 or mEos3.2.

We appreciate the reviewer's suggestion for an application to anaerobic cultures. This would indeed highlight that UnaG's fluorescence does not rely on oxidation of amino acids constituting the chromophore that occurs in most fluorescent proteins. Also, the suggested application would demonstrate potential for opening a window for new biology. However, we intend to highlight UnaG's benefits for more general applications for appealing to more general public. As we listed above, there are a number of advantages of UnaG that would be useful for standard SMLM imaging, in particular for live-cell and multicolor applications.

6. I highly appreciate the fixed cell experiment with 0.2 μM of fresh bilirubin and 0.8 μM of pre-bleached bilirubin. I would recommend to add this figure to the SI.

We appreciate the reviewer's suggestion on the experiment that helped to eliminate the possibility of competitive inhibition by bleached bilirubin. We added the result in SI (Fig. S3) as the reviewer suggested.

7. This sentence is not experimentally shown, is it? "The genetically incorporated UnaG can be readily applied to live-cell SML imaging and offers fast tracking for the cellular dynamics due to the high photon numbers and easily controllable UV-free kinetics." At least not at the quality I would expect, I know of the 1s overreacting conditions experiment but this is not what I am looking for.

As an experimental proof, we obtained 2-sec snapshots of UnaG-labeled clathrin-coated pits as shown in Fig. R8 under non-overactivation condition. To accelerate the SMLM imaging speed, we increased the excitation intensity by ten times for promoting the off-switching rate fit for 10-ms exposure time while using 1 μM bilirubin for high on-switching rate. The 10-fold increase in excitation also helped to avoid overactivation (i.e. overlaps of single-molecule images in a camera frame). Since the on-off duty cycle can be represented as $1/[1+k_{\text{off}}/k_{\text{on}}]$, the 10-fold higher laser intensity reduced the on-off duty cycle by ten times and therefore the probability of image overlaps by ten times. As a consequence, we could resolve the ring-like projections of clathrin-coated pits with single-emitter fitting algorithm (Fig. R8).

Fig. R8 2-sec snapshot of UnaG-labeled clathrin-coated pits in live cell without using overactivation condition. The SMLM image was reconstructed from 200 frames at 10 ms exposure time that are analyzed with standard single-emitter fitting algorithm without further processing.

However, we would like to point that overactivation condition can be useful in live-cell imaging because the imaging speed can be accelerated without increasing the light dose that can induce cell stress. Indeed, “overactivation” was used for massively accelerating the imaging speed by using many state-of-the-art analysis software packages including deep learning algorithms [Marsh et al., *Nat. Methods* **15**, 689 (2018); Ouyang et al., *Nat. Biotech.* **36**, 460 (2018)]. These algorithms successfully demonstrated to reconstruct proper SML images from raw data containing highly overlapped single-molecule fluorescence images, and therefore allowed to significantly reduce the number of camera frames per SML image. Note that PAINT would face the challenge of high background to achieve overactivation condition because high concentration of PAINT probe for increasing the on-switching rate would result in high background as demonstrated in Fig. R2c. Even with TIR illumination, the exponentially decreasing evanescent field can still excite a small fraction of fluorescent probes in solution. This resulted in a noticeable background in Fig. R2c with 1 nM PAINT probes under TIR excitation. In contrast, we could use 1 μ M bilirubin and Epi illumination to achieve background-free overactivated condition for reconstructing 1-sec SML snapshots of ER in Fig. 5 with HAWK, a recently published and publicly available software for artifact-free high-density localization analysis [Marsh et al, *Nat. Methods* **15**, 689 (2018)].

Thanks a lot for this extra work.

Point by point response to the reviewer comments

Manuscript #: NCOMMS-18-09874C-Z

Title: “Bright Ligand-activatable Fluorescent Protein for High-quality Multicolor Live-cell Super-resolution Microscopy”

Authors: Jiwoong Kwon, Jong-Seok Park, Minsu Kang, Soobin Choi, Jumi Park, Gyeong Tae Kim, Changwook Lee, Sangwon Cha, Hyun-Woo Rhee, Sang-Hee Shim

Reviewer’s comments to the 2nd revision (in black):

Authors’ comments to the reviewer’s comments to the 2nd revision (in blue):

Reviewer’s comments to the authors 3rd revision (in green):

Authors’ comments to the reviewer’s comments to the 3rd revision (in red):

At this stage I would like to ask the editor for his/her decision as the authors and me have very different opinions on the novelty and utility of the UnaG approach as a SMLM labelling technique. I would like to clearly state that, in my opinion, the experimental evidence does not show sufficient SMLM data quality - which is needed for this approach to be a useful tool. But I am happy to accept other opinions – In case the manuscript is accepted, I would like to kindly ask to publish the rebuttal files with the manuscript for full transparency. To give a bit more details why I keep my opinion after the last revision phase:

In the first review, the reviewer recognized the benefits of our method and said, “*the labeling system is rather small (about half the size of GFP) and can be ‘refuled’ by the pool of available bilirubin – both are good reasons to check and evaluate it as an interesting and promising SMLM labeling option.*”. This statement indicates that the reviewer acknowledged the novelty and utility of our work in very similar perspectives to ours. In our previous revisions, we may have not succeeded in convincing the reviewer with our experimental results and discussions. In this revision round, we performed additional experiments such as (1) SML imaging on reference structures benchmarking a recent paper [Virant et al., *Nat. Commun.* **9**, 930 (2018)] following the reviewer’s suggestion, (2) direct, quantitative comparisons of UnaG and mMaple3 in labeling coverage and long-term imaging capacity and (3) direct comparisons of background in UnaG and PAINT imaging under Epi and TIR illuminations, (4) SMLM imaging at particular conditions such as endogenous level of bilirubin and non-overactivation conditions, and so on. Needless to say, we would be more than happy to expose all the material including the rebuttals of the previous and current revision rounds.

I thank the authors for all the additional experiments done and their consent to publish the review and rebuttal materials. Comments are below.

We also thank the reviewer for his/her continuous in-depth comments that help us to strengthen our manuscript with many supporting data covering various aspects of SML image quality.

1. To be honest, this answer to my last review sparks the essence of my criticism: “In fact, Fig. 1d(ii) that was obtained without supplementing exogenous bilirubin demonstrates that endogenous level of bilirubin can produce SMLM images albeit the lower image quality.” For me, this is not an SMLM image at all. Nobody should use this tag for SMLM if this is its intrinsic quality. This image has nothing to do with a SMLM image that actually resolves something.

We apologize for giving an inappropriate description. It is better to call Fig. 1d(ii) as a partial, undersampled copy of an SMLM image in Fig. 1d(iv). The exact same UnaG proteins targeted to the ER in the same cell used for Fig. 1d(ii) produced the SMLM image in Fig. 1d(iv) that resolved the ER tubules. By using Fig. 1d(i)-(iv), we intended to emphasize the consequences of various buffer conditions and to visually guide the readers to the switching mechanism. If we have collected much more frames under the condition for Fig. 1d(ii) (i.e. with endogenous level of bilirubin), we should have been able to collect much more localizations in total albeit at the low on-switching rate limited by the bilirubin concentration. Therefore, if we extend the imaging session for Fig. 1d(ii) sufficiently to accumulate localizations above the density level required for resolving the ER tubules, we should have been able to reconstruct a proper SMLM image as in Fig. 1d(iv).

I can follow the reasoning of the authors. Nevertheless, their argumentation remains hypothetical. To prove this, they would have to record ER tubules and not CCPs (as shown below) using only endogenous levels of bilirubin. CCPs should be easier to resolve than ER as they require less staining (less abundant and smaller structure). Thus, for resolving CCPs, most likely less bilirubin is needed than for SMLM images of the ER at high quality. In my opinion, the authors should stay with their initial biological system (the ER) and show that the endogenous levels of bilirubin are sufficient for this one to obtain high quality SMLM images. Otherwise, I would recommend to change the text and would not call Fig. 1d(ii) a super-resolved image and would not state that this is a technique that generally works without external supply.

We appreciate the reviewer for consenting our intention regarding Fig. 1d. In the previous revision, we already removed all the claims about SML imaging at the endogenous level of bilirubin. In this round, we further toned down our description in the main text and the caption for Fig. 1d from 'SML images' to 'localization dataset'. Also, when introducing the localization dataset in the main text, we changed the description of our experiment to 'we tested single-molecule localization capability of UnaG'.

To best answer the reviewer's original question of whether the endogenous level of bilirubin can produce SMLM images, we here present the SMLM images of clathrin-coated pits (CCP) without supplementing exogenous bilirubin (Fig. R1). In a fixed cell, we could resolve the ring-like projections of CCPs with endogenous level of bilirubin (Fig. R1a) by reconstructing the SMLM image from 20,000 frames of slowly switching UnaG. In a live cell, we could reconstruct a SMLM image with just 1,000 frames (Fig. R1b). It is because live cells actively retain intracellular bilirubin concentration. In contrast, in fixed cells, cell-permeable bilirubin can be lost during washing steps with bilirubin-free solutions. However, the live-cell SMLM image in Fig.R1b is blurred due to motion and structure evolution in the live cell during the acquisition time of 50 seconds, which was required for reconstructing the SMLM image at the endogenous bilirubin concentration that still limited the on-switching rate. The blurred image is consistent with the previous result [Huang et al., *Nat. Methods* **10**, 653 (2013)]. When necessary to accelerate the acquisition speed, we can simply increase the concentration of bilirubin in the imaging buffer up to micromolar level without increasing the background under Epi illumination as demonstrated in Fig. R3 (for more details about the minimal background with Epi geometry, please see our response to the reviewer's point #2 below). Indeed, as in our response to the point #7, we performed clathrin imaging in live cells with 1 μM exogenous bilirubin and obtained 2-sec snapshots that resolved the ring-like projections (Fig. R8). Note that the 2-sec SMLM image was obtained under non-overactivation condition (i.e. the raw diffraction-limited images of individual fluorophores largely do not overlap and can be fitted with single Gaussians). It was possible to achieve such condition by increasing the light intensity by ten times. Then, the off-switching rate becomes sufficiently high for lowering the on-off duty cycle to the level that ensures only one molecule to turn on at a given frame within a CCP whose dimension is below the diffraction limit.

Fig. R1 SMLM image of clathrin-coated pits (CCP) with UnaG-CLC with endogenous concentration of bilirubin without supplementing bilirubin in the imaging buffer. **a** CCPs in a fixed Cos7 cell whose ring shapes are resolved. **b** CCPs in a live Cos7 cell. The image is reconstructed from 1,000 frames acquired at 50 ms exposure time that took total 50 s. The acquisition time is long enough for the pits to move and to change shape in live cells resulted in blurred SMLM image as previously observed in [Huang et al., *Nat. Methods* **10**, 653 (2013)]. (Adapted by permission from Springer Nature: Nature, Nature Methods. Video-rate nanoscopy using sCMOS camera-specific single-molecule localization algorithms, Huang et al., © 2019 Springer Nature Limited (2013))

I would like to thank the authors for their extra work. I would further ask the authors to include the live cell image of CCPs into the manuscript (in the supplemental I did find some CCPs data, but only fixed as far as I understand?). To show that the blurriness of b) is indeed motion blur, I would like to request that they – instead of a b/w image – color-code the localizations in the image in time (which e.g. should yield a clearly visible “motion-rainbow” per CCP).

We added our experimental results in Fig. R1 as Fig. S19 with temporarily color-coding on the live-cell SML image. In particular, the pit in the lower left site of the SML image clearly shows the “motion-rainbow” from left to right, indicating the directional motion. We used these data set in the section for live-cell SML imaging to describe what

happens to the localization datasets in live cells when the endogenous level of bilirubin is used for on-switching of UnaG since we removed all the statements for SMLM with endogenous level of BR from the previous revision and further toned down in this revision. The results demonstrated the consequence of the lower concentration at the endogenous level to the imaging speed, which is particularly important in live-cell imaging. This new discussion along with the 2-sec snapshots of clathrin pits with exogenous bilirubin in the new Fig. S18 experimentally demonstrate the utility of the controllable switching rates of UnaG. We appreciate the reviewer's suggestion on these additional datasets that strengthen our discussion on controllable rates and their application to live-cell imaging.

There are more holes in the vimentin structure than blobs which were fitted (which are at the denser areas, a typical fitting result for not well controlled photophysics of the probe). Nobody will profit from an image like this – here people should stay with a good confocal but diffraction-limited image to deduce their biological answers. To be publishable, the technique should provide something new and an application which cannot be done with other methods until now. Which one would this be?

We replaced the vimentin SMLM image in Fig. 4e,f with a new image in which thin, peripheral fibrils are better resolved in a similar manner to the recently published BC2 nanobody images of vimentin that the reviewer mentioned in his/her point #3 [Virant et al., *Nat. Commun.* **9**, 930 (2018)]. In Fig. R2, we compare the new vimentin image with UnaG (Figs. R2c) as well as the published results of mMaple3 (Fig. R2b), PAmCherry (Fig. R2a, left), GFP nanobody (Fig. R2a, middle) and BC2 nanobody (Fig. R2a, right) conjugated with Alexa Fluor 647. The vimentin fibrils labeled with PAmCherry were either highly aggregated or sparsely labeled (Fig. R2a, left). The mMaple3 image (Fig. R2b) show more thin fibrils with better labeling coverage than the PAmCherry result [Wang et al., *Proc. Natl. Acad. Sci.* **111**, 8452 (2014)]. For direct comparison of UnaG and mMaple3, we conducted SMLM imaging of mMaple3-labeled vimentin filaments and placed the images with UnaG and mMaple3 side by side in Fig. R6 (see the reviewer's point #4 for more details). Following the method in Virant et al., we also quantified the lengthwise labeling coverage from these images for thin filaments that showed the largest differences in labeling coverage of various labeling methods in Virant et al. (Fig. R6d). The labeling coverage for UnaG (76%) was better than mMaple3 (56%) and comparable to the reported value for BC2 nanobody (~80% for fibers below 75 nm). When comparing the nanobody images (middle and right panels in Fig. R2a) and UnaG image (Fig. R2c), the lower photon number of UnaG than that of Alexa Fluor 647 is reflected in the thicker filament widths. However, UnaG-labeled vimentin fibrils achieved similar level of lengthwise labeling coverage to the BC2 results probably due to UnaG's small size. In the BC2 nanobody paper, the higher labeling coverage of BC2 nanobody than that of GFP nanobody was attributed to the small size of the tag. Likewise, the difference between UnaG and mMaple3 can be attributed to the difference in molecular weight, as we discussed in more detail in response to the point #4. Therefore, the vimentin images serve to illustrate the benefits of UnaG's small size for SMLM that the reviewer recognized in his first review round.

Reproduced from Wang et al., Characterization and development of photoactivatable fluorescent proteins for single-molecule-based superresolution imaging, PNAS. 111(23):8452-7 (2014)

Fig. R2 Comparison of the SMLM images of vimentin labeled with different fluorophores and methods. **a** PAmCherry (left) and nanobodies conjugated with Alexa 647 for GFP (middle) and BC2 tag (right) from [Virant et al., *Nat. Commun.* **9**, 930 (2018)]. **b** mMaple3 from [Wang et al., *Proc. Natl. Acad. Sci.* **111**, 8452 (2014)]; **c** UnaG. The contrasts of the zoom-ins were re-adjusted for better representing the specific views. UnaG provided better labeling coverage compared to other fluorescent proteins including GFP and comparable coverage to that of BC2 nanobody conjugated with Alexa 647 dye, due to the small size.

We'd like to emphasize that UnaG and Alexa Fluor 647 have complementary properties. While Alexa Fluor 647 offer high photon outputs, live-cell labeling of the organic dye involves with multiple steps including genetic incorporation of a tag and delivery of dye-conjugated tag-binders such as nanobody that are impermeable to cells. For instance, Virant et al. used lipid-based protein transfection protocol to deliver the cell-impermeable nanobodies conjugated with Alexa Fluor 647 to intracellular targets inside live cells. In contrast, UnaG labeling follows standard genetic incorporation method of fluorescent proteins and does not require perturbative means for delivery of the ligand. Bilirubin, an endogenous metabolite, is permeable to cell membrane. Thus, by simply adding bilirubin in the medium, we could increase the on-switching rate when necessary. Also, exogenous bilirubin can be omitted when the endogenous level is sufficient, as we demonstrated in Fig. R1. Although the photon output of UnaG is lower than that of the organic dye, it offers the highest photon counts among fluorescent proteins. In addition, UnaG is unique among fluorescent proteins in terms of the small size (about half), the ability to refuel the fluorescence and the narrow spectral window. These benefits are what the reviewer previously recognized and what we again try to provide experimental validations in our responses to the point #5.

To further demonstrate for which cannot be done with conventional fluorescent proteins, we obtained long-term time-lapse movies of UnaG and mMaple3 that are expressed in the cytosol. The time traces of localization number are in Fig. R7 in response to the reviewer's point #5. By the end of the 25-min movie of mMaple3, the localization number diminishes to nearly zero probably because photobleaching permanently damages the chromophore composed of several amino acids of mMaple3. In contrast, UnaG localization number is nearly unchanged within a similar 25-min-

long movie probably due to the replacement of the bleached bilirubin by fresh bilirubin in solution. This comparison is aimed at supporting the reviewer's previous comment in the first review round that recognized the 'refueling' capability, which cannot be done with conventional FPs. The permanent consequence of photobleaching is common in most fluorescent proteins in which the chromophore is covalently bonded to the protein (Fig. S1). In contrast, UnaG is the only FP with purely noncovalent interactions between the chromophore and protein that make it possible to detach the damaged chromophore and to re-bind with a new, fresh one. Inspired by this unique chemical nature of UnaG, we recently published a computational chemistry paper on relating the composition and dynamics of hydrogen-bonding network connecting the chromophore, protein and water molecules to the fluorescence quantum yields of UnaG variants [Lee et al., *Chem. Sci.* **9**, 8325 (2018)].

Thanks for all this extra work. Concerning the 25 min movie: Can you extend the supplementary figure 14 showing the number of localizations vs time by the benefits that might follow due to this long imaging possibility? For this, the measurement has to be redone with a structural target (e.g. filaments) and cannot be cytosolic. Then, the quality of the corresponding SMLM images could be evaluated. How do the images look like in case of mMaple vs. UnaG? E.g. in resolution, in filament coverage (if filaments were imaged)? Does the resolution and coverage get better for UnaG over time?

As suggested by the reviewer, we performed new experiments of vimentin labeled with UnaG and mMaple3 for an extended acquisition of 32 mins. From these new datasets, we analyzed the labeling coverage and localization density of SML images in different acquisition time ranges of the entire acquisition period (0-32 min) as well as the first and last halves (0-16 and 16-32 min, respectively). UnaG showed better photobleaching resistance again that resulted in much higher labeling coverage and localization density for the SML image acquired in the last half of the acquisition periods (16-32 mins). Now our new experimental results from vimentin filaments replaced the old Fig. S14 with cytosolic expression.

2. UnaG imaging in this manuscript with 200ms integration times is as slow as PAINt. So, please proof experimentally that you are faster (at high quality, thus without "overactivation conditions"). I furthermore don't think that UnaG performs better than PAINt concerning high background. Like in PAINt, in this technique the probes are added. The added metabolites cause background and the technique needs to make use of special imaging geometries such as TIRF. Thus, the criticism towards PAINt should be tuned down or the authors have to proof that they perform better. PAINt manuscripts are in the 3 nm resolution range, UnaG seems to me at least 5 times worse.

We agree that UnaG's photon number is much lower than PAINt. But, the background from bilirubin is nearly non-existent and does not require special imaging geometries. Bilirubin is fluorogenic: free bilirubin in solution is non-fluorescent due to the free dihedral rotations at the central carbon atoms; non-fluorescent bilirubin becomes fluorescent when binding to UnaG protein that restricts the rotation [Kumagai et al., *Cell* **153**, 1602 (2013)]. We recently published a paper on relating the rotational degree of freedom to the fluorescence quantum yield by using computational chemistry simulations [Lee et al., *Chem. Sci.* **9**, 8325 (2018)]. Owing to the non-fluorescent nature of the ligand, there is no need to implement special illumination geometry such as TIR, which we have never used throughout the current work. All of our images have been obtained with standard Epi illumination. To highlight the benefit, we added Fig. R3 that shows the raw diffraction-limited images of individual molecules of UnaG in which the fluorescence background is virtually unchanged up to 1 micromolar concentration of bilirubin under Epi illumination. In contrast, even 1.0 nM PAINt probe increased the fluorescent background significantly under Epi illumination and required TIR to suppress the background (Fig. R3b,c).

Fig. R3 Comparison of the raw images of DNA-PAINT and UnaG SMLM images under different illumination geometries. **a** In UnaG SMLM imaging, in which the unbound ligand of bilirubin is non-fluorescent, the background under Epi illumination geometry were virtually unchanged up to 1,000 nM concentration of exogenously supplemented bilirubin. **b** In PAINT imaging, even 1.0 nM elevated the background significantly in Epi illumination. **c** Total internal reflection illumination (TIR) was required to decrease the fluorescence background of 1.0 nM PAINT probes.

From the initial submission to the current revision rounds, all of our UnaG SMLM images in the manuscript were obtained with standard Epi illumination. We apologize that our method section in the manuscript did not clearly state this fact. Now, we included additional descriptions on the non-fluorescent nature of free bilirubin and the illumination geometry in the manuscript and placed Fig. R3 as Fig. S13 along with a brief discussion to prevent the readers from making false assumption as the reviewer did. We also added this unique benefit to the list of UnaG's advantages, which now reads:

“UnaG offers the unique combination of following advantages simultaneously: (1) simple labeling method of standard genetic incorporation; (2) high photon numbers comparable to EosFP and Atto 488; (3) reversible switching between the dark and green fluorescence states; (4) fluorescence recovery by binding a natural metabolite (i.e. bilirubin) that can be exogenously supplemented when necessary. (5) little fluorescent background under Epi illumination at micromolar ligand concentration that acts as pseudo-unlimited reservoir (Supplementary Fig. S14).”

Thanks to the low background (Fig. R3) and easily controllable kinetics (Fig. 2), UnaG imaging can be faster than PAINT. In Fig. R8 in our response to the point #7, we obtained 2-sec SMLM image with 10-ms exposure time without overactivation condition by increasing the 488-nm intensity by ten times. The 10-fold increase in light intensity linearly accelerated the off-switching rate as we demonstrated in Fig. 2b. The increased off-switching rate allowed us to use the short exposure time of 10 ms and to avoid overactivation even when using 1 μ M exogenous bilirubin by lowering the on-off duty cycle. Again, the non-fluorescent nature of bilirubin allowed us to use the high probe concentration without worrying about the background even in Epi illumination. In contrast, PAINT imaging speed is limited by the chemical nature of the probe. The unbinding rate limits the exposure time in PAINT imaging in the range of hundreds of milliseconds, in general. It may be possible to increase the unbinding rate by changing the chemical nature of the probe, but the chemical change can deteriorate the binding affinity because the dissociation constant (i.e. the equilibrium constant of the dissociation reaction) is basically the unbinding rate constant divided by the binding rate constant. Also, background from fluorescent probes in solution restricts the on-switching rate and the probe concentration is usually in nanomolar level in PAINT.

I am sorry that I missed the fact that all was imaged in EPI-mode and not in TIRF mode and would like to apologize for this mistake. I also would like to thank the authors for clarifying this and for the comparative work added to the manuscript.

Thanks to the reviewer's comments, we more clearly describe our experimental condition and can appeal UnaG's unique advantage of compatibility with Epi-illumination, which is incompatible with PAINT. We again thank the reviewer to make our revised manuscript clearer and more appealing.

3. Better photon numbers than primed conversion: Please proof experimentally that you get more photons and a

better resolution in vivo using a reference structure and image it both, with normal FPs, UnaG and primed conversion FPs.

Previously, we reported ~1,000 photons for mMaple3 and ~1,200 photons for UnaG in Fig. 4a-c. Among primed conversion FPs, mMaple3 offers the largest number of photons, 180 photons under primed conversion (Fig. R4a) [Turkowsky et al., *Angew. Chem. Int. Ed.* **56**, 11634 (2017)]. Also, the measured fluorescence intensity of mMaple3 under primed conversion is at least 5-times lower than conventional conversion mediated by 405 nm at the physiological conditions, even when the primed conversion rate is faster than UV-activation rate (Turkowsky et al., Fig. R4b), indicating single-molecule photon number notably decreased with primed conversion. mEos2 also showed lower photon numbers under primed conversion than UV-activation (Fig. R4c from [Mohr et al., *Angew. Chem. Int. Ed.* **56**, 11628 (2017)]). Since each comparison was performed under identical experimental settings, it is rational to conclude that mMaple3 provides lower photons under primed conversion than UnaG and/or 405-activated mMaple3, by accepting previously published results of three data sets from two independent papers. The photon number comparisons of UnaG and mMaple3 in Fig. 4c and SMLM images of vimentin in Fig. R6 would serve as the experimental proofs for UnaG and 405-activated mMaple3.

I do not agree with the argumentation which contains factual errors:

- a) The referenced measurements of mMaple3 with primed conversion and UV are ensemble and NOT single molecule measurements and thus do not reflect on individual molecule fluorescence. I would argue that the difference in brightness in the referenced data is most likely caused by the efficiency of conversion which leads to less converted molecules and thus lower brightness.
- b) pr-mEos2 is a different fluorophore than mEos2. It has the arginine to threonine mutation at position 69 which causes loss in QY (see e.g. Berardozi et al., doi.org/10.1021/jacs.5b09923, who were the first to characterize and report on this mutation in green-to-red convertible Anthozoans).
- c) the individual brightness of single fluorescent spots depends on the individual setup. A measurement on the authors setup with 1000 counts cannot be simply compared to a literature value of 180 counts. Only comparative measurements on the same setup (e.g. same camera, same filters etc.) can show how different fluorophores compare (and still is then biased to the exact setup, e.g. a fluorophore which is brighter but imaged with a non-optimal filter can appear dimmer than a fluorophore with lower QY but measured under optimal settings).

Since our comparisons rely on multiple experiments for multiple proteins under two different conditions of UV-activation and primed conversion, our previous explanations may have failed to convince the reviewer due to the reasons that the reviewer commented above. We would like to supplement the following reasonings in response to the reviewer's comments to further clarify our argument.

- (i) Indeed, Fig. R4b shows ensemble fluorescence intensities of mMaple3 under 405-nm and 488/730-nm photoconversion conditions, which surely affected by the conversion efficiency, as well as the single-molecule brightness. Thus, we extracted the reference results that shows both of the ensemble intensity (red in Fig. R4b) and the conversion efficiency (black in Fig. R4b). At the physiological conditions (pH 7-8), 405-nm activation showed $>0.5 \times 10^5$ ADC and $\sim 0.05 \text{ s}^{-1}$ for the intensity and conversion rate, respectively. 488/730nm activation resulted in $<1.0 \times 10^4$ ADC and $\sim 0.1 \text{ s}^{-1}$. 405-nm activation showed more than 5-times larger intensity even with 2-time lower conversion rate, that means mMaple3 gives much more single-molecule photons with normal 405-nm activation than primed conversion.
- (ii) We agree to the reviewer that it is not fair to directly compare the photon numbers of mEos2 under 405-nm activation and pr-mEos2 under primed conversion. However, our major intention of including Fig. R4c is to compare mMaple3 and pr-Eos2 under primed conversion. In Fig. R4c, the relative photon numbers of Dendra2 and pr-mEos2 showed almost identical histogram. We used this data to support that mMaple3's photon number is larger than that of pr-mEos2 by using Dendra2 as a reference. It is because Fig. R4a does not include pr-mEos2 and Fig. R4c does not include mMaple3. By combining the two data sets in Fig. R4a and R4c, we concluded that mMaple3 gives out the largest number of photons among primed conversion FPs.
- (iii) We also agree to the reviewer that the absolute photon numbers highly depend on the experimental setup. However, the relative brightness still can be used if a certain fluorophore can be used as a reference to compare results from different setups. Fig. R4a presents the single-molecule photon numbers of various primed conversion

FPs, which surely obtained from an identical experimental setup. Here, mMaple3 gave more photon numbers than other primed conversion FPs including Dendra2, whose photon number distribution is nearly identical to pr-mEos2 [from the point (ii) above and Fig. R4c], all under primed conversion condition. Therefore, we used mMaple3, the FP producing the largest number of photons under primed conversion, as a standard to compare with UnaG.

As we already discussed in the point (i) above about the relative brightness of mMaple3 in different activation condition with Fig. R4b, mMaple3 emits much more photons under normal 405-nm activation than under primed conversion. When we compared mMaple3 and UnaG under 405-nm activation in Fig. 4c, UnaG produced slightly more photons. Thus, we reasoned that mMaple3 under primed conversion would produce much less photons than UnaG under 405-nm activation. Then, since mMaple3 produces the largest number of photons among all the primed conversion FPs reported including mutant FPs engineered for efficient primed conversion, we concluded that primed conversion FPs provide less photons than 405-nm activated mMaple3, and thus UnaG.

Figure below:

Panels A and B reproduced with permissions from Turkowyd et al., A General Mechanism of Photoconversion of Green-to-Red Fluorescent Proteins Based on Blue and Infrared Light Reduces Phototoxicity in Live-Cell Single-Molecule Imaging, *Angew. Chem. Int. Ed. (John Wiley and Sons)*, 56(38):11634-11639 (2017) Copyright © 1999-2019 John Wiley & Sons, Inc. All rights reserved

Panel C reproduced with permissions from Mohr et al., Rational Engineering of Photoconvertible Fluorescent Proteins for Dual-Color Fluorescence Nanoscopy Enabled by a Triplet-State Mechanism of Primed Conversion, *Angew. Chem. Int. Ed. (John Wiley and Sons)*, 56(38):11628-11633 (2017) Copyright © 1999-2019 John Wiley & Sons, Inc. All rights reserved

Fig. R4 Comparison of the brightness of primed conversion FPs. **a** Single-molecule photon numbers of fluorescent proteins under primed conversion. The figure was adopted from Supplementary Figure 8 of [Turkowyd et al., *Angew. Chem. Int. Ed.* **56**, 11634 (2017)]. **b** Relative brightness (red) of 405-mediated mMaple3 (upper panel) and primed converting mMaple3 (lower panel). The figure was adopted from Supplementary Figure 3c(iii) of Turkowyd et al. again. **c** Single-molecule photon numbers of mEos2 under 405-nm-mediated activation and primed conversion. Primed conversion mEos2 showed lower photon counts than 405-activated mEos2. The figure was adopted from Figure 3c of [Mohr et al., *Angew. Chem. Int. Ed.* **56**, 11628 (2017)].

In addition to the photon numbers, UnaG offers several benefits compared to mMaple3, one of the brightest primed conversion FPs (pcFPs), as we emphasize through all the manuscript and response letter, in terms of small size (about half of GFP) that leads into higher labeling coverage (Fig. R6), refueling property for long-term imaging (Fig. R7), dark-to-green transition for streamlined, high-performance multicolor live-cell imaging (Figs.6, S18 and S20). All pcFPs, including mMaple3, share common molecular features such as molecular weight comparable to GFP, covalently linked chromophores, and green-to-red transition. Therefore, many of our experimental comparisons of mMaple3's performance in SMLM in comparison to UnaG intrinsically can be applied to other pcFPs. Also, we can deduce other pcFPs' properties by referencing the mMaple3 results. For instance, Dendra2 is the most commonly used pcFP for SMLM, but some of its SMLM performance such as labeling efficiency [Wang et al., *Proc. Natl. Acad. Sci.* **111**, 8452 (2014)] and photon numbers under primed conversion [Turkowyd et al., *Angew. Chem. Int. Ed.* **56**, 11634 (2017)] are reported as inferior to mMaple3. Thus, we can deduce Dendra2's labeling efficiency and photon numbers to be lower than UnaG.

Nonetheless, we found that the broad spectral window of pcFP (green-to-red emission as well as dual conversion lights in blue and red/infrared) is better to highlight than the low photons or other disadvantages because it makes pcFPs incapable of multicolor imaging modalities that UnaG can easily achieve. The pre-conversion emission in green overlaps with dark-to-green FPs and the post-conversion emission in red overlaps with dark-to-red FPs. By sequential primed conversion followed by UV-activation, a pcFP can be sequentially distinguished from a photoactivatable FP that cannot be activated by primed conversion such as PAmCherry [Mohr et al., *Angew. Chem. Int. Ed.* **56**, 11628 (2017); Virant et al., *Int. J. Mol. Sci.* **18**, 1524 (2017)]. Also, red PAINT probes were distinguished sequentially after imaging and bleaching pcFPs in Virant et al. However, these approaches of sequential multicolor imaging can neither observe different structures in a live cell at the same time nor collect a time-lapse movie composed of multiple SMLM frames in multiple color channels.

Far-red dyes may allow for simultaneous SMLM imaging with pcFPs, albeit the complication in labeling as we discussed in the point #1. However, the red excitation laser for far-red dyes can serve as efficient light source of primed

conversion [Klementieva et al., *Chem. Commun.* **52**, 13144 (2016)]. Klementieva et al. used 250 W/cm² of 640 nm for SMLM imaging of pcFPs, which is 1-2 orders of magnitude lower than the excitation intensity for live-cell SMLM imaging of Alexa Four 647. Therefore, the red laser for exciting a far-red dye can induce primed conversion of pcFPs at a significantly higher rate than needed for proper SMLM imaging, and therefore overly activate pcFPs. Or, the red laser for proper primed conversion rate can be insufficient for high-performance SML imaging of far-red dyes. Overall, it is highly challenging to find laser intensities, particularly for red laser, that give optimal on-switching rate of a pcFP and optimal off-switching rate of a far-red dye at the same time. Probably because of this reason, simultaneous multicolor imaging of pcFPs has not been demonstrated yet, to the best of our knowledge.

Parallel read-out of a green-to-red converting FP and a dark-red fluorophore has been done before and is a standard technique for structural SMLM studies, see e.g. this extensive SMLM work done in yeast: [https://www.cell.com/cell/fulltext/S0092-8674\(18\)30800-6](https://www.cell.com/cell/fulltext/S0092-8674(18)30800-6).

Here, we meant that simultaneous multicolor imaging of pcFPs “with primed conversion” has not been demonstrated. For primed conversion, green-to-red converting FPs require red light source that prohibits the application of far-red dyes. The cell paper that the reviewer mentioned imaged pcFP with 405-nm activation, not with primed conversion strategy.

Without considering primed conversion, most of photo-activatable FPs allow simultaneous two-color imaging with far-red dyes as the reviewer commented and as we have discussed from the first version of the manuscript. However, in the cell paper that the reviewer mentioned, SNAPtag ligands with Alexa Fluor 647 are delivered by permeation of live yeasts with detergent. Also, for mammalian cells, the same SNAPtag ligand requires electroporation or bead loading for temporary disruption of the membrane [Jones and Shim et al., *Nat Methods*, 2011]. In contrast, UnaG labeling works without permeating the cell membrane because external cell-permeable bilirubin can be simply added to the culture medium. Moreover, UnaG uniquely offers simultaneous, crosstalk-free three-color imaging capability (Fig. S20), which has not been demonstrated with green-to-red converting FPs, to the best of our knowledge.

In contrast, we demonstrated simultaneous multicolor imaging capacities of UnaG with simple filter-based approach, by utilizing the background-free dark-to-green transition of UnaG. In our previous revision, we demonstrated (1) two-color imaging with bright red membrane probes (Fig. 6), (2) FP-only two-color imaging with PAmCherry (Fig. S18b), (3) three-color SMLM imaging with Alexa Flour 647 (Fig. S18d), (4) simultaneous observation of structural evolution of mitochondrial matrix and inner membrane during division and fusion (Fig. 6e,f). All of these simultaneous multicolor imaging applications are near impossible for pcFPs to carry out due to the broad emission window in green-to-red and multiple conversion light sources in blue and red/infrared. Moreover, Virant et al. used 4 kW/cm² of 488-nm light for bleaching the residual Dendra2, which is ten times more intense than the excitation intensity of UnaG imaging. As a result, primed conversion multicolor SMLM can be more toxic to cells than UnaG imaging.

Finally, to highlight the more prominent advantage of UnaG that we already demonstrated experimentally in various combinations of colors, probes and applications in Fig. 6, S18 and S20, we deleted the phrase of mentioning low photons as major disadvantage of pcFPs in the main text and instead inserted a sentence that discusses the broad spectral window and the limit in multicolor capability that reads, “*primed conversion FPs convert the emission from green to red upon exposure to blue and red/infrared lasers, making simultaneous multicolor imaging near impossible*”.

I would like to explicitly acknowledge the authors work on simultaneous multi-color read-out. Simultaneous read-out is rare in multi-color SMLM works, most are sequential. UnaG is an excellent candidate for simultaneous multi-color SMLM imaging as discussed above. Nevertheless, multi-color imaging was not the scope of my original comment. Here I was interested in the quality of (single-color) UnaG images compared to other fluorophores. I was interested to know which probe gives me the highest quality for single-color SMLM images. That’s why I suggested the nuclear pore as a reference structure to compare fluorophores. A discussion, which combination of probes is best for multi-color imaging is a completely different (but also nevertheless important) discussion. But also here it would be important to know (when choosing UnaG as one of the two colors), what image quality one can expect for this color channel.

We thank the reviewer to recognize the advantage of UnaG in simultaneous multicolor imaging. To provide objective SLM imaging performance of UnaG, we presented SML images of various targets including vimentin and clathrin as reference structures (Fig. 4-6 and Fig. S7-S9), single-molecule photon number (in comparison with other blue-absorbing dyes, Fig. 4), localization precision (Fig. 4), multicolor and live-cell imaging capability (Fig. 5-6, Fig. S15-S20 and Movie S1). In the previous revision, we demonstrated the higher labeling coverage than mMaple3 (Fig. S11),

potentially due to the smaller size of UnaG. From all the above results along with the additional results in this revision highlighting UnaG's resistance to photobleaching (Fig. S14), we think the readers can judge the SML imaging performance for their dye selection. In particular, since the two-color combination of a green-to-red conversion FPs and a far-red dye has been well demonstrated as the reviewer pointed out with the cell paper on yeast [Mund et al, *Cell*, 2018], the single-color comparison of mMaple3 and UnaG (Fig. 4c, Fig. S11 and the new figure Fig. S14) would guide the readers on the image quality of the FP color channel to combine with a far-red dye.

4. I don't see any experimental proof for this sentence "We would like to emphasize that UnaG offers the best image quality among fluorescent proteins that are workhorses for live-cell imaging" in the manuscript. Even if there is a photon numbers measurement, the higher photon counts do not show in the images. So, please proof the utility of the tool on a reference standard in living cells, e.g. nuclear pore, microtubule hollow structure (giving a profile with a dip in the middle), FtnA oligomers, Glycine receptor and show that you obtain better image quality. Just an image of vimentin does not proof anything. There are high quality images on reference standards. Using reference standards is independent of the setup used or the ability of the experimenter – thus please perform this experiment. This is needed to introduce a new tool, e.g. the recent BC2 nanobody paper performed FtnA standard measurements vs. SNAP tag, the just uploaded back-to-back preprints from the Ries and Jungmann groups (<https://www.biorxiv.org/content/10.1101/582668v1>, <https://www.biorxiv.org/content/10.1101/579961v2>) show nuclear pore imaging with protein tags, dSTORM, GFP nanobodies, DNA-PAINT and FPs at very high quality. This is the current standard of the field the authors have to benchmark against, not images from publications more than 5 years ago.

Among the reviewer's suggested experiments, we benchmarked the BC2 nanobody paper [Virant et al., *Nat. Commun.* **9**, 930 (2018)] since it is published recently in the same journal. Following the reviewer's suggestion, we generated UnaG-FtnA and performed measurement of UnaG-FtnA along with SNAP-FtnA (Fig. R5). For SNAP-FtnA, we obtained 72.5% of completeness of labeling that is consistent to the published value in [Finan et al., *Angew. Chem. Int. Ed.* **54**, 12049 (2015)]. UnaG-FtnA measurement yielded 94.6% of completeness of labeling, consistent to near complete labeling of fluorescent proteins as shown for YFP-FtnA in Finan et al. This is consistent to the high vimentin labeling coverage in Fig. R6. Note that the completeness of labeling for Alexa Flour 647 when using SNAP-tag or BC2 nanobody is lower than fluorescent proteins such as YFP and UnaG probably because of incomplete conjugation reaction of the ligand or incomplete binding of the nanobody to the tag. UnaG's near complete labeling is likely due to the high binding affinity of bilirubin to UnaG ($K_d = 98$ pM). Therefore, despite the lower photon count of UnaG, UnaG has advantage in labeling efficiency in SMLM. We now added the FtnA result as Fig. S12 in the supplementary material.

Fig. R5 Completeness of labeling for UnaG and SNAP-tag labeling systems using FtnA-oligomers of 24 subunits. Mean values of FtnA-24mer fluorescence intensities are presented as percentage of theoretical estimation. **a** Representative image of FtnA-24mer cluster labeled with UnaG. **b** Single-molecule photon number distribution of UnaG in measurement conditions, presented

with the mean value. **c** Intensity distribution of FtnA-UnaG clusters. The mean value was used to calculate the completeness of labeling in **g**. **d-f** Analogous data set with **a-c** for SNAP-tag labeling system. **g** Measured completeness of labeling of UnaG and SNAP-tag that yielded 94.6% and 72.5% for UnaG and SNAP-tag, respectively.

In addition, the BC2 nanobody paper used vimentin as a major reference target. Fig. 1,2 of the paper among the total four images in the main text used vimentin labeled with various methods including PAmCherry, GFP and BC2 nanobodies. Therefore, vimentin SMLM images also serve as good benchmark for us. As we discussed in detail in the point #2 and #5, the labeling coverage of UnaG-labeled vimentin was better than mMaple3 and comparable to the BC2 nanobody results, indicating that the small size of UnaG helps to improve SMLM image quality in terms of labeling coverage.

I would like to thank the authors for this extra work which nicely quantitates the performance of the UnaG labeling and read-out. I am convinced that UnaG labeling performs better than the SNAP-tag as shown in this data. Maybe just one comment on individual spot brightness (adding to the previous comments above): The authors themselves have measured 1200 photons per UnaG. Here they show 222 AD-counts per UnaG which hints at an even lower photon number (e.g. the conversion in our setup is 1000 AD counts ~ 150 photons). Assuming that all measurements are done on the same setup, how can the authors explain these drastically different photon counts? Do they now maybe agree with me that spot photon counts are not easily comparable (see comment 3)?

We are happy to be able to convince the reviewer for the labeling performance of UnaG. Regarding the AD-counts issue, here we used different camera mode (16 bit) to other measurements (12 bit, sensitive), to properly detect the highly bright signal from a FtnA cluster in which 24 fluorophores simultaneously emit fluorescence, which cannot be measured in 12 bit mode due to saturation. Furthermore, to prevent the off-switching and photobleaching, the intensity was measured with a short exposure time of 10 ms, which is much shorter than the average off-switching time of UnaG (and also Alexa647) under acquisition condition. Thus, the measured AD counts in this result is totally different to the single-molecule photon number of UnaG in Fig. 4c.

5. I honestly currently see no advantage of UnaG in SMLM imaging quality in comparison to mMaple3 or mEos3.2. Please proof with an in vivo cell reference structure. The only advantage I see is something the authors don't show: All FPs need oxygen for maturation – UnaG works without and could thus be an ideal tag for anaerobic cultures. Thus, an application of this technique for exploring new biology in e.g. anaerobe archaea would be novel and cannot be done with FP tags.

To directly compare UnaG and mMaple3, we received the mMaple3-Vim plasmids from the authors of [Wang et al., *Proc. Natl. Acad. Sci.* **111**, 8452 (2014)] and performed mMaple3 imaging of vimentin structures (Fig. R6). Also, the image published in Wang et al. is included in Fig. R2b, next to the new UnaG image of vimentin in Fig. R2c. Following the quantitative comparison of SMLM images of vimentin labeled with various tags and fluorophores in [Virant et al., *Nat. Commun.* **9**, 930 (2018)], we measured the lengthwise labeling coverages of thin filaments in SMLM images obtained with UnaG and mMaple3. The average coverage was 76% for UnaG and 56% for mMaple3. The qualitative difference of more continuous fibrils in the UnaG image (Fig. R6a,b) is consistent to the quantitative difference in the values of labeling coverage (Fig. R6c). As we discussed in detail in the point #2 along with additional comparison to PAmCherry and BC2 nanobody, UnaG-labeled vimentin showed higher labeling coverage than mMaple3 probably due to the smaller size. Note that Wang et al. reported the labeling efficiency of mEos3.2 to be much lower than that of mMaple3. We added the labeling coverage comparison of mMaple3 and UnaG as Fig. S11 in the supplementary material.

Fig. R6 Direct comparison of SMLM images of vimentin fibers fused with UnaG and mMaple3 that are imaged in our laboratory. **a** SMLM images of vimentin labeled UnaG and mMaple3. Note that the two images were analyzed and rendered in the exact same way. Specifically, we rendered each image with normalized Gaussian with 10-nm pixel size in ThunderSTORM. **b** Traced and straightened vimentin fibers labeled with UnaG (top) and mMaple3 (bottom), obtained from **a** by using manual segmentation and straighten selection tools in ImageJ. **c** Lengthwise labeling coverage histograms obtained from 88 fibrils for UnaG and 52 fibrils for mMaple3 as exemplified in **b**. We followed the analysis method in [Virant et al., *Nat. Commun.* **9**, 930 (2018)]. Briefly, we manually traced thin, individual filaments and straightened the segmented images using ImageJ. Then, a custom MatLab code converted a straightened image into a binary image and calculated the fraction of covered area of the middle 3 pixels.

As an additional benefit of UnaG to mMaple3, we performed a 25-min-long time-lapse imaging of cytosolic UnaG and mMaple3 in Fig. R7. The UnaG localization numbers were virtually unchanged for more than 25 min probably because of the reversible fluorescence recovery of UnaG by the near infinite pool of free, non-fluorescent ligands at high concentration of 1 μM (blue line in Fig. R7). In contrast, mMaple3 localizations were exponentially (red line in Fig. R7) decreased possibly due to irreversible photobleaching of the chromophore that is covalently attached to the protein (Fig. S1). The mechanism of reversible switching of UnaG is quite unique because it stems from noncovalent, highly specific binding of the ligand to the protein (Fig. S1). In conventional FPs in SMLM including mMaple3, the photoconversion or photoswitching occur through the chemical change of the chromophore such as photo-isomerization or photo-oxidation. In fact, UnaG is the first and only FP with noncovalently bound chromophore [Kumagai et al., *Cell* **153**, 1602 (2013); Lee et al., *Chem. Sci.* **9**, 8325 (2018)]. Purely noncovalent interactions between the chromophore and the protein along with the use of highly concentrated pool of nonfluorescent ligands without complication of delivery or notable cell toxicity, both of which are quite unique for UnaG, contributed to negate the effect of photobleaching in long-term SML imaging as in Fig. R7. We include the time traces of localization numbers in the new Supplementary Fig. S14.

Fig. R7 Number of switching events (i.e. localizations) of UnaG and mMaple3 in extended periods of SML imaging. Localization numbers of cytosolic UnaG (blue) and mMaple3 (red) expressed in fixed Cos7 cells imaged at an exposure time of 40 ms. The UnaG samples were supplemented with 1 μM bilirubin in solution. While the switching events of mMaple3 diminished due to irreversible photobleaching, UnaG continued to produce switched-on spots thanks to the fluorescence recovery by the unbound bilirubin ligands in solution.

Being able to record for a long imaging period like in the authors 25 min movie is indeed an advantage. Nevertheless, being able to record for a long time is not the only parameter that determines image quality. I would like to recommend that the authors repeat the experiment of the supplemental figure 14 with mMaple3 and UnaG fused to a reference structure (e.g. vimentin) to be able to show comparative images of both approaches (see also previous comment on this). Here, UnaG should produce the denser and better resolved image compared to mMaple3 after 25 min, whereas when only using the localizations of the first 5 min both should be rather equal (judging from similar photon counts).

As we discussed in response to the reviewer previous comment, we performed new experiments with mMaple3 and UnaG fused to vimentin for a long recording period of 32 mins for replacing Fig. S14. Also, we quantified the labeling coverages and localization densities for the first half and the last half of the recording period. As the reviewer expected, UnaG's localization density decreased to 65% of that in 0-16 min in the last half, while mMaple3's localization density in 16-32 min decreased to 30%. Also, in the last half of the acquisition period, UnaG resulted in ~2 times higher labeling coverage than mMaple3 while the first half period resulted in small difference in labeling coverage.

Another benefit to point out is that UnaG allow for crosstalk-free, high-resolution, three-color imaging in live cells. We experimentally demonstrated simultaneous three-color live-cell imaging in Fig. S18d by combining UnaG, MitoTracker Red and Alexa Fluor 647, each of which offer the most photon outputs in its color panel of green, red or far-red. It was possible because UnaG fluorescence switch between the nonfluorescent dark state and green fluorescent state while

giving out the largest photon outputs among green switchable fluorophores. This application would be impossible with mMaple3 or mEos3.2 that photoconvert from green to red, covering two channels among the three color windows.

In summary, we experimentally showed UnaG's benefits in SMLM imaging in comparison to mMaple3 or mEos3.2. as follows. (1) The high labeling coverage of vimentin SMLM image would have come from the small size (Fig. R6). (2) During a long-term imaging session in which mMaple3 localization numbers diminished significantly due to the irreversible photobleaching, UnaG localization number stayed virtually unchanged owing to the reversible fluorescence and the nonfluorescent ligand (Fig. R7). (3) Simultaneous three-color live-cell imaging with high photons and little crosstalk is possible with UnaG (Fig. S18d), but not with mMaple3 or mEos3.2.

We appreciate the reviewer's suggestion for an application to anaerobic cultures. This would indeed highlight that UnaG's fluorescence does not rely on oxidation of amino acids constituting the chromophore that occurs in most fluorescent proteins. Also, the suggested application would demonstrate potential for opening a window for new biology. However, we intend to highlight UnaG's benefits for more general applications for appealing to more general public. As we listed above, there are a number of advantages of UnaG that would be useful for standard SMLM imaging, in particular for live-cell and multicolor applications.

6. I highly appreciate the fixed cell experiment with 0.2 μM of fresh bilirubin and 0.8 μM of pre-bleached bilirubin. I would recommend to add this figure to the SI.

We appreciate the reviewer's suggestion on the experiment that helped to eliminate the possibility of competitive inhibition by bleached bilirubin. We added the result in SI (Fig. S3) as the reviewer suggested.

7. This sentence is not experimentally shown, is it? "The genetically incorporated UnaG can be readily applied to live-cell SML imaging and offers fast tracking for the cellular dynamics due to the high photon numbers and easily controllable UV-free kinetics." At least not at the quality I would expect, I know of the 1s overreacting conditions experiment but this is not what I am looking for.

As an experimental proof, we obtained 2-sec snapshots of UnaG-labeled clathrin-coated pits as shown in Fig. R8 under non-overactivation condition. To accelerate the SMLM imaging speed, we increased the excitation intensity by ten times for promoting the off-switching rate fit for 10-ms exposure time while using 1 μM bilirubin for high on-switching rate. The 10-fold increase in excitation also helped to avoid overactivation (i.e. overlaps of single-molecule images in a camera frame). Since the on-off duty cycle can be represented as $1/[1+k_{\text{off}}/k_{\text{on}}]$, the 10-fold higher laser intensity reduced the on-off duty cycle by ten times and therefore the probability of image overlaps by ten times. As a consequence, we could resolve the ring-like projections of clathrin-coated pits with single-emitter fitting algorithm (Fig. R8).

Fig. R8 2-sec snapshot of UnaG-labeled clathrin-coated pits in live cell without using overactivation condition. The SMLM image was reconstructed from 200 frames at 10 ms exposure time that are analyzed with standard single-emitter fitting algorithm without further processing.

However, we would like to point that overactivation condition can be useful in live-cell imaging because the imaging speed can be accelerated without increasing the light dose that can induce cell stress. Indeed, "overactivation" was used for massively accelerating the imaging speed by using many state-of-the-art analysis software packages including deep learning algorithms [Marsh et al., *Nat. Methods* **15**, 689 (2018); Ouyang et al., *Nat. Biotech.* **36**, 460 (2018)]. These algorithms successfully demonstrated to reconstruct proper SML images from raw data containing highly overlapped single-molecule fluorescence images, and therefore allowed to significantly reduce the number of camera frames per SML image. Note that PAINT would face the challenge of high background to achieve overactivation condition

because high concentration of PAINt probe for increasing the on-switching rate would result in high background as demonstrated in Fig. R2c. Even with TIR illumination, the exponentially decreasing evanescent field can still excite a small fraction of fluorescent probes in solution. This resulted in a noticeable background in Fig. R2c with 1 nM PAINt probes under TIR excitation. In contrast, we could use 1 μ M bilirubin and Epi illumination to achieve background-free overactivated condition for reconstructing 1-sec SML snapshots of ER in Fig. 5 with HAWK, a recently published and publicly available software for artifact-free high-density localization analysis [Marsh et al, *Nat. Methods* **15**, 689 (2018)].

Thanks a lot for this extra work.

We also added this result as new Fig. S18 as the editors suggested, to support our argument of "fast tracking of cellular dynamics".

REVIEWERS' COMMENTS:

Reviewer #2 (Remarks to the Author):

I would like to thank the authors for their extra work during this revision. The manuscript was again substantially improved. I still would argue about some details and interpretations but this is beyond the scope of improving the manuscript. All claims of the authors follow a rational and even if I do not always fully agree, their argumentation is sound. Thus, this manuscript with all its extensive characterizations should be published.

Point by point response to the reviewer's comment

Manuscript #: NCOMMS-18-09874D

Title: "Bright Ligand-activatable Fluorescent Protein for High-quality Multicolor Live-cell Super-resolution Microscopy"

Authors: Jiwoong Kwon, Jong-Seok Park, Minsu Kang, Soobin Choi, Jumi Park, Gyeong Tae Kim, Changwook Lee, Sangwon Cha, Hyun-Woo Rhee, Sang-Hee Shim

Reviewer's Comment

I would like to thank the authors for their extra work during this revision. The manuscript was again substantially improved. I still would argue about some details and interpretations but this is beyond the scope of improving the manuscript. All claims of the authors follow a rational and even if I do not always fully agree, their argumentation is sound. Thus, this manuscript with all its extensive characterizations should be published.

We highly appreciate the reviewer for supporting our manuscript to be published. During the intensive revisions, all the reviewer's comments were clearly helped to notably improve our manuscript.